# Regulation of reactive oxygen species during plant immunity through phosphorylation and ubiquitination of RBOHD

DongHyuk Lee[1,6], Neeraj K. Lal[2,6], Zuh-Jyh Daniel Lin[1,3], Shisong Ma[2,4], Jun Liu[1,5], Bardo Castro[1], Tania Toruño [1], Savithramma P. Dinesh-Kumar[2] & Gitta Coaker [1✉]

Production of reactive oxygen species (ROS) is critical for successful activation of immune responses against pathogen infection. The plant NADPH oxidase RBOHD is a primary player in ROS production during innate immunity. However, how RBOHD is negatively regulated remains elusive. Here we show that RBOHD is regulated by C-terminal phosphorylation and ubiquitination. Genetic and biochemical analyses reveal that the PBL13 receptor-like cytoplasmic kinase phosphorylates RBOHD's C-terminus and two phosphorylated residues (S862 and T912) affect RBOHD activity and stability, respectively. Using protein array technology, we identified an E3 ubiquitin ligase PIRE (PBL13 interacting RING domain E3 ligase) that interacts with both PBL13 and RBOHD. Mimicking phosphorylation of RBOHD (T912D) results in enhanced ubiquitination and decreased protein abundance. PIRE and PBL13 mutants display higher RBOHD protein accumulation, increased ROS production, and are more resistant to bacterial infection. Thus, our study reveals an intricate post-translational network that negatively regulates the abundance of a conserved NADPH oxidase.

[1] Department of Plant Pathology, College of Agricultural and Environmental Sciences, University of California, Davis, CA 95616, USA. [2] Department of Plant Biology and the Genome Center, College of Biological Sciences, University of California, Davis, CA 95616, USA. [3]Present address: Donald Danforth Plant Science Center, St Louis, MO 63132, USA. [4]Present address: School of Life Sciences, University of Science and Technology of China, 230027 Hefei, China. [5]Present address: Institute of Microbiology, Chinese Academy of Sciences, 100101 Beijing, China. [6]These authors contributed equally: DongHyuk Lee, Neeraj K. Lal. ✉email: glcoaker@ucdavis.edu

The successful outcome of an immune response requires precise control of the timing, amplitude, and duration of the induced response. Spontaneous activation or inability to dampen signaling after immune activation can have a detrimental effect on the host. While significant progress has been made in deciphering downstream components that positively regulate plant immune signaling at the plasma membrane, our understanding of negative regulators and molecular strategies that deactivate receptor signaling after pathogen perception are less explored.

Plant innate immunity relies on plasma membrane localized pattern recognition receptors (PRRs) to provide resistance against a broad spectrum of pathogens[1]. PRRs recognize conserved microbial features, termed pathogen-associated molecular patterns (PAMPs), resulting in pattern-triggered immunity (PTI)[2,3]. Arabidopsis FLAGELLIN-SENSING 2 (FLS2) is one well-studied PRR that provides resistance against bacterial pathogens. FLS2 has an extracellular leucine-rich repeat (LRR) domain, a single helical transmembrane domain and an intracellular kinase domain. The LRR domain of FLS2 recognizes the 22 amino acid peptide of bacterial flagellin (flg22) as a PAMP[4]. Flg22 acts as "molecular glue" between FLS2 and its co-receptor BAK1 (BRASSINOSTEROID INSENSITIVE 1-associated receptor kinase 1)[5]. Following FLS2−BAK1 complex formation, a series of trans-phosphorylation events between various intracellular kinases associated with the PRR complex leads to activation of downstream signaling including a rapid burst of reactive oxygen species (ROS), activation of mitogen-activated protein kinase (MAPK) cascades, calcium influx, regulation of calcium-dependent protein kinases (CPKs), transcriptional reprogramming and phytohormone regulation[6].

Negative regulation of immunity is important to ensure appropriate response dynamics in the presence and absence of pathogens. This regulation occurs at multiple levels, including the primary receptor complex, downstream signaling components, and transcriptional regulators[7]. For example, the pseudokinase BAK1-Interacting Receptor-like kinase 2 (BIR2) inhibits FLS2–BAK1 complex formation upon flg22 perception[8]. The protein phosphatase PP2C38 associates with and negatively regulates the activity of botrytis-induced protein kinase 1 (BIK1), a major signaling kinase downstream of multiple PRR complexes[8,9]. Furthermore, the plant-specific trihelix transcription factor Arabidopsis SH4-Related3 (ASR3) functions as a transcriptional suppressor to modulate global gene expression upon flg22 perception[10]. The flexible nature of immune regulation allows plants to fine-tune the amplitude, timing, and duration of immune responses.

ROS generation during PTI occurs through nicotinamide adenine dinucleotide phosphate (NADPH) oxidase family members[11]. NADPH oxidases are also termed respiratory oxidase homologs (RBOHs) in plants that produce $O_2^-$ outside of plant cells. In addition to being pivotal for defense against pathogens, various RBOHs control a large number of developmental processes in response to both internal and external cues[12,13]. ROS generated by RBOHs act as secondary messengers for rapid transmission of local and long-distance signaling[14,15]. RBOHD is the primary Arabidopsis RBOH family member responsible for the ROS burst after PAMP perception[14].

RBOHD is a membrane-localized protein with six conserved transmembrane helices and intracellular cytosolic N- and C-termini. Upon PAMP perception, a rapid influx of $Ca^{2+}$ leads to conformational changes in RBOHD's N-terminal EF-hand motifs and phosphorylation by CPKs, resulting in ROS production[16−20]. Multiple intracellular protein kinases induce phosphorylation of RBOHD's N-terminus at activating residues to ensure ROS production including CPKs, BIK1, and SIK1[21−23]. RBOHD is also phosphorylated at activating residues by the receptor Doesn't Respond to Nucleotides 1 (DORN1) upon extracellular ATP perception[24]. While phosphorylation events that activate RBOHD are well characterized, our understanding of how ROS production mediated by NADPH oxidases is inhibited pre- and post-pathogen recognition remains unclear.

Multiple kinases in the Arabidopsis receptor like cytoplasmic kinase (RLCK) subfamily VII act as positive regulators of defense signaling. We have recently shown that the RLCK VII family member, PBS1-like kinase 13 (PBL13), functions as a negative regulator of PTI[25]. PBL13 possesses unique C-terminal repeats not found in other RLCKs. pbl13 mutant plants exhibit enhanced ROS production and are more resistant to virulent Pseudomonas syringae bacteria[25]. However, the underlying mechanism through which PBL13 inhibits PTI is not well understood.

Here, we show that PBL13 directly associates with and phosphorylates the C-terminus of RBOHD at conserved residues. PBL13 phosphorylation sites are important for RBOHD activity and stability. Using protein chip technology, we identified a previously uncharacterized RING domain E3 ubiquitin ligase, PIRE, which directly interacts with both PBL13 and RBOHD's C-terminus. PIRE ubiquitinates RBOHD and pire knockout (KO) lines exhibit enhanced RBOHD protein accumulation, higher PAMP-induced ROS burst and reduced bacterial growth. Mimicking a PBL13 phosphorylated residue at the C-terminus of RBOHD enhanced PIRE-mediated ubiquitination. PIRE constitutively associates with RBOHD, but is strongly phosphorylated upon flg22 perception. In summary, we demonstrate an intricate network of phosphorylation and ubiquitination that acts to regulate the NADPH oxidase RBOHD.

## Results

**PBL13 associates with and directly phosphorylates RBOHD.** PBL13 acts as a negative regulator of plant innate immune responses, including ROS production[25]. Since the majority of PTI-induced ROS in plants is produced by RBOHD and its homologs[13−15], we investigated if RBOHD can associate with PBL13. Previous work demonstrated that epitope-tagged variants of PBL13 (PBL13-3xFLAG) and RBOHD (GFP/HA/FLAG-RBOHD) are functional[21,25,26]. We performed immunoprecipitation (IP) between FLAG-tagged PBL13 and GFP-tagged RBOHD in Arabidopsis protoplasts. GFP-RBOHD was able to pull-down PBL13-3xFLAG (Fig. 1a). However, the membrane-localized control GFP-LT16b was unable to pull-down PBL13-3xFLAG (Fig. 1a). We also performed IPs with HA-tagged PBL13 and YFP-tagged RBOHD in Nicotiana benthamiana plants after transient expression. YFP-RBOHD associated with PBL13-3xHA but not YFP-LT16b-3xFLAG (Supplementary Fig. 1a). To test the association between PBL13 and RBOHD in Arabidopsis plants, we performed IPs using microsomal fractions from transgenic lines expressing PBL13-3xFLAG and antibodies against native RBOHD. We were able to identify association between PBL13-3xFLAG and RBOHD (Supplementary Fig. 1b, c). These results demonstrate that PBL13 associates with RBOHD in planta in the absence of pathogen perception.

We next investigated which region of RBOHD interacts with PBL13. MBP-tagged RBOHD's N-terminus (MBP-RBOHD-N; 1–376 amino acids) or RBOHD's C-terminus (MBP-RBOHD-C; 740–921 amino acids) were co-expressed with HIS-tagged PBL13 (HIS-PBL13) in E. coli and in vitro pull-downs were performed using amylose resin. Interestingly, we found that PBL13 preferentially interacts with the RBOHD-C compared to RBOHD-N (Fig. 1b). MBP alone did not interact with PBL13 (Fig. 1b). Consistent with the in vitro pull-down assay, yeast two-hybrid assays using RBOHD's C-terminus as bait also showed a specific

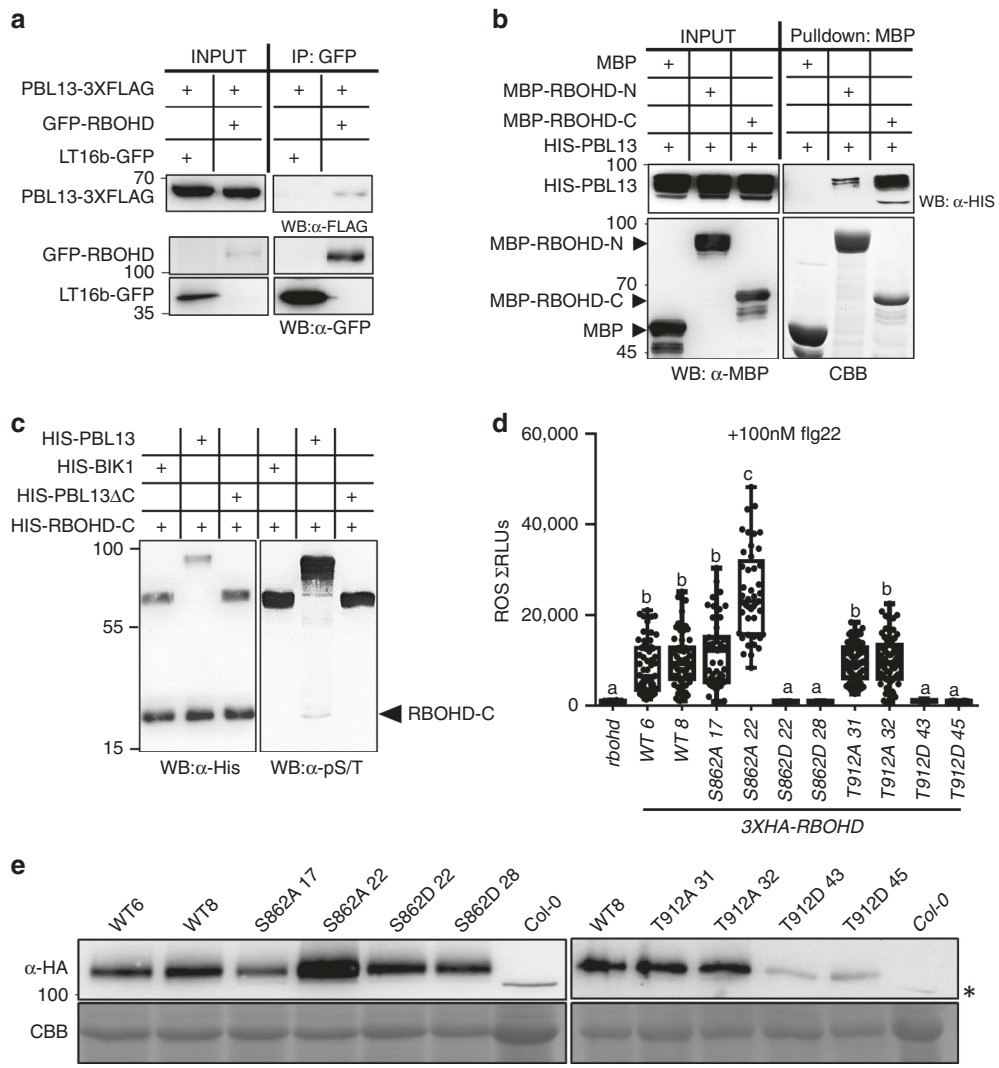

**Fig. 1 PBL13 associates with and phosphorylates the C-terminus of RBOHD. a** PBL13 interacts with RBOHD, but not the membrane-localized control LTI6B. PBL13-3xFLAG was co-expressed with GFP-RBOHD or LTI6b-GFP in Arabidopsis protoplasts and subjected to co-immunoprecipitation using anti-GFP antibodies. **b** PBL13 preferentially associates with the C-terminus of RBOHD (RBOHD-C) in vitro. RBOHD-N or RBOHD-C were co-expressed with HIS-PBL13 in *E. coli* and pulled-down with MBP agarose followed by immunoblotting with anti-MBP antibodies. CBB coomassie brilliant blue stained gel. **c** PBL13 phosphorylates RBOHD-C. In vitro phosphorylation was detected by incubating recombinant HIS-RBOHD-C with HIS-PBL13, HIS-BIK1 or HIS-PBL13ΔC followed by immunoblotting with anti-phospho S/T antibody. **d** Phosphomimetic RBOHD$^{S862D}$ and RBOHD$^{T912D}$ abolish flg22-mediated ROS production. The *rbohd* knockout was complemented with wild type, phosphonull or phosphomimetic mutants of *3xHA-RBOHD*. Transgenic lines were treated with 100 nM flg22 and the ROS burst was measured using a luminometer. For the box-plot, whiskers indicate minimum and maximum values, $n > 40$, line indicates the median, the box boundaries indicate the upper (25th percentile) and lower (75th percentile) quartiles. Statistical differences were determined by ANOVA with post-hoc Tukey HSD test, $\alpha = 0.05$, and letters indicate significant differences. **e** Detection of RBOHD expression by anti-HA immunoblotting in *Arabidopsis* transgenic lines shown in **d**. Astericks = nonspecific band. Twenty micrograms of protein was loaded for complementation lines and 60 μg loaded for Col-0.

interaction with PBL13, but not with the related RLCK PBS1 (Supplementary Fig. 1d).

Since PBL13 can interact with RBOHD-C and is an active serine/threonine kinase, we tested if PBL13 can directly phosphorylate RBOHD-C in vitro. PBL13 possesses unique domain architecture with an extended C-terminus that is absent in other Arabidopsis RLCKs. In order to determine the importance of PBL13's C-terminus for RBOHD phosphorylation, purified recombinant HIS-tagged PBL13 lacking its C-terminus (PBL13ΔC) and BIK1 were subjected to an in vitro kinase assay with RBOHD-C. BIK1 has been previously shown to interact with and phosphorylate RBOHD's N-terminus, but not its C-terminus[22]. His-PBL13ΔC and HIS-BIK1 were capable of autophosphorylation, but they failed to transphosphorylate HIS-

RBOHD-C (Fig. 1c). These results indicate that PBL13 associates with and specifically phosphorylates RBOHD's C-terminus. Furthermore, PBL13's C-terminus repeat region is required for RBOHD phosphorylation, indicating that PBL13's repeat region is either required for substrate specificity and/or transphosphorylation activity.

**PBL13 regulates RBOHD activity and stability.** To identify RBOHD residues phosphorylated by PBL13, we co-expressed RBOHD-C and PBL13 in *E. coli*, purified RBOHD-C and performed mass spectrometry (MS) analyses. The PBL13 catalytic dead mutant (K111A) and empty vector were used as controls. MS analyses of RBOHD-C revealed six serine and threonine

residues phosphorylated by PBL13 but not by PBL13[K111A] or the empty vector control (Supplementary Fig. 2a, Supplementary Table 1). The six phosphorylated RBOHD residues (S780, S862, S907, T910, T911, and T912) were mutated to alanine (6A) in order to assess their requirement for PBL13 phosphorylation. HIS-PBL13 was able to phosphorylate MBP-RBOHD-C, but not MBP-RBOHD-C[6A], indicating that these six sites are the major phosphorylation sites for this kinase (Supplementary Fig. 2b). Quantification of phosphorylation after incubation with PBL13 in vitro revealed that S862 had the highest level of phosphorylation (Supplementary Table 1). In order to test in vivo phosphorylation of RBOHD's C-terminus by PBL13, we generated an antibody recognizing phosphorylated RBOHD S862 (α-pS862) (Supplementary Fig. 2c, d). The pS862 antibody specifically detects phosphorylated RBOHD as the signal is eliminated after phosphatase treatment and anti-pS862 cannot detect the RBOHD S862A variant (Supplementary Fig. 2c, d). Immunoblotting with anti-pS862 revealed that RBOHD S862 phosphorylation in vivo is enhanced after co-expression with wild-type PBL13, but not the PBL13[K111D] mutant (Supplementary Fig 2e).

To investigate the importance of PBL13-induced phosphorylation of RBOHD-C, we performed a trans-complementation assay in *N. benthamiana* that has been used previously in the literature[18,21]. The *N. benthamiana RBOHD* homolog, *RBOHB*, was subjected to virus-induced gene silencing (VIGS) followed by transient expression of Arabidopsis RBOHD[18,21]. To investigate the activity of RBOHD variants, trans-complementation plants were treated with flg22. Trans-complementation with individual RBOHD phosphonull (S/T to A) and phosphomimetic (S/T to D) mutants in *N. benthamiana* revealed that Serine 862 (S862) and Threonine 912 (T912) play an important role upon flg22 perception (Supplementary Fig. 3a, b). Mimicking S862 and T912 phosphorylation inhibited the flg22-induced ROS burst (Supplementary Fig. 3). On the other hand, the S862A non-phosphorylatable mutant exhibited a significantly higher ROS burst with this assay (Supplementary Fig. 3). With the exception of T912D (see below), all RBOHD variants were robustly expressed in *N. benthamiana* (Supplementary Fig. 3).

To further investigate the biological function of S862 and T912, we complemented the Arabidopsis *rbohd* KO mutant with wild type (WT) RBOHD (RBOHD[WT]), S862D (RBOHD[S862D]), T912D (RBOHD[T912D]), S862A (RBOHD[S862A]), and T912A (RBOHD[T912A]) fused to a 3xHA epitope tag. We analyzed two independent transgenic lines for flg22-induced ROS burst. As expected, we were able to restore the flg22-induced ROS burst in the *rbohd* KO by complementing with *3xHA-RBOHD* (Fig. 1d)[26]. Lines expressing phosphonull RBOHD[S862A] exhibited differences in ROS production compared to wild-type RBOHD[WT]. RBOHD[S862A] line 17 exhibited enhanced, but not statistically significant, ROS production. On the other hand, RBOHD[S862A] line 22 exhibited a more robust and statistically significant increase in ROS production (Fig. 1d, e). The level of RBOHD[S86A] expression in line 22 was higher than wild-type RBOHD complementation lines (Fig. 1e). Taken together, a non-phosphorylatable mutant of RBOHD S862 results in a slightly enhanced ROS burst (Fig. 1d and Supplementary Fig. 3). On the other hand, ROS production in the phosphomimetic RBOHD[S862D] or RBOHD[T912D] lines was strongly inhibited and similar to the levels in the *rbohd* KO (Fig. 1d). These results establish that mimicking phosphorylation of S862 and T912 abolishes RBOHD-mediated ROS production.

When checking protein expression levels of RBOHD phospho-site mutants in *N. benthamiana* and Arabidopsis, we noticed that RBOHD[T912D] accumulated at lower levels compared to other mutants (Supplementary Figs. 3a, b and 1e). To test the possibility that PBL13 regulates RBOHD protein accumulation, we generated transgenic Arabidopsis lines expressing *RBOHD*

with a 3xHA tag in WT Col-0, *pbl13*, and *pbl13* complemented with *PBL13-3xFLAG*. RBOHD stability was enhanced in the *pbl13* KO compared to other lines (Fig. 2a, b). RT-PCR results showed that the transcription of *3xHA-RBOHD* was not affected in the *pbl13* KO (Supplementary Fig. 4). These results indicate that PBL13 negatively regulates RBOHD protein accumulation.

In order to determine if mimicking RBOHD phosphorylation at S862 or T912 affects the kinetics of the flg22-induced ROS burst, we investigated the curves for ROS measurements (Supplementary Fig. 5a, b). RBOHD[T912A] lines exhibited a slightly shifted curve with more robust ROS production at earlier time points compared with wild-type *RBOHD* complementation lines (Supplementary Fig 5a). Consistent with their low protein abundance, *RBOHD[T912D]* lines did not exhibit a robust ROS burst upon flg22 treatment (Supplementary Fig. 5a). Similarly, *RBOHD[S862D]* lines and the *rbohd* KO did not exhibit a detectable ROS burst upon flg22 treatment (Supplementary Fig. 5b).

**RBOHD is ubiquitinated and subjected to degradation**. Due to established cross-talk between phosphorylation and ubiquitination[27–29], we tested if RBOHD protein was subjected to ubiquitination. For this, we immunoprecipitated RBOHD from Arabidopsis plants expressing *3xFLAG-RBOHD* followed by anti-ubiquitin immunoblotting. 3xFLAG-RBOHD, but not the Col-0 negative control, displayed polyubiquitination in vivo (Fig. 2c). To further confirm the ubiquitination of RBOHD, 3xHA-RBOHD was expressed in *N. benthamiana* and total ubiquitinated proteins were purified by agarose beads conjugated to tandem ubiquitin binding entities (TUBE). As shown in Fig. 2d, e, a "smear" of high-molecular weight RBOHD was strongly detected from the TUBE-bound fraction but not after incubation with agarose beads. Taken together, these results demonstrate that RBOHD is ubiquitinated *in planta*.

Transmembrane proteins are frequently subjected to ubiquitination, which serves as a signal for sorting, endocytosis, and eventual vacuolar-mediated degradation[30]. In order to determine if RBOHD is regulated by vacuolar degradation, we inhibited vacuolar degradation by Concanamycin A (ConA) treatment[31]. ConA treatment of transgenic plants expressing 3xFLAG-RBOHD in the *rbohd* KO background significantly enhanced the accumulation of RBOHD (Fig. 2f, g). ConA treatment of *3xFLAG-RBOHD/rbohd* lines followed by anti-FLAG IP showed increased accumulation of total RBOHD and ubiquitinated RBOHD (Fig. 2h). Together, these results demonstrate that RBOHD is ubiquitinated in vivo and subjected to vacuolar-mediated degradation.

**The E3 ligase PIRE ubiquitinates RBOHD**. We next sought to identify proteins that can interact with PBL13 and regulate RBOHD stability. We screened a protein microarray containing ~5000 *Arabidopsis* proteins[32] to identify PBL13 interacting proteins. The microarray was incubated with V5-tagged PBL13 protein and positive interactions were detected by probing the array with CY5-tagged V5 antibody (Fig. 3a). We identified a previously uncharacterized E3 ubiquitin ligase, AT3G48070, as a positive interactor with PBL13 (Fig. 3b). AT3G48070, hereafter referred as PIRE (PBL13 interacting RING domain E3 ligase) is a typical RING domain E3 ubiquitin ligase with a putative zinc finger domain (Supplementary Fig. 6a).

Since RBOHD stability is regulated in a PBL13-dependent manner, we tested if RBOHD can interact with both PBL13 and PIRE. Our in vivo co-IP assays indicated that PBL13 could associate with PIRE in *N. benthamiana* and in *Arabidopsis* protoplasts (Fig. 3c and Supplementary Fig. 6b). Using Bimolecular Fluorescence Complementation (BiFC), we detected a

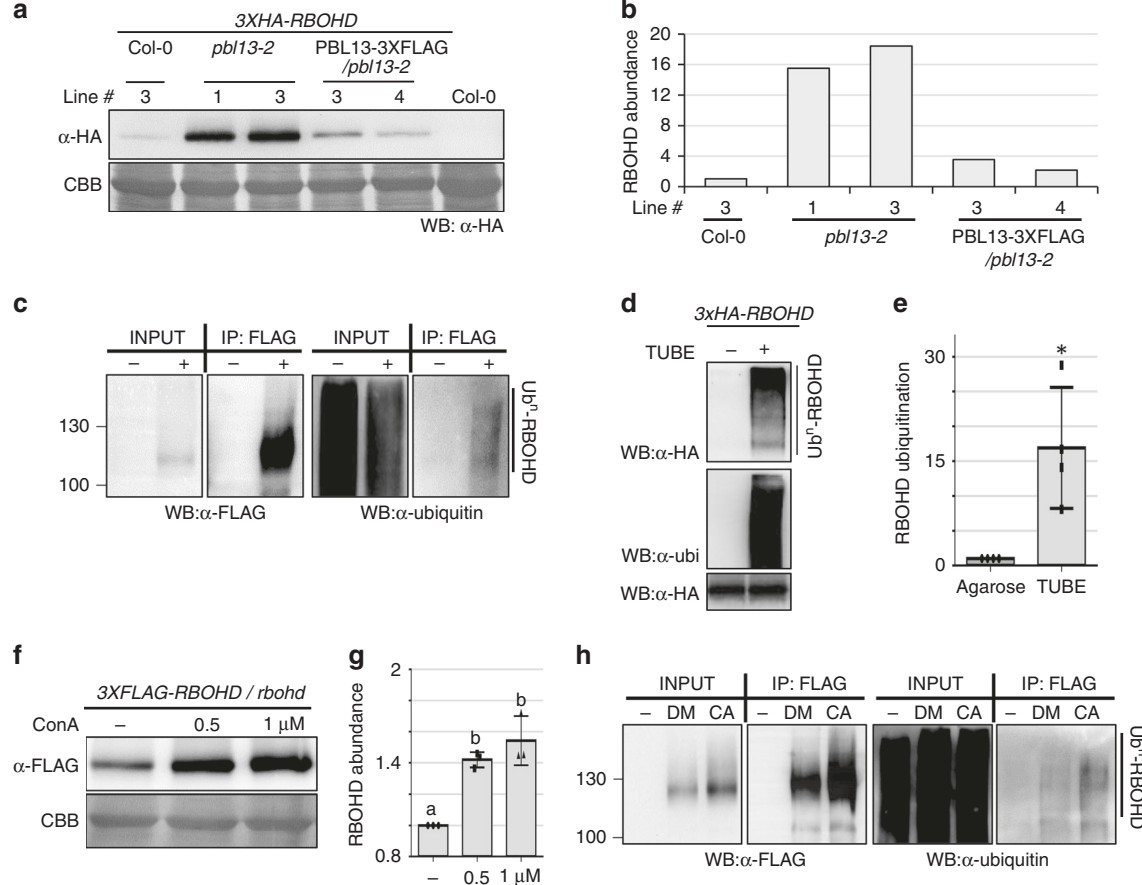

**Fig. 2 RBOHD is ubiquitinated and PBL13 suppresses its accumulation. a** PBL13 suppresses RBOHD protein accumulation. Arabidopsis transgenic lines expressing *3xHA-RBOHD* in Col-0, *plb13-2* knockout, or *pbl13-2* complemented with wild type *PBL13-3XFLAG* were generated, and RBOHD protein levels were detected by immunoblotting with anti-HA. **b** Quantification of bands from blot in **a**, with RBOHD abundance normalized to the intensity in Col-0. Experiments were repeated three times. **c** RBOHD is polyubiquitinated in vivo. Col-0 (−) or transgenic Arabidopsis lines expressing 3xFLAG-RBOHD (+) were used for immunoprecipitation with FLAG antibody. Polyubiquitination was detected by laddering after immunoblotting with ubiquitin antibody. **d** Ubiquitination of RBOHD is detected by Tandem Ubiquitin Binding Entity (TUBE). Ubiquitinated proteins were purified using agarose resin conjugated with TUBE (+) from microsomal fractions expressing 3xHA-RBOHD in *N. benthamiana*. Agarose resin (−) was used as a control. Ubiquitinated RBOHD (top panel) and input of RBOHD (bottom panel) were detected by immunoblotting with anti-HA. Immunoblotting using anti-ubiquitin (middle panel) shows specificity of the TUBE assay to purify ubiquitinated proteins. **e** Quantification of ubiquitination based on the TUBE assay. After incubation with agarose or TUBE, lanes from the HA immunoblot were quantified in Image Lab, and samples were normalized to the intensity of the agarose control. Statistical differences were determined by Student's *t*-test, $\alpha = 0.05$, $n = 4$ blots ± SD and asterisk (*) indicates significant differences. **f** RBOHD is subjected to vacuolar-mediated degradation. Transgenic lines expressing 3xFLAG-RBOHD treated with Concanamycin A (ConA) but not DMSO (−) exhibited enhanced accumulation of RBOHD. RBOHD protein was detected by immunoblot with anti-FLAG. **g** Quantification of the blot shown in **f**, with RBOHD abundance normalized to the intensity after DMSO treatment. Statistical differences were determined by ANOVA with post-hoc Tukey HSD test, $\alpha = 0.05$, $n = 3 \pm SD$ and letters indicate significant differences. **h** Inhibition of vacuolar degradation leads to enhanced accumulation of RBOHD and ubiquitinated RBOHD. Col-0 (−) and transgenic *3xFLAG-RBOHD* lines were treated with Concanamycin A (CA) or DMSO (DM), followed by anti-FLAG immunoprecipitation.

positive association between PBL13 and PIRE on the plasma membrane (Supplementary Fig. 6c). We also generated Arabidopsis transgenic lines expressing YFP-PIRE. We were able to detect an association in Arabidopsis between YFP-PIRE and RBOHD after anti-GFP IP in the absence of pathogen perception (Fig. 3d).

In order to determine if PIRE directly interacts with PBL13 and RBOHD, we performed yeast two-hybrid assays using a catalytic mutant of PIRE (C244S/C247S) as bait. We detected a specific interaction with RBOHD's C-terminus (RBOHD-C) and PBL13, but not with RBOHD's N-terminus (RBOHD-N) (Fig. 3e). In order to confirm a direct interaction between PIRE and PBL13/RBOHD-C, Glutathione S-transferase (GST) pull-down assays were performed using purified recombinant proteins. GST-PIRE was able to pull-down MBP-PBL13 and MBP-RBOHD-C (Fig. 3f).

The BIK1 RLCK, which targets RBOHD's N-terminus, failed to interact with PIRE (Fig. 3f). These data demonstrate that PIRE can interact with both PBL13 and RBOHD.

To test if PIRE is an active E3 ubiquitin ligase, we performed in vitro ubiquitination assays. Purified GST-PIRE was incubated with recombinant Yeast GST-UBE1 (E1), HIS-UBCH5a (E2), FLAG-tagged Ubiquitin and $ATP/Mg^{2+}$. Auto-ubiquitination was detected by probing with GST or FLAG antibodies (Fig. 4a). Elimination of E1 or E2 from the reaction abolished PIRE auto-ubiquitination. RING domain proteins contain eight zinc-coordinating cysteine residues. Mutation of the first two cysteines, C244 and C247, to serine abolished PIRE auto-ubiquitination (Supplementary Fig. 7a). To test if PIRE could directly ubiquitinate RBOHD-C, we performed in vitro ubiquitination with MBP-tagged or GFP-tagged RBOHD-C. PIRE could directly

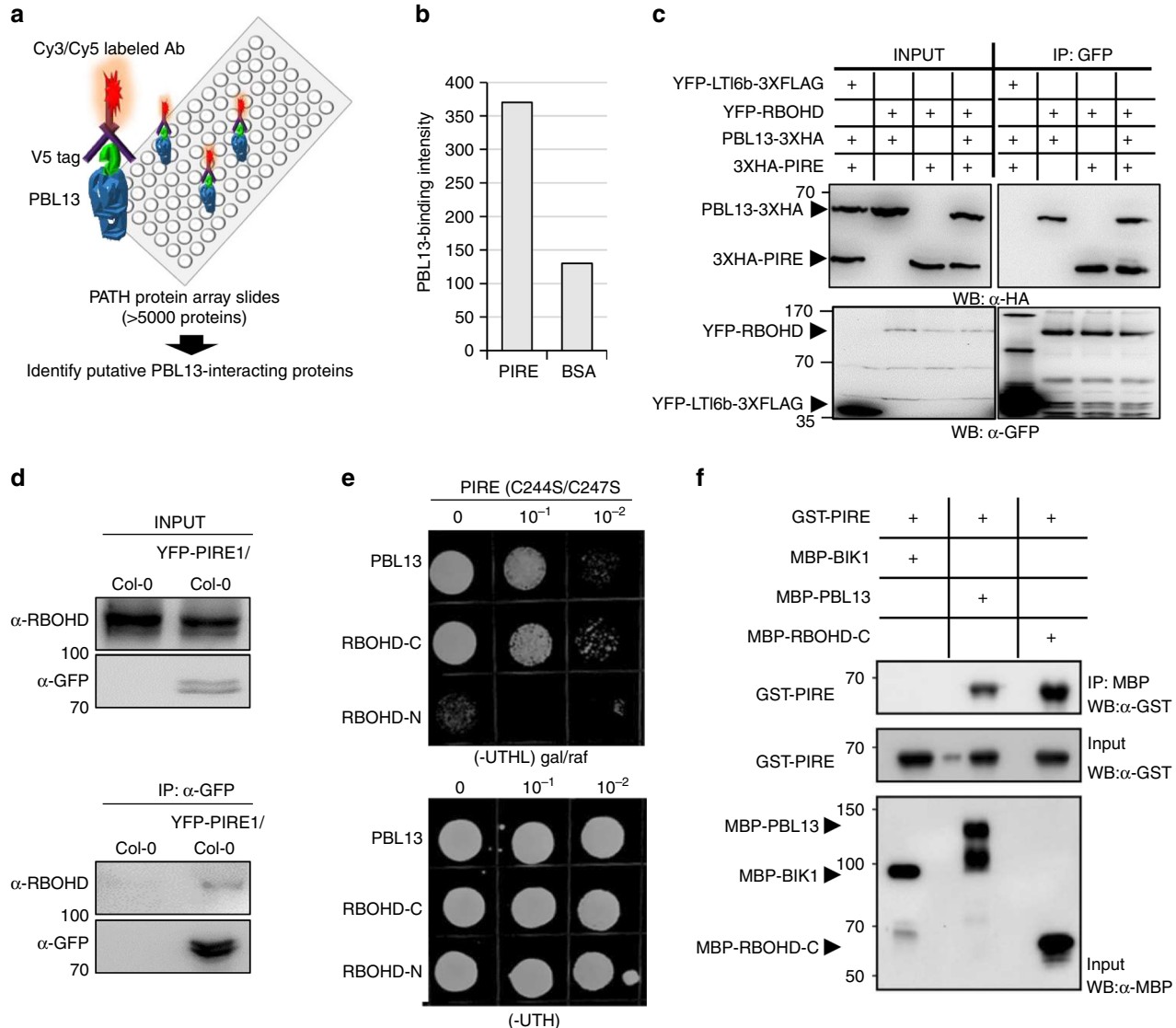

**Fig. 3 PBL13-interacting RING type E3 ubiquitin ligase (PIRE) interacts with PBL13 and RBOHD. a** Schematic representation of the Arabidopsis protein microarray. The PATH protein microarray with ~5000 Arabidopsis proteins was probed once with V5-tagged PBL13. **b** Quantification of PIRE and control protein (BSA)-binding intensity on the PATH array. **c** RBOHD associates with both PIRE and PBL13. YFP-LTI6b, YFP-RBOHD, PBL13-3xHA, and 3xHA-PIRE were transiently expressed in *N. benthamiana* in the indicated combinations. YFP-tagged proteins were subjected to anti-GFP immunoprecipitation and associations were detected by immunoblotting. **d** PIRE associates with RBOHD in Arabidopsis. Microsomal fractions were isolated from Col-0 or an Arabidopsis transgenic line expressing YFP-PIRE and subjected to co-immunoprecipitation using anti-GFP. **e** PIRE interacts with RBOHD-C and PBL13 by yeast two-hybrid assay. Serial dilution of yeast cells containing the indicated plasmids were spotted on -UTHL medium plates. The catalytic dead mutant of PIRE$^{C244S/C247S}$ was in the pTBS1 (bait) plasmid. PBL13, RBOHD-C, and RBOHD-N were screened for interaction with pTBS1-PIRE$^{C244S/C247S}$. -UTHL, SD medium lacking uracil, tryptophan, histidine, leucine with galactose (gal) and raffinose (raf) as a sugar source. **f** PIRE directly interacts with PBL13 and RBOHD-C. Purified recombinant MBP-RBOHD-C, MBP-BIK1, or MBP-PBL13 were incubated with GST-PBL13 and pulled-down with GST agarose. Protein input and pulled-down products were analyzed by immunoblotting.

polyubiquitinate RBOHD-C, as shown by higher molecular weight laddering of RBOHD-C detected by GFP or MBP antibodies and a smear of FLAG-tagged ubiquitin (Fig. 4b and Supplementary Fig. 7b). Polyubiquitination of RBOHD-C was compromised if E1, E2, or PIRE was removed from the reaction, demonstrating specificity of PIRE-mediated ubiquitination of RBOHD-C. A catalytic dead mutant of PIRE (C244S/C247S) was unable to polyubiquitinate RBOHD-C (Supplementary Fig. 7c). Since PBL13 interacts with PIRE, we tested if PBL13 is directly ubiquitinated by PIRE. PIRE cannot polyubiquitinate PBL13, demonstrating specificity for RBOHD (Supplementary Fig. 7d).

Next, we investigated the importance of specific lysine and threonine residues on RBOHD's C-terminus for PIRE-mediated

ubiquitination (Fig. 4c). We performed ubiquitination assays in vitro using GST-tagged PIRE with MBP-tagged RBOHD variants including WT, a C-terminal deletion of the last 15 amino acids (RBOHD$^{-15aa}$), a lysine mutant (RBOHD$^{K909AK013AK918A}$) and a phosphomimetic mutant (RBOHD$^{T910DT911DT912D}$). Compared to wild-type RBOHD-C, the phosphomimetic mutant exhibited enhanced ubiquitination. Furthermore, RBOHD$^{-15aa}$ and RBOHD$^{K909AK013AK918A}$ mutants abolished ubiquitination, indicating that C-terminal phosphorylation can promote PIRE-induced ubiquitination of RBOHD on conserved lysine residues. In order to verify the laddering in Fig. 4c was due to ubiquitination, MPB-tagged RBOHD variants were purified by amylose resin after performing ubiquitination reactions and the

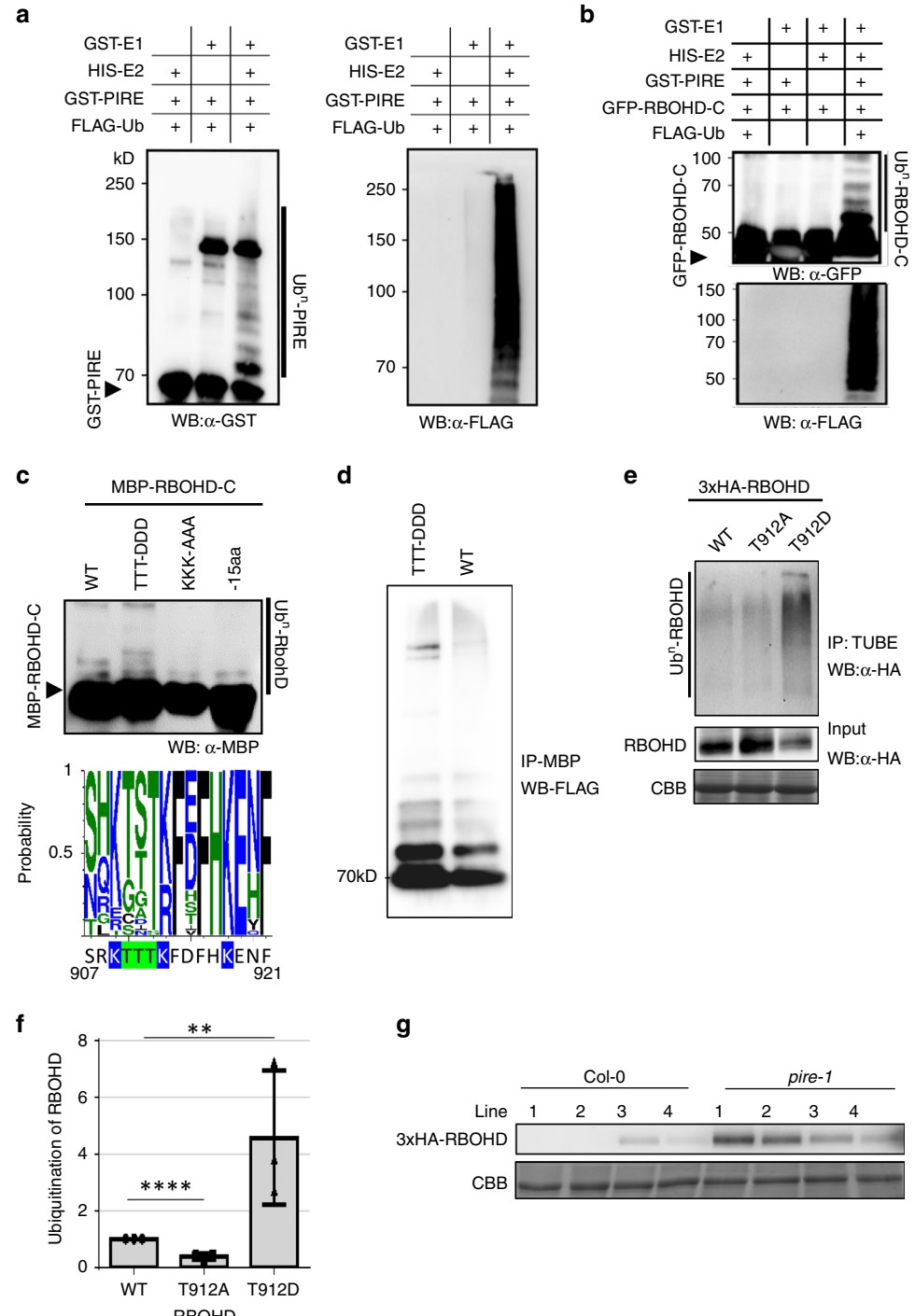

proteins subjected to anti-FLAG western blot to detect FLAG-tagged ubiquitin (Fig. 4d). After purification, the phosphomimetic mutant (RBOHD$^{T910DT911DT912D}$) exhibited enhanced ubiquitination compared to wild-type RBOHD (Fig. 4d). Similar to the in vitro assays, TUBE assays demonstrated that phosphomimetic RBOHD$^{T912D}$ exhibited enhanced ubiquitination, while phosphonull RBOHD$^{T912A}$ displayed significantly reduced ubiquitination in vivo (Fig. 4e, f and Supplementary Fig. 8). Both T912 and C-terminal lysine residues are conserved in other plant RBOHs, indicating that phosphorylation and ubiquitination may be a conserved mechanism for RBOH regulation (Fig. 4c).

Next, we investigated RBOHD protein accumulation in the *pire-1* KO and YFP-PIRE overexpression lines. The *pire-1* KO

exhibited enhanced accumulation of 3xHA-RBOHD compared to Col-0, but PIRE-overexpressing lines showed reduced accumulation of RBOHD (Figs. 4g and 5a). Furthermore, the accumulation of RBOHD in *pbl13* and *pire* KO lines was not enhanced after treatment with ConA to inhibit vacuolar degradation (Fig. 5b), indicating that PBL13 and PIRE could be involved in vacuolar-mediated degradation of RBOHD. We also investigated *RBOHD* transcription by quantitative PCR (qPCR). qPCR analyses did not identify significant differences in *RBOHD* expression in Col-0, *pire*, *pbl13*, and *pbl13 pire* (Fig. 5c). Together, these results demonstrate that PIRE polyubiquitinates RBOHD-C and regulates RBOHD protein accumulation *in planta*.

**Fig. 4 PIRE directly ubiquitinates RBOHD and suppresses RBOHD protein accumulation. a** PIRE is an active E3 ubiquitin ligase. Recombinant GST-tagged PIRE was subjected to in vitro ubiquitination assays with E1, E2, and FLAG-tagged ubiquitin (FLAG-Ub). Autoubiquitination of PIRE was evident by the appearance of higher molecular weight laddering in the GST immunoblot. Overall ubiquitination was detected by anti-FLAG immunoblotting. **b** RBOHD-C is directly ubiquitinated by PIRE. GFP-RBOHD-C was incubated with GST-PIRE in ubiquitination assay buffer. Direct ubiquitination of GFP-RBOHD-C was evident by higher molecular laddering detected by anti-GFP immunoblotting. **c** Mimicking phosphorylation of ROBHD's C-terminus enhances ubiquitination of RBOHD in vitro. Altered ubiquitination of RBOHD variants including wildtype (WT), RBOHD's C-terminal deletion (−15aa), lysine mutant$^{K909AK013AK918A}$ (KKK-AAA), and phosphomimetic mutant$^{T910DT911DT912D}$ (TTT-DDD) was detected by anti-MBP immunoblotting. Bottom: conservation of RBOHD's C-terminus across other plant RBOHs. The sequence logo consists of 29 RBOHs from the following plants species: *Arabidopsis thaliana* (10), *Oryza sativa* (7), *Solanum lycopersicum* (2), *Solanum tuberosum* (4), *Nicotiana benthamiana* (2), and *Nicotiana tabacum* (4). Mutated RBOHD residues are highlighted in blue or green. **d** To confirm the laddering in **c** is dependent on ubiquitination, the ubiquitination reaction was reconstituted with MBP-RbohD-C WT or the phosphomimetic mutant$^{T910DT911DT912D}$ with FLAG-Ubiquitin. The proteins were purified with Amylose beads and the elution was blotted with FLAG antibody. **e** Phosphomimetic RBOHD$^{T912D}$ enhances ubiquitination of RBOHD in vivo. Bottom panels: input, top panel TUBE immunoprecipitation. RBOHD's ubiquitination was detected by TUBE assay after expressing 3xHA-RBOHD variants (WT, T912A, and T912D) in *N. benthamiana*. **f** Quantification of ubiquitination of RBOHD phosphomutants. After incubation with TUBE, lanes from the HA immunoblot were quantified in Image lab and samples were normalized to the intensity of WT RBOHD ubiquitination. Statistical differences were detected by Student's *t* test, $\alpha = 0.05$, $n = 3$ blots ± SD and asterisk (*) indicates significant differences (**$P < 0.01$, ****$P < 0.0001$). **g** PIRE negatively regulates RBOHD protein stability. Arabidopsis transgenic lines expressing *3xHA-RBOHD* in Col-0 or the *pire-1* knockout were generated and RBOHD protein levels were detected by immunoblotting with anti-HA.

Since RBOHD$^{T912D}$ exhibits enhanced ubiquitination, we tested if alteration of this residue also affects association with PIRE. Co-IPs after co-expression of YFP-PIRE and RBOHD$^{T912D}$ or RBOHD$^{T912A}$ in *N. benthamiana* demonstrated that PIRE can associate with T912 variants (Supplementary Fig. 9). Similarly, co-IPs using Arabidopsis transgenic lines expressing YFP-PIRE before and after flg22 treatment demonstrated association between RBOHD and PIRE (Fig. 6a). We detected enhanced accumulation of RBOHD protein compared to Col-0 in both *pbl13* and *pire* KOs at a resting state and upon flg22 treatment (Fig. 6b). Together, these results demonstrate that PIRE constitutively associates with RBOHD, at the time points analyzed.

In order to determine the requirement of PIRE for the differential accumulation of RBOHD$^{T912A}$ and RBOHD$^{T912D}$ protein, we transfected Col-0 and *pire* protoplasts with wild-type *RBOHD*, *RBOHD$^{T912A}$*, and *RBOHD$^{T912D}$*. In the Col-0 background, RBOHD$^{T912A}$ accumulated to a higher level than wild-type RBOHD, while RBOHD$^{T912D}$ accumulated to a lower level than wild-type RBOHD (Supplementary Fig. 10a). In the *pire* KO, RBOHD$^{T912A}$ accumulated to a similar level as wild-type RBOHD in protoplasts (Supplementary Fig. 10b). However, RBOHD$^{T912D}$ still accumulated to a lower level than wild-type RBOHD in the *pire* KO (Supplementary Fig. 10b). qPCR demonstrated that RNA expression levels for *RBOHD*, *RBOHD$^{T912A}$*, and *RBOHD$^{T912D}$* were similar in both Col-0 and *pire1* protoplasts indicating that differences in accumulation are regulated at the protein level (Supplementary Fig. 10c, d). These results indicate that possibly additional components are involved in regulating RBOHD stability when T912 is constitutively phosphorylated.

**PIRE is strongly phosphorylated upon immune activation.** Next, we investigated if PIRE is post-translationally modified upon PAMP perception. Expression of 3xHA-PIRE in *N. benthamiana* or Arabidopsis resulted in two detectable bands by immunoblot, with a weak upper band (Fig. 6a, e). Interestingly, PIRE exhibits a pronounced mobility shift upwards after flg22 treatment (Fig. 6c). The shift is transient upon flg22 perception and similar in kinetics to ROS and MAPK activation (Fig. 6c). The upward mobility shift of YFP-PIRE is removed after lambda phosphatase (λPPase) treatment, indicating that PIRE is dynamically phosphorylated (Fig. 6d). We also detected an enhanced mobility shift in PIRE in Arabidopsis transgenic lines after flg22 treatment (Fig. 6e). We tested if PBL13 can directly

phosphorylate PIRE. GST-PBL13 autophosphorylates and was able to transphosphorylate MBP-RBOHD-C in an ATP-dependent manner (Supplementary Fig. 11). Surprisingly, we found that PBL13 is not able to phosphorylate PIRE (Supplementary Fig. 11), suggesting that other unknown protein kinase(s) dynamically phosphorylate PIRE upon immune activation.

**PBL13 and PIRE regulate ROS production during immunity.** To investigate the importance of PIRE during plant innate immunity, we identified two T-DNA insertion mutants in *PIRE* (Supplementary Fig. 12a, b). Genotyping and RT-PCR verified that both *pire-1* and *pire-2* are KOs (Supplementary Fig. 12c, d). The overall growth phenotype and morphology of the *pire* KOs were similar to Col-0 (Supplementary Fig. 12b). However, in response to flg22, *pire-1* and *pire-2* exhibited a higher ROS burst compared to WT Col-0 (Supplementary Figs. 4c and 12e). The ROS burst level in *pire* and *pbl13* KOs were similar, consistent with increase in RBOHD protein accumulation in these mutants (Supplementary Figs. 12e and 5a). Consistent with increased ROS production upon flg22 perception, *pire* KOs showed enhanced resistance to *Pseudomonas syringae pv. tomato* (*Pst*) DC3000 bacterial infection. *P. syringae* disease symptoms and bacterial titers were significantly reduced in *pire* KOs compared to Col-0, but similar to the *pbl13* KO (Supplementary Fig. 12f, g). While RBOHD accumulation is enhanced in *pire-1*, the accumulation of the FLS2 immune receptor was not altered (Supplementary Fig. 12h).

Next, we investigated the link between PBL13 and PIRE by generating *pbl13-2 pire-1* double KOs. The *pbl13-2 pire-1* double KOs exhibited enhanced RBOHD protein accumulation compared to Col-0, but similar to single KOs (Fig. 7a, b). The level of ROS burst, bacterial titers and disease symptoms after *Pst* DC3000 inoculation in the *pbl13-2 pire-1* double KOs were similar to the single KO lines (Fig. 7c–e). In addition, the *pire-1* KO exhibited enhanced MAPK activation after flg22 treatment, similar to *pbl13-2* (Supplementary Fig. 12i), which implies that PIRE may coordinate other immune responses with PBL13. Taken together, these results demonstrate that PBL13 and PIRE target the C-terminus of RBOHD to appropriately regulate ROS production (Fig. 8).

**Discussion**

Higher organisms utilize ROS as a versatile signaling molecule to regulate diverse cellular processes involved in growth, development,

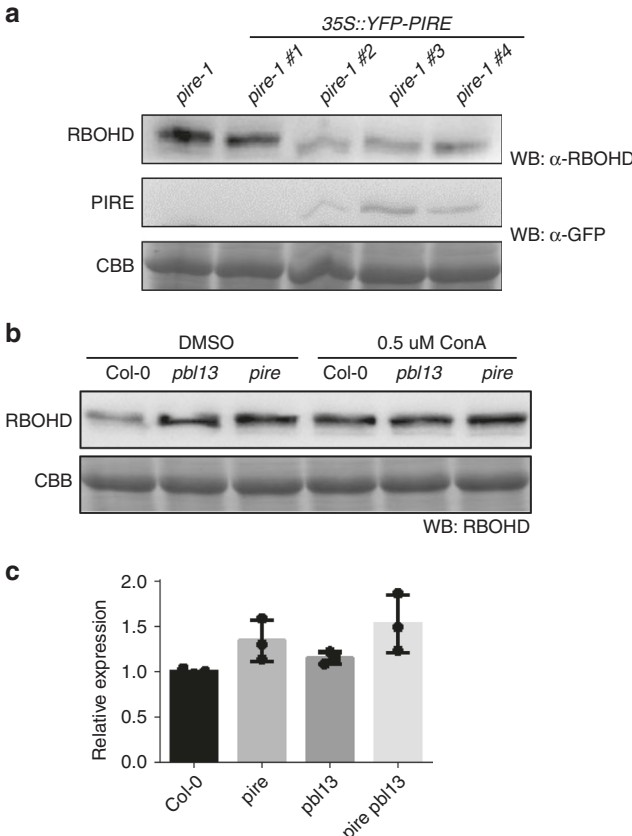

**Fig. 5 RBOHD protein accumulation is reduced in PIRE overexpression lines and enhanced in *pire* and *pbl13* knockout lines. a** Arabidopsis transgenic lines expressing *YFP-PIRE* in the *pire-1* knockout exhibited reduced accumulation of RBOHD compared to *pire-1* knockout line. The expression of RBOHD and PIRE were detected by anti-RBOHD and anti-GFP immunoblotting. CBB coomassie brilliant blue. **b** The accumulation of RBOHD in *pbl13* and *pire* knockout lines is not enhanced after inhibition of vacuolar degradation by Concanamycin A (ConA) treatment. Arabidopsis seedlings were treated with 0.5 μM ConA for 18 h and RBOHD was detected by anti-RBOHD immunoblotting. **c** Transcript expression of *RBOHD* in Col-0, *pire-1*, *pbl13-2*, and *pire-1/pbl13-2* genotypes. cDNA samples from 10 days old plants were subjected to qPCR. Values represent relative expression compared to Col-0, means ± SD ($n = 3$). Data were analyzed using the ΔΔcT method, and normalized against Arabidopsis *ELONGATION FACTOR* 1α (AT2G18720). No significant differences between genotypes were detected by ANOVA, $\alpha = 0.05$.

and stress responses[33–35]. NADPH oxidase-mediated ROS production is a conserved immune response in plants and animals. ROS is directly antimicrobial as well as an essential signal for plant innate immune signaling. The level of ROS should be tightly regulated to avoid cellular damage due to excess ROS production. Our results reveal dual post-translational modifications appropriately regulate RBOHD protein levels.

Similar to mammalian NADPH oxidase proteins (NOXs), all RBOHs have C-terminal FAD-binding and NADPH-binding sites and a functional oxidase domain. Plant RBOHs also have N-terminal Ca²⁺-binding EF-hand motifs[36]. Notably, RBOHD is positively regulated by N-terminal phosphorylation on S22, T24, S39, S163, S339, S343, and S347 by receptor-like kinases (RLK), CPKs, and RLCKs[18–22,24]. Our results demonstrate that RBOHD is negatively regulated by C-terminal phosphorylation on S862 and T912. Both phosphorylation mimics of RBOHD (S862D and T912D) exhibited a compromised ROS burst upon flg22

perception. However, the T912D mutation, but not S862D, induced decreased accumulation of RBOHD, suggesting that phosphorylation at distinct residues can affect RBOHD stability or activity. This dual regulation could ensure precise control of RBOHD protein accumulation at a resting state through phosphorylation of T912 as well as robust inactivation of the enzyme through S862 phosphorylation. Consistent with this hypothesis, Arabidopsis lines expressing RBOHD^T912A exhibited shifted ROS kinetics upon flg22 treatment, with enhanced ROS production at earlier time points compared to wild-type RBOHD lines. In contrast, mimicking S862 phosphorylation completely inhibited ROS production. In this scenario, inhibition of NADPH oxidase activity by S862 phosphorylation could be dynamically reversed by protein phosphatase(s).

Plants must be able to rapidly and appropriately respond to biotic stresses. As a result, many plant receptors and critical signaling nodes are post-translationally regulated. ROS production upon pathogen challenge regulates multiple processes, including callose deposition, systemic signaling, and stomatal closure[14,22,37]. Therefore, we propose that *pbl13* and *pire* KO lines as well as RBOHD phosphorylation mimics will have phenotypes in multiple ROS-regulated immune processes. Given that RBOHD is required for guard cell closure in response to pathogen perception[22], we propose that *pbl13* and *pire* could exhibit more robust and sustained stomatal closure in response to pathogen infection, resulting in strong disease resistance. Given the importance of NADPH oxidases for defense responses and the conservation of RBOHD's C-terminal residues, it is plausible that multiple kinases and E3 ligases will target RBOHs. Recently, the cysteine-rich receptor like kinase CRK2 was found to target RBOHD-C for phosphorylation, including S703, where mimicking phosphorylation enhanced RBOHD activity[38]. CRK2 also phosphorylates RBOHD at position S862 in vitro, and an S862A variant exhibited increased NADPH activity in HEK293 cells, a result that is consistent with our *in planta* data[38]. Thus, it is likely that the C-terminus of RBOHD will be subjected to convergent regulation by multiple kinases to ensure appropriate ROS production.

Different post-translational modifications on the same protein molecule allows for specificity and diversity in signaling[39]. Phosphorylation and ubiquitination are two of the most important post-translational modifications in eukaryotes. Cross-talk between phosphorylation and ubiquitination can diversely influence protein activity, abundance, and subcellular localization[39]. Our results show that mimicking T912 phosphorylation enhances RBOHD's ubiquitination in vivo. Furthermore, the *pbl13*, *pire*, and *pbl13 pire* double KOs all exhibit similar phenotypes including enhanced RBOHD accumulation, higher flg22-induced ROS burst and decreased bacterial growth. These data indicate phosphorylation and ubiquitination work in concert to regulate RBOHD in the absence of pathogen infection.

Molecular crosstalk between phosphorylation and ubiquitination has been demonstrated for a few key immune regulators. Experimental evidence for direct ubiquitination of plant receptor like kinases through a respective E3 ubiquitin ligase has been demonstrated for the flagellin receptor FLS2, chitin receptor LYK5, and brassinosteroid receptor BRI1[28,40,41]. Furthermore, all three receptors are ubiquitinated by the same set of U-Box E3 ligases, PUB12 and PUB13. NPR1 is a receptor for the defense hormone salicylic acid, which results in transcriptional reprogramming towards defense[29,42]. In order to appropriately regulate the timing of defense responses, NPR1 is phosphorylated by an unknown plant kinase, followed by ubiquitination through Cullin3 E3 ubiquitin ligases[29,43].

Proteolysis of ubiquitinated substrates occurs by the 26S proteasome or the vacuole depending on specific ubiquitin-linkages

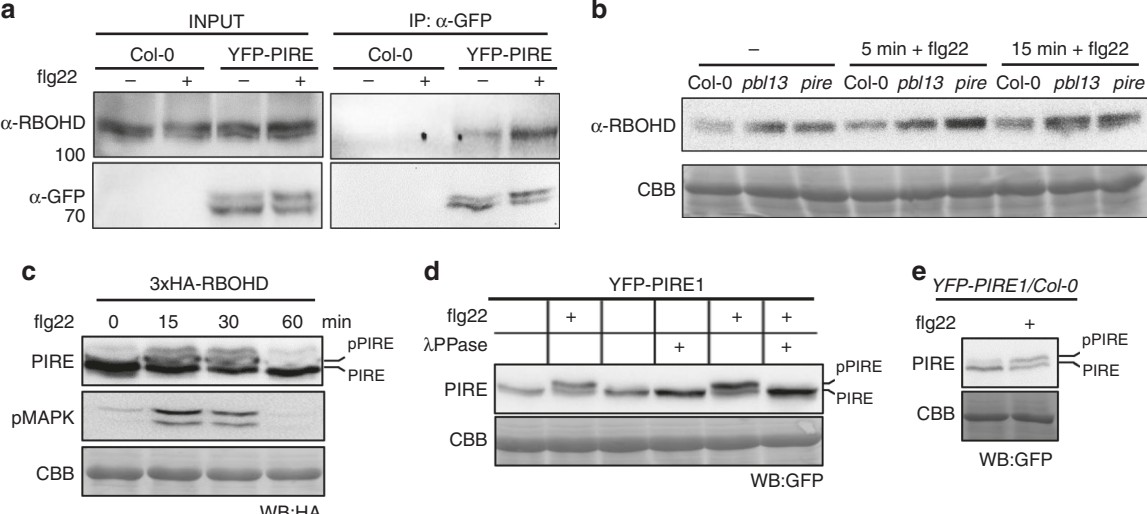

**Fig. 6 PIRE constitutively associates with RBOHD but is dynamically phosphorylated after immune activation. a** PIRE associates with RBOHD after immune activation. YFP-PIRE-expressing Arabidopsis seedlings were treated with flg22 for 15 min. Microsomal fractions were isolated and subjected to co-immunoprecipitation using anti-GFP antibodies. **b** RBOHD expression is enhanced in *pbl13* and *pire* knockout lines at a resting state. RBOHD expression was detected by immunoblotting with anti-RBOHD after flg22 treatment. **c** PIRE mobility shift is detected upon PAMP perception. 3xHA-PIRE was expressed in *N. benthamiana* followed by flg22 treatment. Immune activation by flg22 was confirmed by anti-pMAPK immunoblotting. **d** PIRE is phosphorylated in vivo. 3xHA-PIRE phosphorylation was induced by flg22 treatment in *N. benthamiana*. PIRE's phosphorylation is inhibited by lambda phosphatase (λPPase) treatment. **e** PIRE is phosphorylated in an Arabidopsis transgenic line expressing YFP-PIRE after flg22 treatment.

and subcellular localization[44]. Ubiquitinated transmembrane proteins are endocytosed, followed by sorting, trafficking, and removal through vacuolar degradation[44]. In plants, plasma membrane localized RLKs sensing extracellular stimuli coordinate development, growth, and immunity. Several RLKs undergo internalization through the endocytic pathway[45]. Upon BR perception, the *Arabidopsis* BR receptor BRI1 phosphorylates PUB13 E3 ligase at S344, which promotes the interaction between BRI1 and PUB13, and enhances PUB13's ubiquitin ligase activity, leading to BRI1 ubiquitination[28]. *pub12* and *pub13* KOs exhibited reduced BRI1 ubiquitination and internalization, demonstrating that ubiquitination of BRI1 by PUB12 and PUB13 is a key step in BRI1 endocytosis and degradation[28]. PUB12 and/or PUB13 also mediate ubiquitination of PRR RLCKs, such as FLS2 and LYK5, leading to degradation of the receptor proteins and inhibition of downstream signaling[40,41]. Other PRRs including EFR, PEPR1, and CLV1 are also endocytosed and routed to the vacuole after ligand perception, which indicates that ubiquitination and subsequent vacuolar-mediated degradation may be a common mechanism to remove activated PRRs[28,46–48]. Similar to PRR endosomal trafficking upon ligand perception, the receptor like protein Cf-4 is endocytosed and subjected to vacuolar degradation upon perception of the fungal Avr4 effector[49].

Experiments using single-particle analysis of clathrin- and microdomain mutants have demonstrated that both processes are important for RBOHD membrane distribution and RBOHD is endocytosed at a resting state[50]. Our results reveal that RBOHD is ubiquitinated at a resting state, and RBOHD protein accumulation increases after incubation with ConA, an inhibitor of vacuolar degradation[31]. PBL13 and PIRE also promote vacuolar-mediated degradation of RBOHD, as the single and double KO lines are no longer sensitive to ConA treatment. Furthermore, RBOHD-C can be directly ubiquitinated by PIRE. Post-translational regulation of RBOHD protein abundance should enable rapid and precise modulation of ROS generation in response to pathogen infection. Although PIRE constitutively associates with RBOHD, PIRE is strongly phosphorylated upon flg22 perception. Thus, PIRE may be involved in dynamically regulating RBOHD stability not only at a resting state but also after initial immune activation in order to fine-tune defense responses. The kinase(s) responsible for PIRE phosphorylation is currently unknown, as PIRE is phosphorylated in *Nicotiana*, which lacks PBL13, and PBL13 cannot phosphorylate PIRE in vitro.

ROS regulates diverse immune responses and is mainly produced by RBOHD and its homologs in plants during PTI. The PBL13-mediated phosphorylation sites, T912 and S862, are highly conserved in other plant RBOHs. The conservation of RBOHD phosphorylation sites implicates that phosphorylation and ubiquitination may be a conserved mechanism for regulation of RBOH family members, potentially across diverse plant species. It is possible that PBL13 is involved in regulating different RBOH homologs in *Arabidopsis*. However, the PBL13 kinase is only found in the *Brassicacea*. Alternatively, phosphorylation of distinct RBOHs at sites corresponding to S862 and T912 could be a conserved mechanism of negative regulation, with phosphorylation mediated by diverse protein kinases and E3 ligases.

It is likely that both PBL13 and PIRE possess additional substrates aside from RBOHD. Both *pbl13* and *pire* KO lines exhibited enhanced MAPK activation upon flg22 perception[25]. As the *rbohd* KO line does not alter MAPK activation, it is possible that PBL13 and PIRE target components related to MAPK phosphorylation upon pathogen perception[25,51]. Several RLCKs can directly link activation of PRR receptors to MAPK signaling[52,53]. Future investigation of the role of PBL13 and PIRE1 could uncover a novel mechanisms regulating immune signaling and extend our understanding of how these two major post-translational modifications connect to regulate signal transduction.

## Methods

**Plant materials and growth conditions**. *Arabidopsis thaliana* seeds were stratified at 4 °C for 2 days and sown in soil. Plants were grown in a controlled environmental chamber under the following conditions: 23 °C, 70% relative humidity, a 10-h light/14-h dark photoperiod, and a light intensity of 85 μE/m²/s. *N. benthamiana* plants were grown in the growth chamber under the following

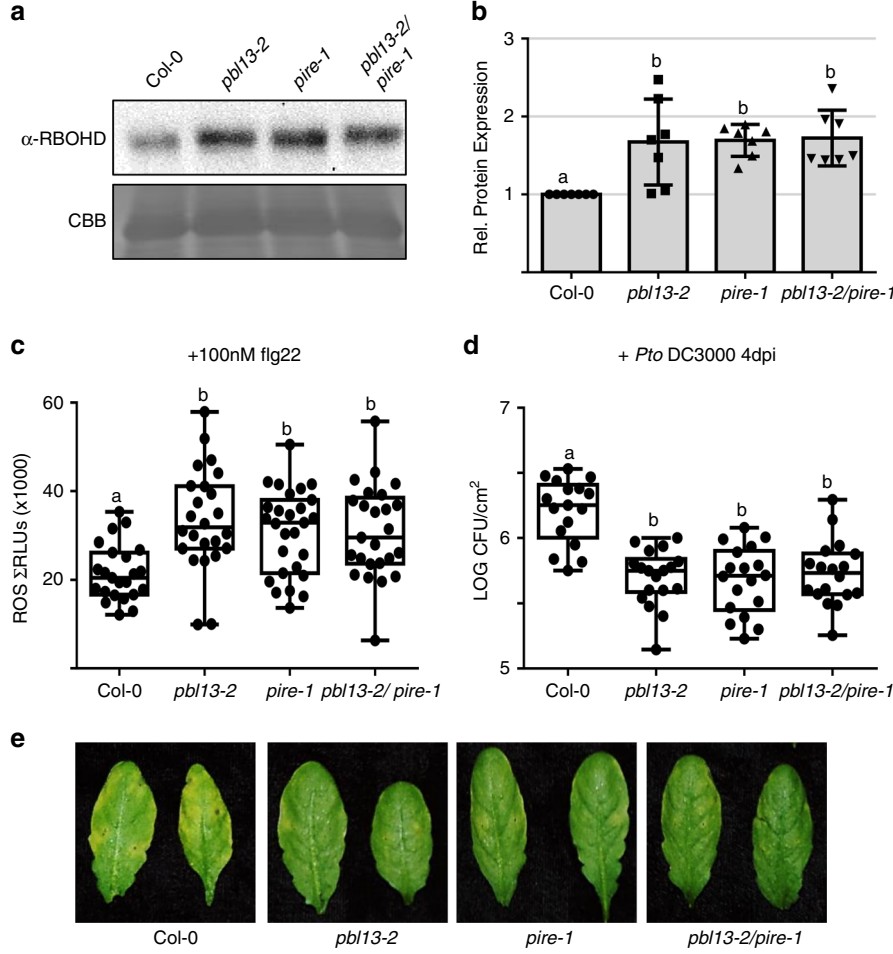

**Fig. 7 PBL13 and PIRE negatively regulate defense responses. a** The expression of RBOHD in *pbl13, pire*, and *pbl13::pire* knockout lines. RBOHD expression was detected by anti-RBOHD immunoblotting. **b** Quantification of RBOHD abundance in different genetic backgrounds. Expression was normalized to Col-0. Statistical differences were determined by ANOVA with post-hoc Tukey HSD test, $\alpha = 0.05$, $n = 7 \pm$ SD and letters indicate significant differences. **c, d** ROS burst **c** and bacterial titer **d** in Col-0, *pbl13, pire*, and *pbl13::pire* knockout lines. For the box-plot, whiskers indicate minimum and maximum values, line indicates the median, the box boundaries indicate the upper (25th percentile) and lower (75th percentile) quartiles, $n > 23$ for ROS burst and $n = 18$ for bacterial titers. Statistical differences were detected by ANOVA over all experiments with post-hoc Tukey HSD test, $\alpha = 0.01$, and letters indicate significant differences. **e** Disease symptoms after *Pto* DC3000 infection on indicated genotypes 5 days post-inoculation (dpi).

conditions: 25 °C, 50% relative humidity, a 14-h light/10-h dark photoperiod, and a light intensity of 180 µE/m²/s.

**Plant genotypes and transgenic Lines.** *A. thaliana* ecotype Columbia (Col-0) was used as the WT. *Arabidopsis* T-DNA insertion lines in the Col-0 background for *pire*-1 (SALK_079510) and *pire*-2 (SALK_138672) were obtained from the Arabidopsis Biological Resource Center (ABRC). T-DNA insertion mutants were genotyped by PCR using T-DNA and gene-specific primers. All primers used in the study are shown in Table S2. To generate *pbl13-2/ pire-1* double KOs, *pbl13-2* and *pire-1* were crossed by emasculation of the flowers and manual pollination. Homozygous double mutants were identified in the F2 generation by genotyping for the *pbl13-2* and *pire-1* alleles. RBOHD expression in single and double KO lines of *PBL13* and *PIRE* were determined by immunoblotting using anti-RBOHD at a concentration of 1:2000 (Agrisera), followed by secondary anti-rabbit-HRP at a concentration of 1:3000 (Biorad). All immunoreactive bands in this study were visualized by chemiluminescence (Super Signal West Femto Chemiluminescent Substrate, Pierce). Chemiluminescence was detected using the Bio-Rad Chemidoc system. Signal intensity was quantified using Image Lab (Bio-Rad).

The *pbl13-2* KO, PBL13 #8-3 complementation line, *rbohd* KO, and *npro::3xFLAG-RBOHD/rbohd* complementation line used in this study were previously described[18,21]. To generate constructs for *RBOHD* phosphorylation variants, *RBOHD* cDNA was PCR amplified and directionally cloned into pENTR/D-TOPO (Invitrogen). Individual *RBOHD* residues were mutated to alanine (A) and aspartic acid (D) to generate phosphorylation null and mimics using PCR-based mutagenesis. *RBOHD* WT and corresponding mutagenized entry clones were recombined into the pGWB15 binary vector by LR reaction, resulting in *35S::3xHA-RBOHD*. The *rbohd, pbl13-2*, PBL13 #8-3, and *pire-1* mutants were transformed with wild-type *RBOHD* or phosphorylation variants using the floral dip method[54]. Transformants were selected on ½ MS media supplemented with 50 µg/mL hygromycin. To evaluate RBOHD expression, total protein was isolated from transgenic lines by grinding in 2× laemmli buffer. Protein samples were separated by SDS–PAGE and immunoblotting was conducted using anti-HA-HRP (Sigma, 12013819001, clone 3F10) at a concentration of 1:1000.

To generate constructs for *YFP-PIRE*, *PIRE* (AT3g48070) cDNA was PCR amplified and directionally cloned into pENTR/D-TOPO (Invitrogen), and the sequenced entry clone was recombined into the pEarleyGate 104 destination vector by LR reaction, resulting in *35S::YFP-PIRE*. Col-0 and *pire-1* KOs were transformed with *35S::YFP-PIRE* as described above. Transformants were selected on ½ MS media supplemented with 30 µg/mL BASTA. To evaluate native RBOHD and YFP-PIRE expression, total protein was isolated from transgenic lines by grinding in 2× laemmli buffer. Immunoblotting was performed using anti-RBOHD at a concentration of 1:2000 (Agrisera, AS15 2962) or anti-GFP-HRP at a concentration of 1: 3000 (Miltenyi Biotec, 130-091-833, clone GG4-2C2.12.10).

To evaluate RBOHD, RBOHD^T912A, and RBOHD^T912D expression in *pire*, mesophyll protoplasts were isolated from Col-0 and *pire-1*. Mesophyll protopasts were isolated from 3-week-old plants by incubating leaf strips in 20 mM MES (pH 5.7), 10 mM CaCl₂, 1.5% cellulase R10 (Yakult Pharmaceutical), and 0.4% Macerozyme R10 (Yakult Pharmaceutical) for 3 h at room temperature[55]. Protoplasts were incubated on ice prior to transfection, concentrations were adjusted to $1.5–3 \times 10^5$ cells/ml and 100 µl was used per transfection. Protoplasts were transfected with 10 µg of pUC-35S-FLAG-RBS containing *FLAG-RBOHD*,

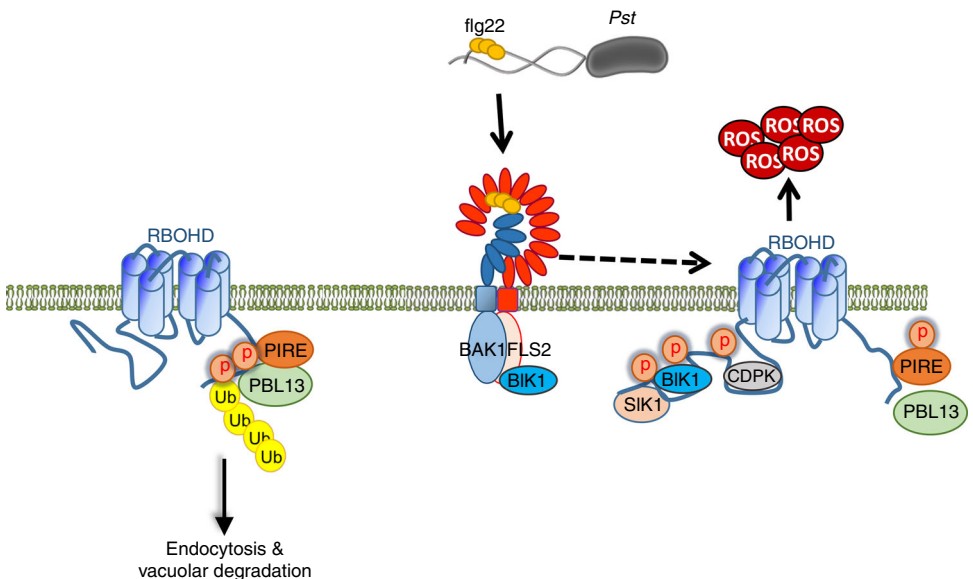

**Fig. 8 Model of PBL13 and PIRE regulation of RBOHD.** RBOHD's C-terminus is phosphorylated and ubiquitinated by PBL13 and PIRE, respectively, at a resting state, which leads to reduced RBOHD stability through vacuolar degradation. Upon flg22 perception, RBOHD is activated by N-terminal phosphorylation, resulting in an apoplastic ROS burst and ROS-mediated immune responses. PIRE is dynamically phosphorylated during immune activation.

*FLAG-RBOHD^{T912A}*, and *FLAG-RBOHD^{T912D}*. After 18 h, total protein was extracted and RBOHD expression was determined by immunoblot with anti-FLAG-HRP at a concentration of 1:5000 (Sigma, SAB4200119, clone 6F7).

**Co-immunoprecipitation.** For co-immunoprecipitation in *N. benthamiana*, pENTR-*LTI6b-3xFLAG* and pENTR-*RBOHD* were recombined into pEarlygate104 to generate YFP-tagged fusions. pENTR-*PBL13* and pENTR-*PIRE* were recombined into pGWB14 and pGWB15, respectively, to generate HA-tagged fusions. These constructs were electroporated into *A. tumefaciens* strain GV3101. *Agrobacterium* suspensions carrying *35S::YFP-LTI6b-3xFLAG*, *35S::YFP-RBOHD*, *35S:: PBL13-3xHA*, and *35S::3xHA-PIRE* were infiltrated in various combinations into *N. benthamiana* leaves (OD₆₀₀ = 0.3). One gram of leaf tissue was harvested at 24 hpi, and total protein was resuspended in 2 mL IP buffer (150 mM Tris–HCl pH 7.5, 150 mM NaCl, 10% glycerol, 10 mM DTT, 1 mM PMSF, 1x complete protease inhibitor (Roche), 1× phosphatase inhibitor (Pierce), 2% IGEPAL). The samples were clarified by centrifugation at 18,407 × *g* at 4 °C for 20 min and filtered using micro-bio-spin column (Biorad). Remaining supernatants were incubated with 20 µl of GFP-Trap (Chromotek) at 4 °C for 2 h. Beads were washed four times with 1 mL IP buffer and bound proteins were eluted in 70 µl 1.5× laemmli buffer.

In Arabidopsis, mesophyll protoplasts were isolated from the transgenic line *35S::PBL13-3xFLAG/pbl13-2* #8-3[25,55]. *LTI6b*, *RBOHD*, or *PIRE* in pENTR were recombined into pBluescript by LR reaction, resulting in *35S::LTI6b-GFP*, *35S:: GFP-RBOHD*, and *35S::GFP-PIRE*. Protoplasts were transfected with 100 µg of the indicated constructs. After 16–20 h of incubation, protoplasts were harvested and the total protein was extracted with 1 mL IP buffer (50 mM Tris–HCl pH 7.5, 150 mM NaCl, 10% glycerol, 5 mM DTT, 1× complete protease inhibitor (Roche),1× phosphatase inhibitors (Pierce), 2% IGEPAL). IPs were performed as described above.

To test the association between native RBOHD with PBL13 or PIRE in Arabidopsis transgenic lines, microsomal fractions were isolated from *35S::PBL13-3xFLAG/pbl13-2* #8-3 and *35S::YFP-PIRE*. Col-0 was used as a control. For isolation of microsomes, 3–4 g of 10-day old Arabidopsis seedlings grown on ½ MS media were ground and resuspended in homogenization buffer (50 mM HEPES, pH 7.5, 250 mM sucrose, 5% glycerol, 10 mM EDTA, 0.5% PVP, 50 mM NaPP, 1 mM NaMo, 25 mM NaF, 1× complete protease inhibitor [Roche])[56]. Cell debris were removed by centrifugation at 8000 × *g* at 4 °C for 10 min. The supernatants were then filtered by Miracloth, followed by ultracentrifugation at 100,000 × *g* at 4 °C for 30 min. The resulting microsomal isolates was solubilized in 700 µl homogenization buffer containing 2% IGEPAL in 3 mg/ml proteins samples at 4 °C for 30 min. IP was performed as described above but the beads were washed five times with the homogenization buffer containing 0.2% IGEPAL. For Co-IP after flg22 treatment in *Arabidopsis* seedlings, 10-day-old Col-0 or *35S::YFP-PIRE*/Col-0 were incubated with water for 18 h and then treated with 0.5 µM flg22 for 10 min. The microsomal isolation and co-IPs were conducted as described above using anti-GFP conjugated beads.

Immunoblotting was performed using anti-HA-HRP at a concentration of 1:1000 (Sigma, 12013819001, clone 3F10), anti-FLAG-HRP at a concentration of 1:5000 (Sigma, SAB4200119, clone 6F7), anti-GFP-HRP at a concentration of 1:3000 (Miltenyi Biotec, 130-091-833, clone GG4-2C2.12.10) or anti-RBOHD at a

concentration of 1:2000 (Agrisera, AS15 2962), followed by secondary anti-rabbit-HRP at a concentration of 1:3000 (BioRad, 170-5046). Experiments were repeated three times with similar results.

**Recombinant proteins purification and in vitro pull-down.** The N-terminus (aa1–376) and the C-terminus (aa756–912) of *RBOHD* cDNA was PCR amplified and directionally cloned into pENTR/D-TOPO (Invitrogen). The pMALc4x *E. coli* expression vector was modified by inserting the *ccdB* cassette (attR1-*ccdB*-CmR-attR2) as an *EcoRI-HindIII* fragment to generate a gateway compatible vector. *RBOHD-N* and –*C* terminal fragments were recombined into pMALc4x by LR reaction, resulting in *MBP-RBOHD-N* or *MBP–RBOHD-C*. *PBL13* and PBL13 lacking its C-terminus (*PBL13ΔC*; aa1–375) were PCR amplified and cloned into pET28a (Novagen), resulting in *HIS-PBL13* or *HIS-PBL13ΔC*.

Expression constructs were transformed into *E. coli* Rosetta (DE3), cultures were grown at 37 °C until OD₆₀₀ = 0.4–0.6, protein expression was induced by 0.3 mM IPTG and cultures were incubated at 28 °C for 4 h prior to harvest. HIS-tagged constructs were cultured in LB, while MBP-tagged constructs were cultured in Rich Media (10 g tryptone, 5 g yeast extract, 5 g NaCl, 2 g dextrose/l). To purify fusion proteins, cells were harvested by centrifugation at 5000 × *g* for 10 min and all steps were performed at 4 °C or on ice. For purification of MBP-tagged proteins, the harvested cells were resuspended and sonicated in MBP column buffer (20 mM Tris–HCl pH 7.4, 200 mM NaCl, 1 mM EDTA, and 0.1% Triton X-100). The soluble fraction was incubated with amylose resin (NEB) at 4 °C for 1 h and then the resin was washed with the MBP-buffer four times. Purified proteins were eluted using 10 mM maltose. To purify HIS fusion proteins, the harvested cells were resuspended and sonicated in His buffer (20 mM Tris–HCl pH 8.0, 300 mM NaCl, 10 mM imidazole, and 0.1% Triton X-100). The soluble fraction was incubated with Ni-NTA-affinity resin (Qiagen) and purified proteins were eluted by 250 mM imidazole.

For in vitro pull-down assays, *MBP, MBP-RBOHD-N*, or *MBP-RBOHD-C* were co-expressed with *HIS-PBL13* in *E. coli*. The soluble fraction from a 10 ml cell culture were bound to 20 µl amylose resin (NEB) at 4 °C for 1 h. The resin was washed four times with MBP-column buffer and bound proteins were eluted with 10 mM maltose. Eluted proteins were detected by immunoblotting with MBP antibody (New England Biolabs, E8030S) at a concentration 1:3000 and rabbit-HRP antibody (Biorad, 170-5046) at a concentration 1:5000. PBL13 was detected by His antibody (Invitrogen, MA1-21315, clone AB_557403) at a concentration 1:2000 and anti-mouse-HRP (BioRad, STAR77) at a concentration 1:5000. This experiment was repeated three times with similar results.

**In vitro kinase assay.** In vitro *kinase assays* were performed as described in previously[25]. 0.5 µg of purified HIS-PBL13, HIS-BIK1, and HIS-PBL13ΔC and 2 µg of HIS-RBOHD-C were incubated in kinase reaction buffer (20 mM Tris–HCl pH 7.5, 10 mM MgCl₂, 1 mM CaCl₂, 100 µM ATP, 1 mM DTT). For analyzing PBL13 phosphorylation of PIRE, we used GST-PIRE, MPB-PBL13, and MBP-RBOHD-C. Reactions were initiated by incubating at 28 °C for 1 h and stopped by adding 3× Laemmli sample buffer. The reactions were subjected to immunoblot with anti-pThreonine antibody (Cell Signaling, 9381) at a 1:3000 concentration, followed by

anti-rabbit-HRP at a concentration of 1:5000 (BioRad, 170-5046). This experiment was repeated twice with similar results.

**Identification of RBOHD phosphorylation.** To identify PBL13-mediated phosphorylation of RBOHD, *MBP-RBOHD-C* was co-expressed with *HIS-PBL13, HIS-PBL13* kinase dead mutant (*KD*), or the empty vector in *E. coli*, and purified as described above. The purified proteins were separated on a 10% precast gel (Biorad) and stained with Coomassie brilliant blue. MBP-RBOHD-C sample was excised from the gel, reduced with 5 mM DTT, alkylated with 50 mM iodoacetamide, and digested overnight with trypsin (Promega) at 37 °C[57]. Tryptic peptides were analyzed by LC–MS/MS using an Orbitrap Q Exactive mass spectrometer (Thermo Fisher Scientific)[58]. Phosphorylation of S, T, or Y residues were allowed as variable modifications with setting for 20 ppm peptide tolerance. Peptide identification and phosphosite assignment were performed using X!Tandem[59]. X!Tandem results were combined and phosphorylated residues were manually verified in Scaffold version 4[60]. The experiment was repeated two times with similar results.

To test RBOHD's phosphorylation by PBL13 *in planta*, the RBOHD phospho-antibodies (anti-pRBOHD) were generated in a rabbit against a phosphorylated S862 RBOHD peptide (CSGTRVKpSHFAKP). Crude antisera was affinity purified and tested for specificity by ELISA (Proteintech). *3XHA-RBOHD* or *3XHA-RBOHD S862A* were co-expressed with either *PBL13-3xFLAG* or *PBL13-3xFLAG* kinase dead mutant (KD) in *N. benthamiana* using *Agrobacterium*-mediated transient expression system. The 3xHA-RBOHD variants were then immunoprecipitated using anti-HA-conjugated agarose beads from microsomal fractions. The microsomal isolation and IP were performed as described above. The eluted samples after IP were subjected to immunoblotting and probed with primary anti-pRBOHD at a concentration of 1:1000, followed by secondary anti-rabbit at a concentration 1:3000 (Biorad, 170-5046).

**Chemical treatment.** Seven-day-old transgenic Arabidopsis *npro::3xFLAG-RBOHD/ rbohd* was grown on solid ½ MS media and then transferred to liquid ½ MS media. Five seedlings were treated with DMSO, 0.5 or 1 µM ConA for 18 h under constitutive light. Total protein was isolated in 2× laemmli buffer. Protein concentrations were measured by the Pierce 660 nm Protein Assay Kit (Pierce) and 5 µg of protein was loaded per lane and separated by SDS–PAGE. The expression of 3xFLAG-RBOHD were detected by immunoblotting with anti-RBOHD (Agrisera, AS15 2962) at a concentration of 1:2000 or anti-FLAG-HRP (Sigma, SAB4200119, clone 6F7) at a concentration of 1:5000, respectively. Statistical differences were detected after image quantification in Image Lab (Bio-Rad) by one-way ANOVA with post-hoc Tukey HSD following a significant *F*-statistic, $n = 3$, $\alpha = 0.05$. The experiment repeated three times with similar results. Similarly, the level of native RBOHD expression was tested in Col-0, *pbl13-2*, and *pire-1* KO lines after ConA treatment. The chemical treatment and sample preparation follow the procedures described above. Native RBOHD was detected by immunoblotting with anti-RBOHD at a concentration of 1:2000 (Agrisera, AS15 2962), followed by secondary anti-rabbit at a concentration of 1:3000 (Biorad, 170-5046). The experiment was repeated three times with similar results.

For dephosphorylation assays *in planta*, *3xHA-RBOHD* and *PBL13-3xFLAG* were co-expressed in *N. benthamiana* using *Agrobacterium*-mediated transient expression system. 3xHA-RBOHD was immunoprecipitated with anti-HA-conjugated agarose beads as described above. After incubation and washing beads, the beads were divided into two and then treated with either 1 µl of lambda phosphatase (λPPase) or water in the presence of 1 mM MnCl₂ in 30 µl reaction volume at 28 °C for 10 min. The reaction was stopped by adding 3xlammli buffer and incubating at 65 °C for 10 min. The level of RBOHD's phosphorylation after λPPase treatment was detected by immunoblotting using anti-pRBOHD S862 at a concentration of 1:1000, followed by secondary anti-rabbit at a concentration 1:3000 (Biorad, 170-5046). To test PIRE dephosphorylation after phosphatase treatment, *YFP-PIRE* was expressed in *N. benthamiana* using Agrobacterium-mediated transient expression (OD₆₀₀ = 0.2). At 22 hpi, 200 nM flg22 was infiltrated into the Agrobacterium-infiltrated regions and incubated for 15 min. Subsequently, the samples were ground in liquid N₂ and total proteins were extracted using cell lysis buffer (50 mM HEPEs, pH 7.5, 100 mM NaCl, 1 mM PMSF, 1X protease inhibitor cocktails [Roche], 1% IGEPAL). The soluble proteins were separated by centrifugation at 20,000 × *g* at 4 °C for 20 min. The samples containing 150 µg of proteins were treated with 1 µl λPPase with 1 mM MnCl₂ at 28 °C for 10 min. The dephosphorylation of YFP-PIRE was detected by immunoblotting using anti-GFP at a concentration of 1:3000 (Miltenyi Biotec, 130-091-833, clone GG4-2C2.12.10).

**In vivo ubiquitination assay.** To detect in vivo ubiquitination of RBOHD, 1 g of 8-day-old transgenic Arabidopsis *npro::3xFLAG-RBOHD/rbohd*[2] or Col-0 grown on ½ MS media was harvested. The microsomal fraction was enriched using the Plasma Membrane Protein Isolation Kit for Plants (Inventbiotech). The total microsomal fraction was solubilized in RIPA buffer (50 mM Tris–Cl, pH 7.5, 150 mM NaCl, 0.5% sodium deoxycholate, 0.1% SDS, 1% NP-40) and clarified by centrifugation at 18,407 × *g* at 4 °C for 20 min. Supernatants were adjusted to a concentration of 0.6–0.8 mg/mL and incubated for 2 h at 4 °C with 20 µl of anti-FLAG M2 gel (Sigma) for IP of RBOHD. Following incubation, the resin was

washed four times with 1 mL RIPA buffer and purified RBOHD was eluted by adding 70 µl 1.5x laemmli buffer. RBOHD was detected by immunoblotting with anti-FLAG-HRP at a concentration of 1:5000. The ubiquitinated proteins were detected by immunoblotting with anti-ubiquitin (Clone P4D1-A11, Millipore Sigma) at a concentration of 1:2000, followed by secondary Trueblot anti-mouse at a concentration of 1:1000 (Rockland Inc.). Experiments were repeated three times with similar results.

**Tandem Ubiquitin Binding Entities assay.** The TUBE assay was performed as described previously with modifications[61]. *3xHA-RBOHD* wildtype (WT) and T912 mutants (RBOHD^T912A or RBOHD^T912D) were transiently expressed in *N. benthamiana* for 24 h (OD₆₀₀ = 0.2), 1 g of leaf tissues were ground and resuspended in lysis buffer (50 mM Tris–HCl, pH 7.5, 150 mM NaCl, 5 mM DTT, 1 mM EDTA, 10% glycerol, 1 mM PMSF, and 1× protease inhibitor [Roche], 2% IGEPAL, 50 µM PR-619 [LifeSensors], 5 mM 1-10-phenanthroline). The resuspensions were incubated at 4 °C for 30 min with rotation, followed by centrifugation at 20,000 × *g* at 4 °C for 20 min. The resulting supernatants were filtered by Miracloth and precleared at 4 °C for 2 h with control agarose beads (LifeSensors) in order to avoid unspecific binding caused by the agarose matrices. The precleared samples were then divided into two fresh tubes and incubated with either the TUBE-conjugated agarose (LifeSensors) or the control agarose beads at 4 °C for 18 h under rotation. After incubation, the beads were washed five times with TBST buffer (20 mM Tris, pH 8.0, 150 mM NaCl, 0.1% Tween-20) and total bound proteins were eluted by 1.5× laemmli buffer after incubating at 65 °C for 10 min. The specificity of TUBE assay to pull down ubiquitinated proteins were validated by immunoblotting using anti-ubiquitin (Millipore Sigma, 05-944, clone P4D1-A11) at a concentration 1:2000, followed by secondary anti-mouse at a concentration 1:5000 (Biorad, 170-5046). The level of RBOHD's ubiquitination was determined by immunoblotting using HA-HRP antibodies at a concentration of 1:1000 (Sigma, 2013819001, clone 3F10). Band intensity was quantified in Image Lab (BioRad) and statistical differences were detected by a two-tailed Student's *t*-test, $n = 3$, $\alpha = 0.05$. The experiment was repeated three times with similar results.

**Bimolecular fluorescence complementation.** For BiFC, PIRE was fused to the citrine C-terminus (YC) and PBL13 was fused to the N-terminal 155 amino acid residues of citrine (YN) under the control of a 35S promoter and a NOS terminator[62]. Plasmids were transformed into *Agrobacterium* strain GV3101 and infiltrated at an OD₆₀₀ = 0.5 in a 1:1 ratio. Twenty-four h after infiltration 10 µM of MG132 was infiltrated in the same leaf and fluorescence imaging was performed 24 h after MG132 infiltration using Zeiss 710 Laser scanning confocal system. A 25 mW Argon laser (Coherent) with appropriate emission filters were used to image chloroplast (458 nm laser) and citrine (514 nm laser).

**Bacterial strains and disease assays.** *Pseudomonas syringae* pv. *tomato* DC3000 (*Pto* DC3000) expressing the pVSP61 empty vector[63] was grown on nutrient yeast-glycerol (NYG) media for 2 days at 28 °C in the presence of 25 µg/ml kanamycin and 100 µg/ml rifampicin. Four-week-old Arabidopsis leaves were syringe infiltrated at a concentration of $1 \times 10^5$ CFU/ml bacteria. Bacterial titers were analyzed 4 days post inoculation after grinding leaf tissue in 10 mM MgCl₂, dilution plating onto NYG media containing 25 µg/ml kanamycin and 100 µg/ml rifampicin, and incubation at 28 °C for 2 days[64]. There were six biological replicates (individual plants) per treatment and the experiment was repeated three times with similar results. Statistical differences were detected by two-way ANOVA (genotype × time) with post-hoc Tukey HSD following a significant *F*-statistic, $n = 18$, $\alpha = 0.05$.

**ROS burst.** In Arabidopsis, ROS production after treatment with 100 nM flg22 (GeneScript, 95% purity) was measured using a TriStar Luminometer in a 96-well plate[65]. All experiments were repeated at least four times with minimum 16 biological replicates per replication. Statistical differences were detected by two-way ANOVA (genotype × time) with post-hoc Tukey HSD following a significant *F*-statistic, $n > 40$, $\alpha = 0.05$. Outliers were detected by the ROUT method and removed from the analyses[66].

Trans-complementation of *NbRBOHB* in *N. benthamiana* was used to analyze the activity of specific *RBOHD* mutants. VIGS of *NbRBOHB* or *GUS* was performed by infiltrating *N. benthamiana* at the two leaf stage with *Agrobacterium* strain GV3010 containing pBINTRA6 (RNA1) or pTV00 containing *NbRBOHB* or *GUS* (RNA2)[18,21]. *RBOHD* and its phosphorylation mutants in pGWB15 were electroporated into *Agrobacterium* strain GV3101 and used to trans-complement *NbRBOHB*-silenced plants[18,21]. *Agrobacterium* carrying *35S::GFP* in pCB01 was used as a control. Leaf discs were excised at 24 hpi and ROS production after flg22 treatment was measured as described above. The expression of RBOHD variants after trans-complementation was detected by immunoblotting using anti-HA–HRP antibody at a concentration of 1:1000. Experiments included a minimum of six biological replicates (individual plants) and were repeated three times with similar results. Statistical differences in ROS production were detected by ANOVA with post-hoc Tukey HSD following a significant *F*-statistic, $n > 18$, $\alpha = 0.05$. Outliers were detected by the ROUT method and removed from the analyses[66].

**RT-PCR and qPCR**. Arabidopsis seedlings of Col-0, *35S::3xHA-RBOHD*/Col-0 #3, *35S::3xHA-RBOHD*/*pbl13-2* #1, and *35S::3xHA-RBOHD*/PBL13-3xFLAG #8-3 #3 were grown on ½ MS media for 8 days. RNA was extracted using TRIzol reagent (Invitrogen). RT-PCR was conducted using MMLV reverse transcriptase (Promega). AmpliTaq Polymerase (Thermo Fisher) was used for PCR reactions. The expression of *3xHA-RBOHD* was detected by a HA-specific forward primer and a RBOHD reverse primer, and the pair of primers for *ELONGATION FACTOR 1-α* (*EF1-α*) was used as an endogenous control. The PCR products were separated on a 1.5% agarose gel and were visualized after ethidium bromide staining.

For qPCR analyses, Col-0, *pire-1*, *pbl13-2*, and *pbl13-2 pire1* seedlings were grown for 10 days on MS media. TRIzol reagent (Invitrogen) was used for RNA extraction and cDNA was synthesized utilizing MMLV reverse transcriptase (Promega) with 2 μg of RNA. For qPCR on protoplasts after transfection, protoplasts were harvested 18 h after transfection frozen and processed using the RNeasy plant mini kit following manufacturer specifications (QIAGEN). cDNA was synthesized as described above. Primers were designed to amplify the C-terminal region of the FLAG tag and the N-terminal region of *RBOHD* to avoid signal from endogenous *RBOHD*. *RBOHD* expression was quantified on the CFX96 real-time PCR detection system (Bio-Rad). SsoFast EvaGreen Supermix was utilized for qPCR reaction following the manufacturer's protocol. Thermocycling steps were as follows: 95 °C for 30 s followed by 40 cycles alternating between 5 s at 95 °C and 15 s at 60 °C. The melt curve was calculated after the final cycle by alternating between 65 °C for 5 s and 95 °C for 5 s. Gene expression was calculated using the ΔΔcT method and normalized against Arabidopsis *ELONGATION FACTOR 1α* (AT2G18720). Primers used in all experiments are included in Supplementary Table 2.

**Yeast two-hybrid assay**. *RBOHD-C* (756-912) and *PIRE* C244S/C247S were cloned into pTBS1 with LexA DNA-binding domain under the control of galactose inducible promoter (bait vector); *PBS1, RBOHD-N, RBOHD-C, PBL13* were cloned into pJG4–5 containing activation domain (prey vector) using Gibson assembly. The resulting plasmids were sequenced. pTBS1-*RBOHD-C* or pTBS1-*PIRE C244S/C247S* were transformed into EGY48 yeast strain and plated on complete supplementary media plates lacking uracil and histidine (−UH). Positive colonies were streaked and colony PCR was performed to confirm transformation. EGY48 with pTBS1-*RBOHD-C* or pTBS1-*PIRE* C244S/C247S were grown overnight in liquid SD(−UH) media and used to transform the prey vector with *PBS1, RBOHD-N, RBOHD-C, PBL13*, or vector alone and selected on plates lacking uracil, tryptophan, and histidine (−UTH). Positive clones were confirmed by colony PCR against both bait and prey plasmids. Serial dilutions of positive clones were spotted on plates lacking uracil, tryptophan, and histidine, and leucine (−UTHL) with galactose/raffinose or glucose as sugar source.

**In vitro interaction between PBL13, RBOHD, PIRE, and BIK1**. The coding region of *PIRE, PBL13, RBOHD-C*, and *BIK1* was amplified from a cDNA library. *PIRE* was cloned into the GST vector (Addgene #29707) linearized with *Ssp1*. *PBL13* and *RBOHD-C* were cloned into the MBP vector (Addgene #29708) linearized with *Ssp1*. Cloning was performed using Gibson assembly. Sequenced plasmids were transformed into the BL21 expression strain of *E. coli*. 10 ml of overnight grown cultures were inoculated in 1 l of 2XYT media (RPI corp.). At an OD$_{600}$ of 0.6, recombinant protein expression was induced by adding 0.5 mM isopropyl β-D-1-thiogalactopyranoside (IPTG) and incubated for 16 h at 23 °C. Cells were harvested by centrifugation at 5283 × g for 10 min. Cells were resuspended in lysis buffer (50 mM Tris 300 mM NaCl pH 7.5) and lysed using microfluidizer. The lysate was clarified by centrifugation at 37,565 × g for 30 min. GST-PIRE was incubated with GST agarose beads and MBP-tagged proteins were incubated with amylose beads for 1 h at 4 °C. GST and amylose beads were washed with 20 column volumes of lysis buffer. GST protein was eluted with lysis buffer + 20 mM glutathione. MBP protein was eluted with lysis buffer + 20 mM maltose. Proteins were further purified using HiTrap SP ion exchange chromatography and HiLoad 16/600 Superdex 200 pg size exclusion chromatography.

In vitro pull-downs were performed by incubating 1 μg of GST PIRE with 3 μg of MBP-PBL13, MBP-BIK1, or MBP-RBOHD-C with amylose beads calibrated with 30 mM HEPES, 180 mM NaCl, and 0.2% triton X100. Proteins were incubated with moderate shaking at 4 °C for 2 h. Amylose beads were washed with 100 column volumes of wash buffer 30 mM HEPES, 220 mM NaCl, and 0.2% triton X100 for 3 h. Protein bound to amylose resin was eluted by boiling the resin in SDS sample buffer.

**In vitro ubiquitination assays**. *HIS-GFP-PBL13* or *RBOHD-C* were cloned into the Addgene vector 29716 linearized with *Ssp1*. Cloning and protein purification was performed as mentioned above with the exception of using cobalt resin instead of amylose or GST, and proteins were eluted with lysis buffer + 500 mM imidazole.

In vitro ubiquitination assays were performed by incubating substrate protein in ubiquitination reaction buffer. In brief, 100 ng GST-E1 from yeast, 4 μg human FLAG-ubiquitin, 200 ng of UbcH5a, 500 ng of GST PIRE, and 1 μg of substrate was incubated in 30 μl buffer containing 50 mM Tris–HCl (pH 7.6), 10 mM MgCl$_2$, 4 mM ATP, and 1 mM DTT. The reaction was incubated at 30 °C for 3 h, and analyzed by immunoblot analysis.

**MAPK assay**. Seven-day-old Arabidopsis seedlings of Col-0 or *pire-1* grown on ½ MS media were placed in water for 20 h, followed by incubation with 1 μm flg22. At 0-, 5-, and 15-min after flg22 treatment, the samples were ground in liquid N$_2$ and resuspended in 50 mm HEPEs (pH 7.5), 50 mm NaCl, 10 mm EDTA, 0.2% Triton X-100, 1× Protease Inhibitor Cocktail (Sigma), and 1× Halt Phosphatase Inhibitor Cocktail (Thermo). Total proteins were isolated by centrifugation at 20,000 × g at 4 °C for 20 min and Laemmli buffer was added for subsequent SDS–PAGE. Activated MAPK3 and MAPK6 were detected by immunoblotting using p44/42 MAPK antibodies at a concentration of 1:3000 (Cell Signaling Technology, 9102S), followed by secondary rabbit antibodies at a concentration of 1:5000 (Biorad, 170-5046).

**RBOH C-terminal conservation**. The last 15 C-terminal residues of different RBOHs were aligned utilizing ClustalW. The sequence logo consists of 29 RBOHs from the following plants: *Arabidopsis thaliana* (10), *Oryza sativa* (7), *Solanum lycopersicum* (2), *Solanum tuberosum* (4), *Nicotiana benthamiana* (2), *Nicotiana tabacum* (4). The sequence logo was generated utilizing Weblogo.

**Reporting summary**. Further information on research design is available in the Nature Research Reporting Summary linked to this article.

## Data availability
All data supporting the findings of this study are available in the manuscript and its supplementary files or are available from the corresponding author upon request. The data underlying Figs. 1a–e, 2a–d, 3b–f, 4a, c–g, 5a–c, 6a–e, 7a–d, and Supplementary Figs. 1a–c, 2b–e, 3a–c, 4, 5a–c, 6b, 7a–d, 8a, b, 9a, 10a–d, 11, 12c–f, 12h–i are provided as a Source Data file.

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

## Acknowledgements

We thank Judy Callis for providing the *pire-1* seed. This work was supported by grants from the NIH (R01-GM092772) and UC Davis RISE awarded to G.C.; N.K.L. was supported by UC Davis RISE and NSF-IOS-1354434 funds awarded to S.P.D.-K. S.M. was supported by NSF-DBI-0723722 and NSF-DBI-1042344 awarded to S.P.D.-K.

## Author contributions

D.H.L., N.K.L., S.P.D.-K., and G.C. designed experiments. D.H.L. performed plant-based assays, in vitro biochemistry, phosphorylation assays, and mass spectrometry. N.K.L. performed ubiquitination assays and in vitro biochemistry. B.C. performed the conservation analyses, qPCR, and some plant-based assays. T.T. performed qPCR and protoplast assays. Z.-J.D.L. and J.L. performed initial phenotyping of *pire-1* and purified protein for microarray experiments. S.M. performed protein microarray experiments. N.K.L., D.H.L., S.P.D.-K., and G.C. wrote the manuscript.

## Competing interests

The authors declare no competing interests.

**Additional information**

