## [Peer Review File · Nature Communications]

Reviewers' comments:

Reviewer #1 (Remarks to the Author):

The paper by Lee et al describes the importance of PBL13 and the newly identified PIRE protein to control RBOHD abundance via ubiquitination. The work is quite novel but suffers from the lack of a few critical experiments, as well as issues related to the quality of other data.

In my opinion, the two critical experiments missing are

1. The authors do not show data demonstrating *in vivo* that PBL13 can directly phosphorylate RBOHD (or PIRE). This seems essential to their model. Moreover, it is odd that in some of their experiments that S-A mutants show a phenotype while the phosphomimic (S-D) do not. The authors really need to confirm that what they see *in vitro* occurs *in vivo*.
2. The story starts with PBL13 and then focuses on the interaction of this protein with PIRE. The assumption is that PBL13 phosphorylates PIRE but this is never shown in the paper. The authors need to address the mechanism of PBL13-PIRE interaction and its physiological consequence. If PIRE is phosphorylated by PBL13, how does it affect the stability of PIRE protein or E3 ligase activity by phosphorylation?

More minor comments:

p. 4 & 10 RBOHD was also recently shown to be phosphorylated by the receptor for extracellular ATP (Chen et al. *Nature Commun*) and this should be mentioned in the intro.

I did not go back and review the earlier papers but was wondering how the data using BIK1 in figure 1C corresponds to earlier work showing that BIK1 can phosphorylate RBOHD? Should be mentioned somewhere in the paper.

The authors should consult the Nuhse et al. 2007 *Plant J* paper to see whether this earlier global phosphoproteomic analysis detected the phosphorylated residues of RBOHD that they identified. This would lend support that these are indeed phosphorylated in planta.

There are a number of questions/concerns about the quality of the data presented. I have listed the majority of these concerns below.

1. In IP experiments (Fig 1a, 3c, Supplementary Fig. 1 and 5), you can see that the GFP or YFP tagged RBOHD is not detected in 'Input' but RBOHD is seen in IP. Of course, it seems to be enriched by IP, but if no band is detected in the input, it is doubtful whether it is the band that is actually caused by the interaction. It seems necessary to verify this. Likewise, in Supplementary Figure 1a, the YFP-RBOHD in the bottom of input panel, I could see very weak signal in both sides, and it is hard to see the protein in the figure. Why are the signals so low? Did the authors run the protein and use correct antibody for this? Please check this.
2. In Figure 1b: In the bottom of INPUT panel, the 3rd lane has a strong band at the size of MBP-RBOHD-N. I think this is for MBP-RBOHD-C. Why do they have both MBP-RBOHD-C and N bands? Did the authors use correct proteins? Please check whether this is correct or not.
3. In Fig. 2b and Fig. 3b and Fig. 4e, all experiments have been done only once? Or, if you have multiple experiments, describe both the mean and the error.
4. *In vitro* MBP pull-down experiments show that PBL13 strongly interacts with the C-term of the RBOHD protein but RBOHD N-term also interacts with PBL13 protein. Thus, I think the N-term of RBOHD is also phosphorylated by PBL13. Therefore, it is necessary to experimentally verify whether PBL13 also phosphorylates the N-term as well as C-term of RBOHD and how it affects ROS production.
5. The residues of S862, S907, T910, T911 and T912 of RBOHD are shown to be highly conserved in *Arabidopsis* RBOHB and RBOHC. Verify whether PBL13 causes protein interactions and phosphorylation with these RBOHB or RBOHC. It more needs to be discussed in the 'Discussion'.
6. According to previous reports, PBL13 has a palmitoylation site at the N-term and is located at

the membrane. I think PIRE should localize on the plasma membrane to ubiquitinate RBOHD. So, Author need to show the subcellular localization of PIRE. In addition, it is necessary to investigate whether the PIRE protein actually interacts with ROBHD on the Plasma membrane. ex) BiFC.

7. According to previous reports, PBL13- RBOHD interaction was disappeared under flg22 treatment. Verify that the interaction between PIRE and RBOHD diminishes or disappears when treated with flg22 or other elicitor.

8. In supplementary Figure 1b

The position of plates is different from the guide figure (left-side of 1b). Please place plates precisely. It seems that EV (empty vector control) grow in -UTHL media. Why is it growing in - UTLH media? Is it correct? Why don't the authors check the interaction with serial dilution method?

Very minor points:

-Page 4, "RBOHB and RBOHC mediated ROS generation play an important role in root hair formation and seed ripening, respectively." \ "RBOHB and RBOHC mediated ROS generation play an important role in seed ripening and root hair formation, respectively."

-Page 4, NADPH oxidase\ NADPH oxidase

-Page 5, In vitro \ In vitro

-Page 7, Ubiquitinated\ Ubiquitinated

-Page 10 ROBHD\RBOHD, fl22\flg22

-Page 17 OD600\OD600 , rbohD\ rbohD

- In Fig. 1b, IP\ pull-down

- In Fig. 3e, two bands of PBL13-MBP appear when blotted with anti MBP antibody? If upper band is correct size, what is lower band? Is it a truncated protein? And Detection by anti GST antibody showed another band between 1 and 2 lane. Why does the band appear here?

-In Fig. 3d, is the series dilution in the loading control panel (bottom) correct? It looks all same. If so, indicate how much dilution has been done. Ex) 10-1 10-2 10-3

-In Fig. 4a-b, strong bands appear at about 150kd. In particular, these bands were showed in the western blotting both using anti GST and Flag antibodies. Please clarify what is these bands.

- In Supplementary Fig. 1b, -UTH picture needs to match up with diagram.

- In Supplementary Fig. 2a, why is not S907 listed? And it seems that the mark of pT in T911 is wrong

- In Supplementary Fig. 2b, PBL13 phosphorylation sites\ Phosphorylation sites of RBOHD by PBL13

- In Supplementary Fig. 4. In Method section, Authors describe "The expression of RBOHD was detected by the gene specific primers for RBOHD" in RT-PCR section (Page 18). However, in the Primer list, Reverse anneals to RBOHD but Forward primer binds to pGWB15 vector sequence. So perhaps in the Supplementary Fig. 4, RBOHD transcripts were not detected in Col-0. Or RBOHD was not detected in Col-0 because other plants were overexpressed by 35S promoter? Make this part clear.

In addition, 35S promoter, which is a constitutively expression, is used instead of the native promoter (pGWB15 vector). Therefore, it is thought to be the result of investigating the stability of RBOHD mRNA by pBL13 protein after trans-transcription rather than transcription.

- Supplementary Fig. 6b E2 and FLAG-UB need to describe, and also need to align the lines.

- In Supplementary Fig. 7, EF1-a transcripts were detected in -RT reaction. I think it means that contamination of genomic DNA. So, this does not need to be shown here. Instead of -RT, it is recommended to show the results of genotyping rather than this. In addition, Line numbers are all different.

In the 'Methods' section, Col-0, 35S::3xHA-RBOHD/Col-0, 35S::3xHA-RBOHD/pbl13-2, 35S :: 3xHA-RBOHD / PBL13-3xFLAG # 8-3

But Figure legends, 3xHARBOHD /Col-0 # 3, 3xHA-RBOHD / pbl13-2 # 1, or 3xHA-RBOHD / PBL13-3XFLAG # 3.

in the picture, The Line number is 1 and 2. So, specify the correct line numbers or names.

Reviewer #2 (Remarks to the Author):

This manuscript by Lee et al. identifies two novel components of the regulatory system surrounding PAMP recognition and particularly regulating the NADPH oxidase RBOHD, PBL13 and PIRE. These novel regulators seem to regulate RBOHD stability and add to the increasingly intricate network of regulation surrounding plant immunity. The information the authors present is highly valuable to the plant immunity field, but some concerns need to be addressed to support the hypothesis the authors proposed.

- 1) In vitro phosphorylation assays using purified proteins can be due to artificial effects in the test tubes (Fig 1C). The authors should compare the RBOHD protein in WT vs pbl13 mutants to see whether the same faint phosphorylated RBOHD band shift in Fig 1C can be observed. Can authors observe the same phosphorylation sites (Sup Fig 2) using tagged RBOHD-HA purified from WT Arabidopsis, under un-induced or inducing conditions? The reason this is raised is because curiously, no RBOHD band shift is observed in Fig 2A, perhaps the plants need to be induced with elicitor or pathogens to observe the shift? Or the gel should be run longer or under better conditions to observe the shift? If no shift is observed in vivo, it will be difficult to support its relevance in planta.
- 2) Similarly, in vitro ubiquitination assays with purified proteins with E3s can also be artificial (Fig 4). The authors should overexpress PIRE in Arabidopsis RBOHD-HA transgenic line to test whether RBOHD level is affected, and whether polyubiquitination bands of RBOHD can be observed with and without MG132. What is the PIRE overexpression phenotype?
- 3) Genetic evidence to support the authors' hypothesis is mostly lacking. If the authors' claim is correct, you would predict that the enhanced resistance phenotypes of pbl13 and pire can be fully suppressed by rbohhd knockout. Detailed phenotypic analysis, including ROS, PTI, and resistance assays of rbohhd pbl13 and rbohhd pire double mutants are crucial for the current study.
- 4) The authors seems to suggest a model where PBL13-phosphorylated RBOHD is a preferred substrate for PIRE. If that is the case, you would predict that overexpression of PIRE in pbl13 mutant background would have little or no effect on its ROS and enhanced resistance phenotypes. Is that true? At the same time, overexpression of both PBL13 and PIRE simultaneously should yield a rbohhd knockout mutant phenotype. The authors should seriously test that.
- 5) PIRE has a highly close paralog AT5G62910 in Arabidopsis. They even share similar splicing patterns! What is the evolutionary pattern of PIRE and AT5G62910? Is it similar to RBOHD and PBL13? With this paralog in mind, I am surprised that the authors can still observe a clear phenotype in the single mutant. Is AT5G62910 relevant? The authors should overexpress AT5G62910 to see whether it can reduce RBOHD levels and interact with PBL13 and RBOHD. What is the double mutant phenotype of pire with AT5G62910 knockout?
- 6) The interaction between RBOHD and PBL13 was not confirmed using stable transgenic lines. This was an issue for many of the experiments, use of stable transgenics is usually preferable to either expression in Tobacco or protoplast.
- 7) In figure 1 E, the levels of both the T912D overexpressers appear to be significantly lower than the other lines, this makes it difficult to draw a conclusion for this mutation. Of less significance but also of note is that the nonspecific band present in Col is missing from the other treatments.
- 8) The claim that Concanamycin A treatment increases the accumulation of ubiquitinated RBOHD is misleading as the increase in ubiquitinated protein correlates with the overall increase in protein.
- 9) Why do the authors choose to examine interactors of PBL13 rather than RBOHD, when they are attempting to identify regulators of RBOHD? Is there a reason this screening technique wasn't used with RBOHD?
- 10) Figure 3 E doesn't have a set of input samples for the IP.
- 11) Figure 4 E doesn't have expression controls for the transient expression. Again, this would be more reliable using stable transgenics.

12) The ROS data shown appears highly variable with extremely large ranges. It is unclear if there is a solution to this but given that this data is central to the paper it is perhaps worth considering additional methodologies.

Reviewer #3 (Remarks to the Author):

In the manuscript entitled "Dual regulation of reactive oxygen species through direct ubiquitination and phosphorylation of the plant immune regulator RBOHD", Lee and Lal et al., showed a novel negative regulatory mechanism of RBOHD by PBL13-mediated phosphorylation and PIRE-mediated ubiquitination. The authors found PBL13 phosphorylates the C-terminus of RBOHD and the phosphorylation is required for the RBOHD activity and stability. Moreover, the authors identified a novel E3 ubiquitin ligase PIRE which ubiquitinates RBOHD *in vitro*. Overall, the finding is novel and very interesting. However, the authors should provide more evidence to convince readers of their model in Nature communication.

Major points

- The accumulating evidence show that signaling pathway from FLS2 to RBOHD is different from that to MAPKs activation. For example, BIK1 is required for RBOHD activation but not for MAPKs activation (Feng, F. et al. Nature 485, 114–118). Since *pbl13* mutant shows enhanced flg22-inducible ROS burst and MAPKs activation, PBL13 should also regulate another component required for MAPKs activation, if RBOHD is a true substrate of PBL13. The authors should carefully discuss this point in the discussion. The authors should check endogenous FLS2 and BAK1 protein levels (the antibodies are commercially available) in *pbl13* and *pire* mutants to show the specific accumulation of RBOHD in those mutants. Although the authors show that *pire-1* mutant does not induce the accumulation of BAK1-3xFLAG by transient expression system in protoplast in Fig.4e, it is more proper and accurate way to check the endogenous BAK1 protein by the specific antibody. The authors should also check flg22-inducible MAPKs activation in *pire-1* and *pire-2* mutant to check if PIRE specifically regulates RBOHD or not.
- The authors showed the interaction among PBL13, RBOHD and PIRE by *in vitro* binding assay, Y2H assay and *in vivo* co-IP assay after overexpression in *N. benthamiana* or Arabidopsis protoplast. These interactions may be very transient *in vivo* and difficult to see using endogenous proteins or using expressed proteins by own promoters in stable Arabidopsis transformants. If so, it is important to show more genetic link among these three proteins. For example, the authors can check the RBOHD protein level in a *pbl13* *pire* double mutant together with their single mutants to check if PBL13 and PIRE function in the same pathway to regulate RBOHD stability.
- Although the authors found the *in vitro* phosphorylation sites of RBOHD by PBL13, there is no evidence about *in vivo* phosphorylation at these sites and the requirement of PBL13 for their phosphorylation *in vivo*. The authors can try to find S912 phosphorylation after treatment with ConA or in *pire* mutant background.
- Is the phosphorylation of RBOHD at T912 required for the ubiquitination by PIRE?
The authors can check the role of T912 phosphorylation on RBOHD ubiquitination by comparing the ubiquitination of WT RBOHD and T912D variant *in vitro* and possibly *in vivo* after treatment with ConA. The authors might also see the reduced ubiquitination of RBOHD variant carrying T912A mutation *in vivo*. It is also worth trying to compare ubiquitinated RBOHD level in Col-0 and *pbl13-2* background after treatment with ConA.
- S862D does not cause the reduction of RBOHD protein but abolish the ROS production. This result suggests that PBL13 does not only induce RBOHD degradation through ubiquitination by PIRE but directly inhibit the RBOHD activity by the phosphorylation at S862. The authors should

carefully explain this possibility in the discussion.

- Do authors check the possibility that PBL13 phosphorylates PIRE to activate?

Reviewer #4 (Remarks to the Author):

The manuscript by Lee et al. "Dual regulation of reactive oxygen species through direct ubiquitination and phosphorylation of the plant immune regulator RBOHD" aims to show the mechanism of negative regulation of the NADPH oxidase, RBOHD. It is a very important question as, while much is now known about the mechanism of RBOHD activation in response to PAMPs, it is unclear how RBOHD is deactivated or kept inactive. The authors show that PBS1-like kinase 13 (PBL13), which was previously characterized as a negative regulator of PTI that associates with RBOHD (Lin et al., 2015), can interact directly with and phosphorylate the RBOHD C-terminus. The authors further use mutational analysis of RBOHD phosphosites to suggest that this negatively regulates RBOHD activation and protein accumulation. Furthermore, the authors use a protein array to identify a previously uncharacterized RING domain E3 ubiquitin ligase, PIRE, as an interactor of both PBL13 and the RBOHD C-terminus. PIRE directly ubiquitinates RBOHD and genetically acts as a negative regulator of plant defence. These results are novel and of great interest to the community. However, several issues remain with the presented work and the resulting model, and would need to be addressed before the work could be suitable for publication. Major comments:

- Generally, the manuscript is very unclear as to whether/how PBL13 regulates PIRE-mediated ubiquitination/degradation of RBOHD. Is phosphorylation of RBOHD by PBL13 required for PIRE-mediated RBOHD degradation? Is PIRE-mediated RBOHD ubiquitination PBL13-dependent? Is RBOHD ubiquitination PIRE-dependent? As this model is the central, critical thrust of the manuscript, this should be at least somewhat resolved. Does PBL13 directly phosphorylate PIRE? Is PBL13 interaction with PIRE required for its activity? Do RBOHD T912A or S862A variants interact with PIRE? etc.

1. Further re: the model – do the authors propose that PBL13 regulates RBOHD only via degradation? What is the role of Ser862 in that case (as S862D expresses to WT levels but severely impairs ROS production). A critical experiment would be to express WT or S/T Δ D RBOHD in *rbohD pbl13* mutants to determine whether these phosphor-mimetics alone can rescue the enhanced ROS burst phenotype of *pbl13*.

2. It is not always clear from the manuscript whether the proposed mechanism of negative RBOHD regulation happens only at a resting state (and if so, what is a purpose of it?) or it happens as well as a negative feedback loop after RBOHD activation to ensure the correct timing of ROS production. However, there is lack of evidence in the manuscript supporting the second idea and thus, the statement should be clarified. For example, is the association between PBL13 and PIRE affected upon PAMP treatment? Is PIRE-mediated RBOHD ubiquitination increased after PAMP treatment?

3. Fig 2C: the ubiquitination signal on RBOHD in IP sample is not clear. A better image is required.

o It will be interesting to know which type of ubiquitination it is as it is often K-63 ubiquitination and not K-48 which is associated with endocytosis.

o Will RBOHD T912A variant or RBOHD S862A variant get ubiquitinated? It is important to know whether phosphorylation on these sites is a requirement for further ubiquitination, which again relates to the overall interpretation/model.

4. Fig. 4e: it would be good to know how the RBOHD abundance in *pire-1* compares with that in *pbl13*.

Minor comments:

- Introduction/Discussion: several negative regulators of immune receptor complexes have been characterized in Arabidopsis and rice, for example. These should be mentioned and discussed.

- Introduction; p3, 2nd paragraph: at the end of the paragraph, it would be better to cite recent reviews covering PTI signaling, as the individual references currently chosen are not necessarily the primary ones.
- Results are often over-interpreted in the manuscript, for example:
 - o pg 7 "These results indicate that PBL13-mediated phosphorylation of RBOHD T912 negatively affects RBOHD protein accumulation" – the results being referred to indicate that RBOHD levels are higher in pbl13 plants, and are not specific to Ser912.
 - o pg 10 "These findings indicate that PBL13's C-terminal region is important for substrate specificity." The results could also indicate that the truncation is required for trans-phosphorylation generally, as no trans-phosphorylation controls are included (e.g. generic substrate).
 - o pg 5 "these results demonstrate that PBL13 associates with RBOHD in planta in the absence of pathogen perception" – this statement is not fully correct as in *N. benthamiana* PTI is already activated due to *Agrobacterium* infiltration and it is difficult to make a conclusion about the dynamics of interaction. It should be done in stable transgenic *Arabidopsis* lines +/- PMP treatment. At the least, this statement should be re-worded/clarified.
- Fig 1c: the transphosphorylation of the RBOHD C-terminus is weak, which is not necessarily a problem, however, obvious controls are lacking given that this is a cold kinase assay using antibody-based detection instead of ³²P incorporation. Essential controls are +/- ATP addition and +/- substrate (to ensure bands are not derived from kinase breakdown product).
- It would be very informative to determine which of the RBOHD phosphosites is the predominant target of PBL13. This could easily be done by comparing trans-phosphorylation of S/T/A variants in vitro.
- The authors include bar graphs of western blot band intensities as separate panels in figures (e.g. Fig 2b, 4e). These bars are simply densitometric values from the corresponding blot panels in the same figure (one replicate) and as such are not very informative. If the authors would create proper graphs (with error) representing at least 3 independent experiments this would be clearer (this was done for 2e at least), but, as is, these are simply quantification of an inherently qualitative result and should be removed. If the authors wish to include semi-quantification of their individual reps, these could be more simply added as numbers beneath the corresponding bands.
- Fig 4e includes BAK1 as a control, but these bands are not semi-quantified, and BAK1 is an inappropriate control as it is involved in the pathway and it is not known whether BAK1 protein level is affected in *pire* mutant. Accumulation should also be compared with/without PAMP.
- Fig 1A: The input for GFP-RBOHD should be shown, it will be interesting to see it +/- PAMP treatment.
- Fig 3C: it will be of high interest to see whether RBOHD interaction with PBL13 and PIRE is affected by PAMP treatment. Does it happen in resting state, is it changed after PAMP treatment? It should be checked in stable *Arabidopsis* lines.
- Fig. 3E: what is the outcome of direct interaction of PBL13 and PIRE? Will PBL13 phosphorylate PIRE and change its ligase activity?
- Accumulation levels of RBOHD WT, T912D, and T912A should be compared in Col-0 and *pire* to determine whether accelerated degradation of RBOHD T912D is indeed PIRE-dependent.
- Suppl. Fig. 1B: representation of the plates is confusing as they are not at the same position. As there are no serial dilutions, the result is less clear.
- Suppl. Fig 2: Please use various, less-closely related species to demonstrate that RBOHD phosphosites are conserved
- Suppl. Fig 5: please show the inputs for GFP-RBOHD and GFP-PIRE
- It would be nice to have a wider discussion on the interplay between phosphorylation and ubiquitination in the context of receptor kinase signaling and the control of endocytosis/degradation.

Response to Reviewer's Comments

We thank the reviewers' for their constructive comments on our manuscript. We have performed new experiments that verify the genetic and biochemical importance of this work, which is now presented in three new main figures (Figures 4-6), new supplemental figures (Figures S7-S9) and included revisions to all remaining figures.

Reviewer #1:

The paper by Lee et al describes the importance of PBL13 and the newly identified PIRE protein to control RBOHD abundance via ubiquitination. The work is quite novel but suffers from the lack of a few critical experiments, as well issues related to the quality of other data. In my opinion, the two critical experiments missing are

#1-1. The authors do not show data demonstrating *in vivo* that PBL13 can directly phosphorylate RBOHD (or PIRE). This seems essential to their model. Moreover, it is odd that in some of their experiment that S-A mutants show a phenotype while the phosphomimic (S-D) do not. The authors really need to confirm that what they see *in vitro* occurs *in vivo*.

We thank the reviewer for this valuable suggestion. We have generated phosphorylation specific antisera recognizing RBOHD pS862, demonstrated antibody specificity and PBL13-mediated phosphorylation *in vivo* (see Fig. S2). We also attempted to identify RBOHD C-terminal phosphorylation *in vivo* using mass Spectrometry (MS/MS) after digestion with trypsin and chymotrypsin, but were unable to obtain good coverage of the C-terminus surrounding S862/T912, indicating this region of the protein is difficult to identify *in vivo*. This is not a limitation of our ability to generally detect RBOHD phosphorylation, as we can routinely detect phosphorylation of the N-terminus (see our paper in *Cell Host Microbe* 2018: 24: 379-391). Therefore, we focused our experiments on demonstrating the importance of PBL13 for RBOHD stability using antisera recognizing native RBOHD (see Fig. 6a and b).

We did observe variation in significance for the effect of S862A (Fig. 1d) in two independent transgenic lines. While both lines exhibited a higher average ROS level compared to wild type, only S862A line 22 was statistically significant, which may be due to its higher expression level (Fig. 1e). When analyzed for individual experimental replications, S862A line 17 occasionally exhibited a significantly higher ROS burst than wild type, but we believe it is more appropriate to analyze all replications together to demonstrate the most robust results (n >16 individual plants). In our trans-complementation assay, S862A resulted in statistically higher ROS burst compared to wild type (Fig. S3a). We have now explained in the text that our data cumulatively indicates that S862A weakly activates ROS, but S862D completely blocks ROS production. Our phosphomimic lines exhibit consistent phenotypes across biological assays. We have included these results in the revised manuscript text (see page 9, lines 18-23).

#1-2. The story starts with PBL13 and then focuses on the interaction of this protein with PIRE. The assumption is that PBL13 phosphorylates PIRE but this is never shown in the paper. The authors need to address the mechanism of PBL13-PIRE interaction and its physiological

consequence. If PIRE is phosphorylated by PBL13, how it affects the stability of PIRE protein or E3 ligase activity by phosphorylation?

We performed *in vitro* kinase assays using PBL13 and PIRE. Surprisingly, PBL13 did not phosphorylate PIRE, as shown in the new Supplementary Fig 9. Therefore, we investigated if PIRE could be phosphorylated by other kinase(s) *in vivo*. We observed dynamic phosphorylation of PIRE upon *flg22* perception *in vivo* as shown in the new Fig. 5. These results indicate that PIRE can also be regulated by phosphorylation from an unknown protein kinase(s). See text (page 15, lines 7-16).

More minor comments:

#1-3. p. 4.&10 RBOHD was also recently shown to be phosphorylated by the receptor for extracellular ATP (Chen et al. Nature Commun) and this should be mentioned in the intro.

We have included this in the introduction and cited this manuscript (see page 5, lines 8-9).

#1-4. I did not go back and review the earlier papers but was wondering how the data using BIK1 in figure 1C corresponds to earlier work showing that BIK1 can phosphorylate RBOHD? Should be mentioned somewhere in the paper.

BIK1 was previously demonstrated to phosphorylate RBOHD's N-terminus *in vitro* (Molecular Cell 2014, 54:43-5; Cell Host Microbe 2014, 15:329-338.). We now more clearly describe that BIK1 phosphorylates RBOHD's N-terminal region upon PAMP perception in the introduction and results.

#1-5. The authors should consult the Nuhse et al. 2007 Plant J paper to see whether this earlier global, phosphoproteomic analysis detected the phosphorylated residues of RBOHD that they identified. This would lend support that these are indeed phosphorylated in planta.

Please see our response to your comment #1-1. In the revised version, we demonstrate phosphorylation of S862 *in planta*. The Nuhse 2007 paper reported the identification of a single phosphorylated residue on RBOHD's C-terminus, S703. As we did not identify this residue as phosphorylated by PBL13, it is likely that RBOHD's C-terminus is also regulated by phosphorylation through multiple kinases in a similar manner as RBOHD's N-terminus. This possibility is now described in the discussion (see page 20, lines 14-21).

There are a number of questions/concerns about the quality of the data presented. I have listed the majority of these concerns below.

#1-6. In IP experiments (Fig 1a, 3c, Supplementary Fig. 1 and 5), you can see that the GFP or YFP tagged RBOHD is not detected in 'Input' but RBOHD is seen in IP. Of course, it seems to be enriched by IP, but if no band is detected in the input, it is doubtful whether it is the band that is actually caused by the interaction. It seems necessary to verify this. Likewise, in Supplementary

Figure 1a, the YFP-RBOHD in the bottom of input panel, I could see very weak signal in both sides, and it is hard to see the protein in the figure. Why are the signal so low? Did the authors run the protein and use correct antibody for this? Please check this.

We now show RBOHD accumulation in the input for all IPs, including Fig 1a and 3c. We also performed additional experiments to examine RBOHD-PBL13 and RBOHD-PIRE association using Arabidopsis transgenic lines with antisera recognizing native RBOHD, which shows clear RBOHD expression in the input (Fig 3d, Supplementary Fig 1b). For the text corresponding to this see page 6 line 17, page 12 line 5).

#1-7. In Figure 1b: In the bottom of INPUT panel, the 3rd lane have strong band at the size of MBP-RBOHD-N. I think this is for MBP-RBOHD-C. Why they have both MBP-RBOHD-C and N bands? Did the authors use correct proteins? Please check whether this is correct or not.

This was the signal from MBP-RBOHD-N, but there was a “smear” in the immunoblot due to high protein levels. To make these results clear, we have included a different replication clearly showing N and C terminal proteins (see Fig. 1b).

#1-8. Fig. 2b and Fig. 3b and Fig. 4e, all experiments have been done only once? Or, if you have multiple experiments, describe both the mean and the error.

In Fig. 2b we tested two transgenic lines to show altered expression of RBOHD in *pbl13* knockout lines and tested RBOHD expression in each generation (T0, T1, T2). We show the results of two independent replications in Fig. 2a.

We performed protein chip assay to screen PBL13-interacting proteins one time and subsequently tested PBL13 and PIRE interaction with multiple approaches (yeast-two hybrid, in vitro pull-down and co-IP). The protein chip assay is no longer available as all slides have been utilized.

We have replaced the protoplast assay in the previous Fig. 4e with stable transgenic lines expressing RBOHD in Col-0 and the *pire* KO (Fig. 4f). We included four lines as replicates in Fig. 4f. We have described the replicates and error bars in these figures.

#1-9. In vitro MBP pull-down experiments show that PBL13 strongly interacts with the C-term of the RBOHD protein but RBOHD N-term is also interact with PBL13 protein. Thus, I think the N-term of RBOHD is also phosphorylated by PBL13. Therefore, it is necessary to experimentally verify whether PBL13 also phosphorylates the N-term as well as C-term of RBOHD and how it affects ROS production.

We thank the reviewer for this valuable suggestion. We also detected phosphorylation of RBOHD's N-terminus but did not find obvious phenotypes after mutational analysis of N-terminal sites that were identified as phosphorylated by PBL13 *in vitro*. Since both PBL13 and PIRE preferentially associate with RBOHD's C-terminus, which has not been previously reported, we have focused the manuscript on regulation of the C-terminus.

#1-10. The residues of S862, S907, T910, T911 and T912 of RBOHD are shown to be highly conserved in Arabidopsis RBOHB and RBOHC. Verify whether PBL13 causes protein interactions and phosphorylation with these RBOHB or RBOHC. It more needs to be discussed in the 'Discussion'

We are currently investigating this possibility as well as the importance of T912/S862 residues in diverse RBOH homologs. As PBL13 does not possess obvious developmental phenotypes, it is more likely that phosphorylation is a conserved mechanism, but the kinase(s) involved are distinct. However, we believe this is beyond the scope of the current manuscript, but have included this possibility in the discussion (see page 20, lines 12-21).

#1-11. According to previous reports, PBL13 has a palmitoylation site at the N-term and is located at the membrane. I think PIRE should localize on the plasma membrane to ubiquitinate RBOHD. So, Author need to show the subcellular localization of PIRE. In addition, it is necessary to investigate whether the PIRE protein actually interacts with RBOHD on the Plasma membrane. ex) BiFC.

We have performed BiFC assay using PBL13 and PIRE and detected the association on plasma membrane (Fig. S5c). We have conducted co-IPs using microsomal fractions from stable transgenic lines expressing YFP-PIRE and demonstrated PIRE is present and associates with RBOHD (Fig. 3d, Fig 5a).

#1-12. According to previous reports, PBL13- RBOHD interaction was disappeared under flg22 treatment. Verify that the interaction between PIRE and RBOHD diminishes or disappears when treated with flg22 or other elicitor.

We performed co-IP analyses in stable transgenic lines expressing YFP-PIRE. Our results indicate that PIRE is dynamically phosphorylated upon flg22 perception, but constitutively associates with RBOHD (new Fig. 5).

#1-13. In supplementary Figure 1b The position of plates is different from the guide figure (left-side of 1b). Please place plates precisely. It seems that EV (empty vector control) grow in - UTHL media. Why is it growing in -UTLH media? Is it correct? Why don't the authors check the interaction with serial dilution method?

We changed the figure to match the guide. There is weak background in the negative control, but the signal is much stronger in the PBL13+RBOHD-C.

Very minor points:

#1-14. Page 4, "RBOHB and RBOHC mediated ROS generation play an important role in root hair formation and seed ripening, respectively." □ "RBOHB and RBOHC mediated ROS generation play an important role in seed ripening and root hair formation, respectively."

We have removed this statement when the introduction was edited.

#1-15. Page 4, NADPH oxidase □ NADPH oxidase
We have fixed this (see page 5, line 11).

#1-16. Page 5, In vitro □ *In vitro*
We have fixed this (see page 7, line 9).

#1-17. Page 7, Ubiquitinated □ Ubiquitinated
We deleted this.

#1-18. Page 10 ROBHD □ RBOHD, fl22 □ flg22
We deleted this.

#1-19. Page 17 OD600 □ OD600, rbohD □ rbohD
We have fixed this (see page 24 line 13, page 29 line 12).

#1-20. In Fig. 1b, IP □ pull-down
We have fixed this (Fig. 1b).

#1-21. In Fig. 3e, two bands of PBL13-MBP appear when blotted with anti MBP antibody? If upper band is correct size, what is lower band? Is it a truncated protein? And Detection by anti GST antibody showed another band between 1 and 2 lane. Why does the band appear here?

The lower band is truncated form of MBP-PBL13 (now Fig. 3f). The additional band is due to sample spill over.

#1-22. In Fig. 3d, is the series dilution in the loading control panel (bottom) correct? It looks all same. If so, indicate how much dilution has been done. Ex) 10⁻¹ 10⁻² 10⁻³

We added the dilution rate on the figure (see Fig. 3e).

#1-23. In Fig. 4a-b, strong bands appear at about 150kd. In particular, these bands were showed in the western blotting both using anti GST and Flag antibodies. Please clarify what is these bands.

In Fig. 4a, the 150kDa band is GST-UBA1 (E1 enzyme). The ubiquitin is transferred to E1 even in the absence of an active E2/E3 enzyme hence a single flag band is observed in that blot. The labelling was off on earlier gels and has been corrected. We now note tags on E1 and E2 in all figures.

#1-24. In Supplementary Fig. 1b, -UTH picture needs to match up with diagram.

We matched the figure and diagram, see response to reviewer 1, minor point 13.

#1-25. In Supplementary Fig. 2a, why is not S907 listed? And it seems that the mark of pT in T911 is wrong

We changed Fig. S2a to Fig. S2b in the revised version. We have added S907 identification and corrected T911 fragmentation.

#1-26. In Supplementary Fig. 2b, PBL13 phosphorylation sites □ Phosphorylation sites of RBOHD by PBL13

We have now included the conservation of RBOHD's C-terminal phosphorylation in Fig. 4c.

#1-27. In Supplementary Fig. 4. In Method section, Authors describe “The expression of RBOHD was detected by the gene specific primers for RBOHD” in RT-PCR section (Page 18). However, in the Primer list, Reverse anneals to RBOHD but Forward primer binds to pGWB15 vector sequence. So perhaps in the Supplementary Fig. 4, RBOHD transcripts were not detected in Col-0. Or RBOHD was not detected in Col-0 because other plants were overexpressed by 35S promoter? Make this part clear.

We have clarified in the methods section (see page 34, line 20) and in the figure legend (Fig. S4) that RBOHD expression was detected with an HA specific forward primer and an RBOHD reverse primer.

#1-28. In addition, 35S promoter, which is a constitutively expression, is used instead of the native promoter (pGWB15 vector). Therefore, it is thought to be the result of investigating the stability of RBOHD mRNA by pBL13 protein after trans-transcription rather than transcription.

The main point is that RBOHD is regulated at the protein level.

#1-29. Supplementary Fig. 6b E2 and FLAG-UB need to describe, and also need to align the lines.

We added E2 and FLAG-UB in the figure and aligned the lines (Now Fig. S6c).

#1-30. In Supplementary Fig. 7, EF1-a transcripts were detected in –RT reaction. I think it means that contamination of genomic DNA. So, this does not need to be shown here. Instead of –RT, it is recommended to show the results of genotyping rather than this.

We included genotyping data in Fig. S10c and d.

#1-31. In addition, Line numbers are all different.

In the ‘Methods’ section, Col-0, 35S::3xHA-RBOHD/Col-0, 35S::3xHA-RBOHD/pbl13-2, 35S :: 3xHA-RBOHD / PBL13-3xFLAG # 8-3 But Figure legends, 3xHARBOHD /Col-0 # 3, 3xHA-RBOHD / pbl13-2 # 1, or 3xHA-RBOHD / PBL13-3XFLAG # 3. in the picture, The Line number is 1 and 2. So, specify the correct line numbers or names.

We changed the method section (see page 34, lines 16). The numbers in the figures represent different replicates. The actual line number are given with a # sign and included in the figure legends.

Reviewer #2:

This manuscript by Lee et al. identifies two novel components of the regulatory system surrounding PAMP recognition and particularly regulating the NADPH oxidase RBOHD, PBL13 and PIRE. These novel regulators seem to regulate RBOHD stability and add to the increasingly intricate network of regulation surrounding plant immunity. The information the authors present is highly valuable to the plant immunity field, but some concerns need to be addressed to support the hypothesis the authors proposed.

#2-1. In vitro phosphorylation assays using purified proteins can be due to artificial effects in the test tubes (Fig 1C). The authors should compare the RBOHD protein in WT vs *pbl13* mutants to see whether the same faint phosphorylated RBOHD band shift in Fig 1C can be observed. Can authors observe the same phosphorylation sites (Sup Fig 2) using tagged RBOHD-HA purified from WT Arabidopsis, under un-induced or inducing conditions? The reason this is raised is because curiously, no RBOHD band shift is observed in Fig 2A, perhaps the plants need to be induced with elicitor or pathogens to observe the shift? Or the gel should be run longer or under better conditions to observe the shift? If no shift is observed in vivo, it will be difficult to support its relevance in planta.

Please see response to reviewer 1, point #1-1. We have demonstrated phosphorylation of S862 *in planta* and have also performed extensive genetic analyses to verify the importance of both ubiquitination and phosphorylation. In our hands, not all phosphorylated proteins exhibit band shifting, which is hypothesized to occur after extensive phosphorylation at many sites. For example, although RBOHD is well-known to be N-terminally phosphorylated upon flg22 perception, there is no published data demonstrating a band shift.

#2-2. Similarly, in vitro ubiquitination assays with purified proteins with E3s can also be artificial (Fig 4). The authors should overexpress PIRE in Arabidopsis RBOHD-HA transgenic line to test whether RBOHD level is affected, and whether polyubiquitination bands of RBOHD can be observed with and without MG132. What is the PIRE overexpression phenotype?

We appreciate the valuable suggestion. We tested endogenous RBOHD expression with native antisera and 3xHA-RBOHD expression in the *pire* KO line (Fig. 6a and Fig. 4f). RBOHD exhibited increased protein accumulation in the *pire* KO. We have also generated PIRE overexpression lines and demonstrated that overexpression inhibits RBOHD protein accumulation using native antisera recognizing RBOHD (Fig. S8a). Therefore, we used the vacuolar protease inhibitor ConA to demonstrate increased RBOHD accumulation after ConA treatment in wild type, but not in the *pire* or *pbl13* knockout (Fig. S8b). We have also used tandem ubiquitin binding entities (TUBE) to verify RBOHD ubiquitination and the requirement for T912 (Fig. 2d and Fig. 4d). Together, these data demonstrate the importance of RBOHD ubiquitination, the T912 residue and PIRE.

Ubiquitinated transmembrane proteins are endocytosed and subjected to vacuolar degradation (Physiological Reviews 2017, 97:253-281). While MG132 can block proteasome-mediated degradation, proteasomal degradation cross-talks with vacuolar/lysosomal degradation includes

autophagy (FEBS Letters 2010, 584:1393-1398). Therefore, MG132 could indirectly affect vacuolar-mediated degradation. RBOHD is endocytosed and sensitive to wortmannin (Plant Cell 26:1729-1745), and thus we tested RBOHD accumulation after Concanamycin A treatment.

#2-3. Genetic evidence to support the authors' hypothesis is mostly lacking. If the authors' claim is correct, you would predict that the enhanced resistance phenotypes of *pbl13* and *pire* can be fully suppressed by *rboh*d knockout. Detailed phenotypic analysis, including ROS, PTI, and resistance assays of *rboh*d *pbl13* and *rboh*d *pire* double mutants are crucial for the current study.

We have now performed more extensive genetic analyses using stable transgenic lines and crossing. We have generated *pbl13::pire* double knockout mutants and subjected them to detailed phenotypic analyses. The *pbl13*, *pire* and *pbl13::pire* lines exhibited similar levels of RBOHD accumulation, an increase in ROS and enhanced disease resistance (new Fig. 6). Our results are consistent with PBL13 and PIRE working in concert to regulate RBOHD stability.

We hypothesize that PBL13 and PIRE have additional targets beyond RBOHD. We have previously demonstrated that the *pbl13* KO exhibits enhanced MAPK activation upon PAMP treatment (*Plant Physiology* 2015, 169:2950-62) and now demonstrate that the *pire* KO also exhibits enhanced MAPK activation (Fig. S10i). A previous publication demonstrated the *rboh*d KO does not exhibit alterations in MAPK activity compared to wild type Col-0 (*Plant Physiology* 2012, 159:1845-1856.). Furthermore, when RBOHD is completely eliminated (*rboh*d KO), there are pleiotropic effects on growth and defense, potentially due to guarding by another NLR (*New Phytologist* 2019, 221:2160-2175). Therefore, we did not generate higher order KOs in the *rboh*d background.

#2-4 The authors seems to suggest a model where PBL13-phosphorylated RBOHD is a preferred substrate for PIRE. If that is the case, you would predict that overexpression of PIRE in *pbl13* mutant background would have little or no effect on its ROS and enhanced resistance phenotypes. Is that true? At the same time, overexpression of both PBL13 and PIRE simultaneously should yield a *rboh*d knockout mutant phenotype. The authors should seriously test that.

Given the extensive and different requests from all four reviewers, we have attempted to focus our experiments in ways that can address multiple requests. We have generated *pbl13::pire* double mutants, also requested by reviewer 3, and demonstrated that they exhibit similar phenotypes as the single knockouts (see our response to your point #2-3 above; Fig. 6).

#2-5 PIRE has a highly close paralog AT5G62910 in Arabidopsis. They even share similar splicing patterns! What is the evolutionary pattern of PIRE and AT5G62910? Is it similar to RBOHD and PBL13? With this paralog in mind, I am surprised that the authors can still observe a clear phenotype in the single mutant. Is AT5G62910 relevant? The authors should overexpress AT5G62910 to see whether it can reduce RBOHD levels and interact with PBL13 and RBOHD. What is the double mutant phenotype of *pire* with AT5G62910 knockout?

We agree that this is an interesting avenue of research and are planning on comprehensively investigating this family of E3 ligases. However, we feel this is outside the scope of the current study as we can clearly identify PIRE related phenotypes. Furthermore, the closest relative to

PIRE in Arabidopsis exhibits differential expression patterns, with 55% amino acid similarity. Therefore, we hypothesize that additional PIRE family members may regulate different NADPH oxidases.

Relative expression profiles of PIRE1 (the E3 described in this MS) and PIRE2 in biotic stress conditions (68), hormone treatments (45), and tissue samples (46)

#2-6. The interaction between RBOHD and PBL13 was not confirmed using stable transgenic lines. This was an issue for many of the experiments, use of stable transgenics is usually preferable to either expression in Tobacco or protoplast.

We have generated stable transgenic lines and verified that both PBL13 and PIRE are able to associate with RBOHD in Arabidopsis (Fig. S1b, Fig. 3d, and Fig. 5a).

#2-7. In figure 1 E, the levels of both the T912D overexpressers appear to be significantly lower than the other lines, this makes it difficult to draw a conclusion for this mutation. Of less significance but also of note is that the nonspecific band present in Col is missing from the other treatments.

We reasoned that T912D does not accumulate is because it is targeted for enhanced ubiquitination by PIRE (Fig. 1e and Fig. 4c). The nonspecific band in Col-0 is because we loaded 3x the amount of protein in the Col-0 lane only to demonstrate there are no cross-reacting bands. We have now clarified this in the figure legend of Fig. 1e.

#2-8. The claim that Concanamycin A treatment increases the accumulation of ubiquitinated ROBHD is misleading as the increase in ubiquitinated protein correlates with the overall increase in protein.

We have changed this statement (see page 11, line 11).

#2-9. Why do the authors choose to examine interactors of PBL13 rather than RBOHD, when they are attempting to identify regulators of RBOHD? Is there a reason this screening technique wasn't used with RBOHD?

Full-length RBOHD is difficult to purify as it has six transmembrane domains. We were interested in identifying additional PBL13 interacting proteins and this is why we chose PBL13 as a probe. Unfortunately, there are no more PATH arrays left to probe with additional proteins.

#2-10. Figure 3 E doesn't have a set of input samples for the IP.

The figure does have input samples included. We have noted this (now Fig. 3f) for clarity.

#2-11. Figure 4 E doesn't have expression controls for the transient expression. Again, this would be more reliable using stable transgenics.

We generated transgenic lines and tested expression of RBOHD in those lines (see Fig. 4f and Fig. S8a; page 14 and line 10-17).

#2-12. The ROS data shown appears highly variable with extremely large ranges. It is unclear if there is a solution to this but given that this data is central to the paper it is perhaps worth considering additional methodologies.

We have included all experimental repeats and show all data points for growth curves and ROS burst in order to be transparent and demonstrate the true data range. This variability is also observed by other labs when all experiments are analyzed together (Nature 2018, 553:342-346). Our ROS burst analyses is similar to “beautiful” published data when single repeats are considered (see below), but we believe it is more appropriate to analyze all replications together to demonstrate the most robust results over time and many experimental plant replicates.

Reviewer #3:

In the manuscript entitled “Dual regulation of reactive oxygen species through direct ubiquitination and phosphorylation of the plant immune regulator RBOHD”, Lee and Lal et al., showed a novel negative regulatory mechanism of RBOHD by PBL13-mediated phosphorylation and PIRE-mediated ubiquitination. The authors found PBL13 phosphorylates the C-terminus of RBOHD and the phosphorylation is required for the RBOHD activity and stability. Moreover, the authors identified a novel E3 ubiquitin ligase PIRE which ubiquitinates RBOHD in vitro. Overall, the finding is novel and very interesting. However, the authors should provide more evidence to convince readers of their model in Nature communication.

Major points

#3-1. The accumulating evidence show that signaling pathway from FLS2 to RBOHD is different from that to MAPKs activation. For example, BIK1 is required for RBOHD activation but not for MAPKs activation (Feng, F. et al. Nature 485, 114–118). Since pbl13 mutant shows enhanced flg22-inducible ROS burst and MAPKs activation, PBL13 should also regulate another

component required for MAPKs activation, if RBOHD is a true substrate of PBL13. The authors should carefully discuss this point in the discussion.

We agree with this statement and it is likely that the PBL13 kinase has multiple targets in plants. We have also found that like *pbl13*, *pire* also affects MAPK activation (Fig. S10). We have included this possibility in the discussion (see page 21 and lines 1-5).

#3-2. The authors should check endogenous FLS2 and BAK1 protein levels (the antibodies are commercially available) in *pbl13* and *pire* mutants to show the specific accumulation of RBOHD in those mutants. Although the authors show that *pire-1* mutant does not induce the accumulation of BAK1-3xFLAG by transient expression system in protoplast in Fig.4e, it is more proper and accurate way to check the endogenous BAK1 protein by the specific antibody. The authors should also check *flg22*-inducible MAPKs activation in *pire-1* and *pire-2* mutant to check if PIRE specifically regulates RBOHD or not.

We have previously published that FLS2 expression is not altered in the *pbl13* knockout (Plant Physiology 2015, 169:2950-62). We have verified that FLS2 expression is not altered in *pire-1* (see Fig. S10h). We have also investigated MAPK activation in *pire*, revealing that MAPK activation is enhanced upon *flg22* treatment (Fig. S10i). These data indicate that PIRE regulates additional proteins aside from RBOHD.

#3-3. The authors showed the interaction among PBL13, RBOHD and PIRE by in vitro binding assay, Y2H assay and in vivo co-IP assay after overexpression in *N. benthamiana* or Arabidopsis protoplast. These interactions may be very transient in vivo and difficult to see using endogenous proteins or using expressed proteins by own promoters in stable Arabidopsis transformants. If so, it is important to show more genetic link among these three proteins. For example, the authors can check the RBOHD protein level in a *pbl13* *pire* double mutant together with their single mutants to check if PBL13 and PIRE function in the same pathway to regulate RBOHD stability.

This is an excellent suggestion. We have generated *pbl13::pire* double mutants and subjected them to detailed phenotypic analyses. The *pbl13*, *pire* and *pbl13::pire* lines exhibited similar and higher levels of RBOHD accumulation, an increase in ROS and enhanced disease resistance (new Fig. 6). Thus, PBL13 and PIRE function in the same pathway to regulate RBOHD stability.

#3-4. Although the authors found the in vitro phosphorylation sites of RBOHD by PBL13, there is no evidence about in vivo phosphorylation at these sites and the requirement of PBL13 for their phosphorylation in vivo. The authors can try to find T912 phosphorylation after treatment with ConA or in *pire* mutant background.

Please see response to reviewer 1, point #1-1. We have demonstrated phosphorylation of S862 *in planta* and we have performed extensive genetic analyses to verify the importance of both ubiquitination and phosphorylation.

#3-5. Is the phosphorylation of RBOHD at T912 required for the ubiquitination by PIRE? The authors can check the role of T912 phosphorylation on RBOHD ubiquitination by comparing the ubiquitination of WT RBOHD and T912D variant in vitro and possibly in vivo

after treatment with ConA. The authors might also see the reduced ubiquitination of RBOHD variant carrying T912A mutation *in vivo*. It is also worth trying to compare ubiquitinated RBOHD level in Col-0 and *pbl13-2* background after treatment with ConA.

As shown in Fig. 4c-e, we used Tandem Ubiquitin Binding Entities (TUBE) to pull down ubiquitinated RBOHD and demonstrated enhanced ubiquitination of T912D (Fig. 4e). We also verified that the *pire* KO exhibits enhanced RBOHD accumulation (Fig. 4f). Both *pbl13* and *pire* KO are not sensitive to ConA treatment (Fig. S8b). Furthermore, C-terminal deletions of RBOHD block ubiquitination by PIRE and phosphorylation mimics surrounding T912 enhance PIRE ubiquitination *in vitro* (Fig. 4c).

#3-6. S862D does not cause the reduction of RBOHD protein but abolish the ROS production. This result suggests that PBL13 does not only induce RBOHD degradation through ubiquitination by PIRE but directly inhibit the RBOHD activity by the phosphorylation at S862. The authors should carefully explain this possibility in the discussion.

We thank the reviewer for the valuable suggestion. We included the regulation of RBOHD activity, not stability, in the discussion (see page 17. lines 14-22).

#3-7. Do authors check the possibility that PBL13 phosphorylates PIRE to activate?

We performed *in vitro* kinase assays using PBL13 and PIRE. PBL13 did not phosphorylate PIRE, as shown in the new Fig. S9. Therefore, we investigated if PIRE could be phosphorylated by other kinase(s) *in vivo*. We observed dynamic phosphorylation of PIRE upon *flg22* perception *in vivo* as shown in the new Fig 5.

Reviewer #4:

The manuscript by Lee et al. “Dual regulation of reactive oxygen species through direct ubiquitination and phosphorylation of the plant immune regulator RBOHD” aims to show the mechanism of negative regulation of the NADPH oxidase, RBOHD. It is a very important question as, while much is now known about the mechanism of RBOHD activation in response to PAMPs, it is unclear how RBOHD is deactivated or kept inactive. The authors show that PBS1-like kinase 13 (PBL13), which was previously characterized as a negative regulator of PTI that associates with RBOHD (Lin et al., 2015), can interact directly with and phosphorylate the RBOHD C-terminus. The authors further use mutational analysis of RBOHD phosphosites to suggest that this negatively regulates RBOHD activation and protein accumulation. Furthermore, the authors use a protein array to identify a previously uncharacterized RING domain E3 ubiquitin ligase, PIRE, as an interactor of both PBL13 and the RBOHD C-terminus. PIRE directly ubiquitinates RBOHD and genetically acts as a negative regulator of plant defence. These results are novel and of great interest to the community. However, several issues remain with the presented work and the resulting model, and would need to be addressed before the work could be suitable for publication.

Major comments:

#4-1. Generally, the manuscript is very unclear as to whether/how PBL13 regulates PIRE-mediated ubiquitination/degradation of RBOHD. Is phosphorylation of RBOHD by PBL13 required for PIRE-mediated RBOHD degradation? Is PIRE-mediated RBOHD ubiquitination PBL13-dependent? Is RBOHD ubiquitination PIRE-dependent?

In order to investigate how PBL13 and PIRE work together to mediate RBOHD stability, we analyzed the ability of RBOHD C-terminal phosphorylation and lysine mutants for their ability to be ubiquitinated by PIRE (Fig. 4c). Mimicking C-terminal threonine phosphorylation (T912D) enhanced ubiquitination and C-terminal lysine residues were required for ubiquitination (Fig. 4c). Furthermore, we used tandem ubiquitin binding entities (TUBE) to verify RBOHD ubiquitination and the requirement for T912 (Fig. 2d, e and Fig. 4d, e). We generated *pbl13::pire* double mutants and demonstrated that *pbl13*, *pire* and *pbl13::pire* KOs exhibited similar and higher levels of RBOHD accumulation, an increase in ROS and enhanced disease resistance (new Fig. 6). The *pbl13*, *pire*, *pbl13::pire* KOs are no longer sensitive to Concanamycin A treatment, which inhibits vacuolar degradation of ubiquitinated proteins (Fig. S8). Together, these data demonstrate the importance of PBL13-mediated phosphorylation of RBOHD at T912 for subsequent ubiquitination by PIRE.

#4-2. As this model is the central, critical thrust of the manuscript, this should be at least somewhat resolved. Does PBL13 directly phosphorylate PIRE? Is PBL13 interaction with PIRE required for its activity? Do RBOHD T912A or S862A variants interact with PIRE? etc.

Please see response to reviewer 1, point #1-2. We performed in vitro kinase assays using PBL13 and PIRE. Surprisingly, PBL13 did not phosphorylate PIRE, as shown in the new Fig. S9. Therefore, we investigated if PIRE could be phosphorylated by other kinase(s). PIRE was able to constitutively associate with RBOHD (Fig. 5a) and RBOHD T912D and T912A variants were still able to associate with PIRE (Fig. S7). We observed dynamic phosphorylation of PIRE upon *flg22* perception in vivo as shown in the new Fig. 5. These results indicate that PIRE can also be regulated by phosphorylation from an unknown protein kinase(s).

Minor comments:

#4-3. Further re: the model – do the authors propose that PBL13 regulates RBOHD only via degradation? What is the role of Ser862 in that case (as S862D expresses to WT levels but severely impairs ROS production). A critical experiment would be to express WT or S/T□D RBOHD in *rbold pbl13* mutants to determine whether these phosphor-mimetics alone can rescue the enhanced ROS burst phenotype of *pbl13*.

We hypothesize that regulation is likely two-fold through enzyme inactivation coupled with degradation. We have included a description in the discussion (see page 17 and lines 14-22).

#4-4. It is not always clear from the manuscript whether the proposed mechanism of negative RBOHD regulation happens only at a resting state (and if so, what is a purpose of it?) or it happens as well as a negative feedback loop after RBOHD activation to ensure the correct timing of ROS production. However, there is lack of evidence in the manuscript supporting the second idea and thus, the statement should be clarified. For example, is the association between PBL13

and PIRE affected upon PAMP treatment? Is PIRE-mediated RBOHD ubiquitination increased after PAMP treatment?

See response to reviewer 1, comment #1-12. Our current data convincingly demonstrate that this occurs at a resting state to appropriately regulate the initial burst of ROS. There are also two lines of evidence that support a negative feedback loop after RBOHD activation. First, PIRE is dynamically phosphorylated upon flg22 perception (rise and fall that corresponds to the ROS burst kinetics, Fig. 5a-d). Second, *pbl13* and *pire* KO lines exhibit enhanced RBOHD accumulation at a resting state (Fig. 2a, Fig. 4f, and Fig. 6a). We have clarified the importance of RBOHD regulation by PIRE at a resting state and during immune activation in the discussion.

#4-5. Fig 2C: the ubiquitination signal on RBOHD in IP sample is not clear. A better image is required.

Detection of *in vivo* ubiquitination is technically challenging and our results are similar in quality to those published previously (*Plant Cell* 2017, 29:726-745.). To support RBOHD's ubiquitination *in vivo*, we used Tandem Ubiquitin Binding Entities (TUBE) to pull down ubiquitinated RBOHD and demonstrated enhanced ubiquitination of T912D (Fig. 4c-e).

#4-6. It will be interesting to know which type of ubiquitination it is as it is often K-63 ubiquitination and not K-48 which is associated with endocytosis.

We agree with the reviewer, but this is beyond the scope of the current manuscript. Ubiquitination of transmembrane proteins is thought to be a signal for sorting and subsequent endocytosis (*Physiological reviews* 2016, 97: 253-281). Transmembrane proteins can also exhibit mixed K-63/K-48 linkages, making it difficult to determine cellular fate (*J Lipid Research* 2013, 54:1410-1420; *Proceedings of the National Academy of Sciences* 2018, 115:E1401-E1408.). Our analyses using Concanamycin A clearly establish a role of vacuolar degradation likely mediated by K-63 linkages on transmembrane proteins.

#4-7. Will RBOHD T912A variant or RBOHD S862A variant get ubiquitinated? It is important to know whether phosphorylation on these sites is a requirement for further ubiquitination, which again relates to the overall interpretation/model.

We now show in Fig. 4 that T912D exhibits enhanced ubiquitination and T912A has a reduced ubiquitination signal *in planta*. T912A is not completely blocked in ubiquitination. Thus, phosphorylation of T912 is a major signal, it is not the only one. We used *in vitro* assays to demonstrate C-terminal threonine phosphorylation enhanced ubiquitination and C-terminal lysine residues were required for ubiquitination (Fig. 4c).

#4-8. Fig. 4e: it would be good to know how the RBOHD abundance in *pire-1* compares with that in *pbl13*.

We tested RBOHD protein accumulation in *pbl13*, *pire*, *pbl13::pire* and Col-0 (Fig. 6a). RBOHD exhibited similar and higher level of protein accumulation in *pbl13*, *pire*, and

pbl13::pire compared to Col-0. We also demonstrated that RBOHD was no longer sensitive to Concanamycin A treatment in *pbl13* or *pire* KOs (Fig. S8b).

Minor comments

#4-9. Introduction/Discussion: several negative regulators of immune receptor complexes have been characterized in Arabidopsis and rice, for example. These should be mentioned and discussed.

We included a description of the negative regulators in introduction (see page 4, lines 4-15) and discussion (see page 18, line 15).

#4-10. Introduction; p3, 2nd paragraph: at the end of the paragraph, it would be better to cite recent reviews covering PTI signaling, as the individual references currently chosen are not necessarily the primary ones.

We cited *Annual Review of Phytopathology* (2017) 55:109-137 (see page 4, line 2).

• Results are often over-interpreted in the manuscript, for example:

#4-11. pg 7 “These results indicate that PBL13-mediated phosphorylation of RBOHD T912 negatively affects RBOHD protein accumulation” – the results being referred to indicate that RBOHD levels are higher in *pbl13* plants, and are not specific to Thr912.

The RBOHD T912D mutant exhibited reduced accumulation in Arabidopsis and *N. benthamiana*. We changed the text to “These results indicate that PBL13 negatively regulates RBOHD protein accumulation.” (see page 10 and line 15).

#4-12. pg 10 “These findings indicate that PBL13’s C-terminal region is important for substrate specificity.” The results could also indicate that the truncation is required for trans-phosphorylation generally, as no trans-phosphorylation controls are included (e.g. generic substrate).

We thank the reviewer for this suggestion. We now describe this possibility in the results (see page 8, lines 4-7).

#4-13. pg 5 “these results demonstrate that PBL13 associates with RBOHD in planta in the absence of pathogen perception” – this statement is not fully correct as in *N. benthamiana* PTI is already activated due to Agrobacterium infiltration and it is difficult to make a conclusion about the dynamics of interaction. It should be done in stable transgenic Arabidopsis lines +/- PMP treatment. At the least, this statement should be re-worded/clarified.

We now verified that PBL13 could associate with RBOHD at a resting state using stable Arabidopsis transgenic lines (Fig. S1b).

#4-14. Fig 1c: the transphosphorylation of the RBOHD C-terminus is weak, which is not necessarily a problem, however, obvious controls are lacking given that this is a cold kinase assay

using antibody-based detection instead of ³²P incorporation. Essential controls are +/- ATP addition and +/- substrate (to ensure bands are not derived from kinase breakdown product).

We performed *in vitro* kinase assay of RBOHD C-terminus by PBL13 +/- ATP. We were able to detect clear phosphorylation of RBOHD-C terminus by PBL13 in the presence of ATP (Fig. S9).

#4-15. It would be very informative to determine which of the RBOHD phosphosites is the predominant target of PBL13. This could easily be done by comparing trans-phosphorylation of S/T/A variants *in vitro*.

We agree this could be done with more experimental replicates, but the most important data is the biological phenotype *in planta*, which clearly exists for S862 and T912 residues.

#4-16. The authors include bar graphs of western blot band intensities as separate panels in figures (e.g. Fig 2b, 4e). These bars are simply densitometric values from the corresponding blot panels in the same figure (one replicate) and as such are not very informative. If the authors would create proper graphs (with error) representing at least 3 independent experiments this would be clearer (this was done for 2e at least), but, as is, these are simply quantification of an inherently qualitative result and should be removed. If the authors wish to include semi-quantification of their individual reps, these could be more simply added as numbers beneath the corresponding bands.

Please see response to reviewer 1, point #1-8.

#4-17. Fig 4e includes BAK1 as a control, but these bands are not semi-quantified, and BAK1 is an inappropriate control as it is involved in the pathway and it is not known whether BAK1 protein level is affected in *pire* mutant. Accumulation should also be compared with/without PAMP.

We have removed this data and generated transgenic lines expressing RBOHD in the *pire* KO as well as verified that RBOHD exhibits higher expression in the *pire* KO using antisera recognizing native RBOHD (Fig. 4f and Fig. 6a). We have also quantified RBOHD accumulation +/- *flg22* (Fig. 5b).

#4-18. Fig 1A: The input for GFP-RBOHD should be shown, it will be interesting to see it +/- PAMP treatment.

We have included the input in Fig. 1a. We have previously demonstrated that PBL13 disassociates upon *flg22* perception (Plant Physiology 2015, 169:2950-62).

#4-19. Fig 3C: it will be of high interest to see whether RBOHD interaction with PBL13 and PIRE is affected by PAMP treatment. Does it happen in resting state, is it changed after PAMP treatment? It should be checked in stable Arabidopsis lines.

Please see response to reviewer 1, point #1-2. We performed co-IP analyses in stable transgenic lines expressing YFP-PIRE and demonstrated constitutive association between PIRE and RBOHD. Our results indicate that PIRE is dynamically phosphorylated upon flg22 perception by an unknown kinase, but constitutively associates with RBOHD (new Fig. 5).

#4-20. Fig. 3E: what is the outcome of direct interaction of PBL13 and PIRE? Will PBL13 phosphorylate PIRE and change its ligase activity?

Please see response to reviewer 1, point #1-2. As shown in Fig. S9, PBL13 does not directly phosphorylate PIRE *in vitro*. The association between PBL13 and PIRE could facilitate PIRE's interaction with RBOHD to regulate its stability. Mimicking phosphorylation of RBOHD T912 enhanced ubiquitination by PIRE, demonstrating that phosphorylation enhances RBOHD's ubiquitination by PIRE (Fig. 4c)

#4-21. Accumulation levels of RBOHD WT, T912D, and T912A should be compared in Col-0 and *pire* to determine whether accelerated degradation of RBOHD T912D is indeed PIRE-dependent.

We tested RBOHD expression in Col-0 and *pire* knockout lines, showing enhanced expression of RBOHD in *pire* KOs (Fig. 4f and Fig. 6a).

#4-22. Suppl. Fig. 1B: representation of the plates is confusing as they are not at the same position. As there are no serial dilutions, the result is less clear.

We fixed the diagram.

#4-23. Suppl. Fig 2: Please use various, less-closely related species to demonstrate that RBOHD phosphosites are conserved

We have now performed sequence analysis using data from 6 species, including 29 RBOH homologs. The phosphosites and lysine residues are highly conserved. This data is now shown in Fig. 4c.

#4-24. Suppl. Fig 5: please show the inputs for GFP-RBOHD and GFP-PIRE

We have repeated the IPs and now show inputs for IP between RBOHD and PIRE in Fig. 3d.

#4-25. It would be nice to have a wider discussion on the interplay between phosphorylation and ubiquitination in the context of receptor kinase signaling and the control of endocytosis/degradation.

We thank the reviewer for this valuable suggestion. We extended the discussions section (see page 18, lines 16-20; page 19, lines 4-11)

Reviewers' comments:

Reviewer #1 (Remarks to the Author):

I was not a reviewer on the original submission but read through the reviews and the authors' rebuttal carefully and felt, in general, that the authors had responded in a constructive way to each of the previous criticisms, as well as providing important new experimental results.

Of the issues remaining from this rebuttal is the question of how PBL13 impacts PIRE activity. One possibility, now excluded by the authors, is that PBL13 directly phosphorylates PIRE. The authors now show that PIRE is phosphorylated *in vivo* upon elicitation but, based on their *in vitro* experiments, this is not mediated by PBL13. These results, although helpful in excluding one possibility, do not provide affirmative information to support the authors' model. The previous reviewers saw this as a weakness and I believe this weakness remains.

Here are few additional comments based on my own reading of the manuscript and questions that I had about experimental procedures and some of the results:

The authors use a variety of epitope tagged proteins in their work. However, it is not clear whether any of these proteins in the current work or previous publications were shown to function normally (e.g., by testing that the epitope tagged versions would complement the corresponding gene mutations). For example, epitope tagging of FLS2 appears to inactivate the protein. Since artifacts can arise it is important to show that epitope tagging does not affect activity and, therefore, the results obtained with such constructs are likely to reflect the native protein.

What is the ROS phenotype when the RBOHDK909AK013AK918A mutant variant is expressed in the *rbohD* mutant background or in the *pbl13* *pire* background? This would be a good direct test of the importance of ubiquitination for the phenotypes they are measuring.

Figure 5. Did they check for PIRE phosphorylation *in planta* using the *pbl13* mutant background. *In vitro* assays are useful but no substitute for *in planta* measurements. This would likely require MS analysis since gel shifts usually require multiple phosphorylation, only one of which could be impacted by *pbl13*.

ROS production in leaves is generally thought to mediate stomatal aperture and by so doing pathogen resistance. However, the authors used direct infiltration of leaves in their pathogen assays, which largely negates the importance of stomatal entry of the pathogen, suggesting some other mechanism by which regulation of RBOHD is impacting immunity. The authors should address this providing a better explanation as to how they see PBL13, PIRE and RBOHD affecting immunity that would act on infiltrated as opposed to flood inoculated leaves.

Reviewer #2 (Remarks to the Author):

The authors have carried out additional experiments which addressed most of my concerns. The current version is much improved as a result and I have no further comments or suggestions.

Reviewer #3 (Remarks to the Author):

The authors answered the most comments I raised, and the manuscript is significantly improved. I want to raise a few more points that need to be revised before publishing in Nature communication.

In Fig.3D and 5A, YFP-PIRE showed double bands in both the presence or absence of flg22, while the authors claimed that flg22 treatment induces the mobility shift of PIRE based on the data of Fig5CDE. This looks controversial and confusing. I think the truth would be that some portion of PIRE is phosphorylated even in the absence of flg22, but flg22 treatment boosts the phosphorylation. The author should provide more clear data or explain more carefully.

There must be two negative regulations of RBOHD. One would be the basal regulation of RBOHD to control the basal activity or protein level as shown in Fig.7. The other would be negative regulation to shutdown the activated RBOHD. Although the basal RBOHD protein level is high in pbl13 and pire mutants, this does not mean that PBL13 and PIRE work only on the basal regulation. In addition, the phosphorylation of S862 by PBL13 would directly inhibit the RBOHD activity. The authors should show representative kinetic data of ROS production in pbl13 mutant, pire mutant and plants expressing S862A, S862D, T912A, and T912D, and discuss more based on the kinetic data.

Reviewer #4 (Remarks to the Author):

In the revised manuscript, Lee, Lal, et al further clarify the roles of PBL13 and PIRE in the regulation of RBOHD. The authors have performed experiments that significantly improve the quality of the manuscript. However, a mechanistic dynamic link between PBL13 and PIRE is still missing beyond their simple biochemical interaction, as for example PBL13 does not phosphorylate PIRE, and therefore the interdependence between both proteins for the control of RBOHD accumulation is still unclear.

Below are additional specific points:

1. The authors do not provide any evidence that PBL13 can regulate PIRE or vice-versa. While the phosphomimetic mutant analysis suggests that phosphorylation leads to enhanced PIRE-dependent degradation of RBOHD, this is only true for one site. Thus, there is currently no strong evidence of an interplay between these two regulators of RBOHD, apart from interaction. While the work is still of interest, this requires changes to the manuscript, which could largely be achieved by text changes. For example, sentences such as "a we have identified a link between PBL13-mediated RBOHD-C phosphorylation and the E3 ubiquitin ligase PIRE" (lines392-393) must be removed/amended.

2. Regarding the initial comment: "#4-21.Accumulation levels of RBOHD WT, T912D, and T912A should be compared in Col-0 and pire to determine whether accelerated degradation of RBOHD T912D is indeed PIRE-dependent." The authors replied: "We tested RBOHD expression in Col-0 and pire knockout lines, showing enhanced expression of RBOHD in pire KOs (Fig. 4f and Fig. 6a)." This is not an appropriate response, as it fails to address the original comment, which requested protein degradation assays for RBOHD wt and mutants in Col vs pire vs pbl13 backgrounds.

3. Higher RBOHD protein levels in the mutants in Fig.6A could be explained by higher RBOHD expression in the mutants in Fig.6B. The results presented in Fig. 6B seem in contradiction to the statement done in relation to results presented in Fig. S4 (where RBOHD expression was measured by non-quantitative PCR which is not correct). Please address.

4. Regarding the initial comment: "It would be very informative to determine which of the RBOHD phosphosites is the predominant target of PBL13. This could easily be done by comparing trans-phosphorylation of S/T↔A variants in vitro." The authors replied: "We agree this could be done with more experimental replicates, but the most important data is the biological phenotype in planta, which clearly exists for S862 and T912 residues."

While this is true regarding p-site mutational analyses, the authors still do not show any evidence that PBL13 actually phosphorylates these sites *in vivo*, though they now do provide antibody data suggesting S862 may be phosphorylated *in vivo*. However, the issue still remains that there are no data regarding site-specificity of RBOHD-C transphosphorylation by PBL13 *in vitro* or *in vivo*.

5. The authors suggest the C-terminal repeats of PBL13 may be required for transphosphorylation and/or substrate recognition (lines 162-163). The former could be tested easily with use of a generic substrate (i.e. myelin basic protein) to test whether the truncated PBL13 is actually transphosphorylation-competent.

6. In the abstract: "the PBL13 receptor-like cytoplasmic kinase phosphorylates RBOHD's C-terminus and two phosphorylated residues (S862 and T912) are required for RBOHD activity and stability, respectively". Please reformulate, as it gives the impression that T912 phosphorylation is required for RBOHD stability, while it actually promotes its degradation.

7. In Fig. 4c not only the blot with α -MBP should be shown but as well a blot with α -FLAG(Ub) antibodies to demonstrate that the laddering is indeed ubiquitination. The same way as it is done in Fig. 4a,b.

8. Why is the western blot in Fig. 4d cut in 2 parts when it is the same western blot developed with α -HA antibodies? Please show the whole blot so that it is clear that the smear is indeed coming up from RBOHD.

9. It is not clear what signal the authors quantify in Fig. 4e as one can hardly see any signal in either WT or T912A lines. Quantification does not make much sense in this case, and the difference between the lines is clear in Fig. 4d. Similarly, is the quantification shown in Fig. 2e really required/appropriate?

10. Please provide the curves for ROS measurements, so that one can compare the dynamic of this response between the lines tested.

11. Fig. S8: This is quite an important result and could be moved to main figures. Same comment for Fig. S1.

12. There is higher RBOHD levels compared to Col-0 at 5 and 15 min after flg22 treatment in *pire* but not in *pbl13*. Would it suggest that RBOHD regulation by PIRE after flg22 treatment is PBL13-independent? In agreement with your answer to #4-18.

13. Lines 101-102: it is not correct to state generally that BIK1 is a "positive regulators of defense signaling" as it has recently been shown to act as a negative regulator of defense signaling mediated by LRR-RLPs (Wan et al., *New Phytol.* 2019). Similar comment for line 124, where it is wrongly indicated that PBL3 is the only RLCK acting a negative regulator of plant innate immune responses.

14. The manuscript still requires significant editing for style and clarity, e.g.:

a. All protein names should be consistently capitalized.

b. Line 79, nicotinamide should not be capitalized.

c. Lines 128-130 should be re-worded to make clear *Lti6b* was the control bait not prey.

d. Line 158 "The BIK1 kinase" should just be "BIK1".

e. Line 202: "mimicking dephosphorylation" should rather be a "non-phosphorylatable mutant", as it otherwise gives a false impression of dynamic regulation.

f. Line 220: "Due to the established cross-talk between phosphorylation and ubiquitination"; add reference(s).

etc...

15. Fig 4c. please specific species names.

16. Mutant nomenclature is incorrect throughout the manuscript, e.g. "pbl13-2::pire-1" should be "pbl13 pire-1"

17. Given the scale of Fig. 4e, it seems inappropriate to include a break in the y-axis to increase the apparent differences.

We thank reviewers for their helpful comments on the revised manuscript. All reviewers appreciated the extensive new experiments that we performed to address most of the issues raised during the last review. We have taken editorial guidance into consideration focusing on the relevance of PIRE and PBL13 mediated phosphorylation and ubiquitination respectively when addressing reviewer comments, in bold below. In this version of the revision, we have textually addressed all comments (highlighted in yellow), included kinetics of the ROS burst in different genetic backgrounds, verified no alteration in RBOHD at the transcript level by qPCR in different genetic backgrounds and investigated the importance phosphorylated residues in the pire knockout. New data are shown in Fig. 5c, Fig. S2b, Fig. S5, Fig. S9 and Table S1.

Reviewer #1:

I was not a reviewer on the original submission but read through the reviews and the authors' rebuttal carefully and felt, in general, that the authors had responded in a constructive way to each of the previous criticisms, as well as providing important new experimental results.

Of the issues remaining from this rebuttal is the question of how PBL13 impacts PIRE activity. One possibility, now excluded by the authors, is that PBL13 directly phosphorylates PIRE. The authors now show that PIRE is phosphorylated in vivo upon elicitation but, based on their in vitro experiments, this is not mediated by PBL13. These results, although helpful in excluding one possibility, do not provide affirmative information to support the authors' model. The previous reviewers saw this as a weakness and I believe this weakness remains.

While we agree that this is an interesting area for future research, this is beyond the scope of the current manuscript and would require extensive new experimentation to uncover a possible kinase(s) that phosphorylate PIRE. As per editorial guidance from the previous review, we focused our efforts on RBOHD as opposed to how PBL13 impacts PIRE activity in the revision. However, we do plan to investigate this line of research in the future.

Here are few additional comments based on my own reading of the manuscript and questions that I had about experimental procedures and some of the results:

The authors use a variety of epitope tagged proteins in their work. However, it is not clear whether any of these proteins in the current work or previous publications were shown to function normally (e.g., by testing that the epitope tagged versions would complement the corresponding gene mutations). For example, epitope tagging of FLS2 appears to inactivate the protein. Since artifacts can arise it is important to show that epitope tagging does not affect activity and, therefore, the results obtained with such constructs are likely to reflect the native protein.

We have previously demonstrated that epitope-tagged versions of PBL13 can complement the *pbl13* knockout (Plant Physiology 2015, 169:2950-2962). Previously published reports have shown that the epitope-tagged RBOHD functions normally (Molecular Cell 2014, 54:43-55; J Experimental Botany 2016, 67:1663-1667; etc). Finally, we have demonstrated in this manuscript that YFP-tagged PIRE and FLAG-tagged PBL13 exhibit the expected phenotypes for RBOHD protein accumulation (Figures 2, 3,

and Supplemental figures). We have now explicitly stated this in the manuscript on page 6, lines 124-128.

What is the ROS phenotype when the RBOHDK909AK013AK918A mutant variant is expressed in the *rboh*d mutant background or in the *pbl13* *pire* background? This would be a good direct test of the importance of ubiquitination for the phenotypes they are measuring.

We have presented data demonstrating that RBOHD is ubiquitinated *in planta* (Figures 2 and 4) and RBOHD protein accumulation is affected in a PBL13 and PIRE-dependent manner (Figures 2, 3, and 6). To further understand the relationship between PIRE and PBL13 we analyzed the accumulation of RBOHD WT, T912D, and T912A in Col-0 and *pire* protoplasts (see Fig S9). We observed expected pattern of accumulation in Col-0 (higher accumulation of T912A and lower accumulation of T912D). In *pire*, RBOHD WT and T912A accumulated to the similar level. However, the accumulation of RBOHD T912D was reduced in both Col-0 and *pire*. These data indicate that mimicking or “forcing” complete phosphorylation at T912 may enable additional components (perhaps other E3 ligases) to target RBOHD in the absence of PIRE. These data would also be consistent with the concept of resiliency built into key immune signaling modules. For example, RBOHD-N is phosphorylated by multiple kinases (BIK1, CDPKs, SIK1, etc) and a recent preprint has identified the cysteine rich receptor like kinase CRK2 can phosphorylate RBOHD S862 (doi.org/10.1101/618819). See page 16, lines 341-349 and page 20, lines 429-434.

Figure 5. Did they check for PIRE phosphorylation in planta using the *pbl13* mutant background. In vitro assays are useful but no substitute for in planta measurements. This would likely require MS analysis since gel shifts usually require multiple phosphorylation, only one of which could be impacted by *pbl13*.

As noted above, per previous editorial guidance we focused our efforts on RBOHD and not how PIRE is regulated by PBL13. This will be a future area of research that we will be addressing.

While it is possible that PBL13 may phosphorylate PIRE, we can still detect PIRE shifting when transiently expressed in *Nicotiana benthamiana*, which lacks PBL13. PIRE is conserved across land plants, but PBL13 is not found outside *Brassicaceae*. We have clearly described these possibilities in the discussion on page 22 lines 488-490.

ROS production in leaves is generally thought to mediate stomatal aperture and by so doing pathogen resistance. However, the authors used direct infiltration of leaves in their pathogen assays, which largely negates the importance of stomatal entry of the pathogen, suggesting some other mechanism by which regulation of RBOHD is impacting immunity. The authors should address this providing a better explanation as to how they see PBL13, PIRE and RBOHD affecting immunity that would act on infiltrated as opposed to flood inoculated leaves.

We have now described in the discussion section how PBL13, PIRE and RBOHD could affect immunity in stomata (see page 20, lines 442-446). Briefly, we know that RBOHD is also required for guard cell closure in response to pathogen perception and we expect that the *pbl13* and *pire* knockout lines would thus exhibit stronger disease resistance after surface inoculation than syringe infiltration.

Reviewer #2:

The authors have carried out additional experiments which addressed most of my concerns. The current version is much improved as a result and I have no further comments or suggestions.

We thank the reviewer for their positive assessment of the revised manuscript and acknowledgement of the extensive experiments we performed in the previous revision.

Reviewer #3:

The authors answered the most comments I raised, and the manuscript is significantly improved. I want to raise a few more points that need to be revised before publishing in Nature communication.

We thank the reviewer for the positive assessment our revision.

In Fig.3D and 5A, YFP-PIRE showed double bands in both the presence or absence of flg22, while the authors claimed that flg22 treatment induces the mobility shift of PIRE based on the data of Fig5CDE. This looks controversial and confusing. I think the truth would be that some portion of PIRE is phosphorylated even in the absence of flg22, but flg22 treatment boosts the phosphorylation. The author should provide more clear data or explain more carefully.

We thank the reviewer for this careful observation. As reviewer noticed, there is a small amount of PIRE phosphorylation at a resting state, but flg22 perception transiently boosts phosphorylation. We have now described these results on page 16, lines 353-356.

There must be two negative regulations of RBOHD. One would be the basal regulation of RBOHD to control the basal activity or protein level as shown in Fig.7. The other would be negative regulation to shutdown the activated RBOHD. Although the basal RBOHD protein level is high in *pbl13* and *pire* mutants, this does not mean that PBL13 and PIRE work only on the basal regulation. In addition, the phosphorylation of S862 by PBL13 would directly inhibit the RBOHD activity. The authors should show representative kinetic data of ROS production in *pbl13* mutant, *pire* mutant and plants expressing S862A, S862D, T912A, and T912D, and discuss more based on the kinetic data.

We thank this reviewer for the suggestion and agree with this model of dual regulation of RBOHD. We have included a more detailed discussion on dual regulation on page 19, lines 425-429.

We have included representative curves for ROS measurements in the new Supplemental Figure S5, which revealed that RBOHD^{T912A} lines exhibited a slightly shifted curve, with more robust ROS production at earlier time points compared with wild-type RBOHD complementation lines (Supplemental Fig 5a). The *pbl13*, *pire* and *pbl13 pire* knockouts also exhibited a shifted ROS curve, with more ROS production at earlier time points (Supplemental Fig 5c). Collectively, these data indicate that phosphorylation and ubiquitination of RBOHD can regulate both the amplitude and dynamics of ROS production upon flg22 perception. We have addressed these data in the results and discussion (see page 11, lines 227-234 and page 19, lines 409-416.

While our manuscript was under revision, a pre-print posted on BioRxV notes that the receptor-like kinase CRK2 can also phosphorylate RBOHD in vitro, including S862 (doi.org/10.1101/618819). Thus, it is possible that multiple kinases converge upon RBOHD's C-terminus to ensure appropriate phosphorylation kinetics of critical C-terminal residues. Similarly, multiple kinases converge upon the N-terminus of RBOHD (BIK1, CDPKs, SIK1, etc). We have included this in the discussion on page 20, lines 429-434.

Reviewer #4:

In the revised manuscript, Lee, Lal, et al further clarify the roles of PBL13 and PIRE in the regulation of RBOHD. The authors have performed experiments that significantly improve the quality of the manuscript. However, a mechanistic dynamic link between PBL13 and PIRE is still missing beyond their simple biochemical interaction, as for example PBL13 does not phosphorylate PIRE, and therefore the interdependence between both proteins for the control of RBOHD accumulation is still unclear.

We thank the reviewer for the positive assessment of our revised manuscript. Per editorial guidance from the previous revision, we primarily focused our experiments on regulation of RBOHD as opposed to how PIRE and PBL13 regulate each other. In the future, we will be following up on the relationship between PIRE and PBL13

Specific points:

1. The authors do not provide any evidence that PBL13 can regulate PIRE or vice-versa. While the phosphomimetic mutant analysis suggests that phosphorylation leads to enhanced PIRE-dependent degradation of RBOHD, this is only true for one site. Thus, there is currently no strong evidence of an interplay between these two regulators of RBOHD, apart from interaction. While the work is still of interest, this requires changes to the manuscript, which could largely be achieved by text changes. For example, sentences such as “a we have identified a link between PBL13-mediated RBOHD-C phosphorylation and the E3 ubiquitin ligase PIRE” (lines392-393) must be removed/amended.

We have removed or amended such statements.

2. Regarding the initial comment: “#4-21.Accumulation levels of RBOHD WT, T912D, and T912A should be compared in Col-0 and *pire* to determine whether accelerated degradation of RBOHD T912D is indeed PIRE-dependent.” The authors replied: “We tested RBOHD expression in Col-0 and *pire* knockout lines, showing enhanced expression of RBOHD in *pire* KOs (Fig. 4f and Fig. 6a).”

This is not an appropriate response, as it fails to address the original comment, which requested protein degradation assays for RBOHD wt and mutants in Col vs *pire* vs *pbl13* backgrounds.

We agree that the proposed experiments will provide better insights on the relationship between PIRE and PBL13. Therefore, we analyzed the accumulation of RBOHD WT, T912D, and T912A in Col-0 and *pire* protoplasts (data shown in Fig S9). We observed expected pattern of accumulation in Col-0 (higher accumulation of T912A and lower accumulation of T912D). In *pire*, RBOHD WT and T912A

accumulated to the similar level. However, the accumulation of RBOHD T912D was reduced in both Col-0 and *pire*. These data indicate that mimicking or “forcing” complete phosphorylation at T912 may enable additional components (perhaps other E3 ligases) to target RBOHD in the absence of PIRE. These data would also be consistent with the concept of resiliency built into key immune signaling modules. For example, RBOHD-N is phosphorylated by multiple kinases (BIK1, CDPKs, SIK1, etc) and a recent preprint has identified the cysteine rich receptor like kinase CRK2 can phosphorylate RBOHD S862 (doi.org/10.1101/618819). See page 16, lines 341-349 and page 20, lines 429-434.

3. Higher RBOHD protein levels in the mutants in Fig.6A could be explained by higher RBOHD expression in the mutants in Fig.6B. The results presented in Fig. 6B seem in contradiction to the statement done in relation to results presented in Fig. S4 (where RBOHD expression was measured by non-quantitative PCR which is not correct). Please address.

The data presented in Figure 6B is protein expression not RNA expression. This was originally noted in the legend. For clarity, we have now relabeled the graph as “relative protein expression”. We have also included new data in Figure 5C, where *RBOHD* transcript levels were analyzed by qPCR in Col-0, *pbl13*, *pire*, and *pbl13/pire* genotypes. qPCR analyses did not identify significant differences in RBOHD expression in Col-0, *pire*, *pbl13*, and *pbl13/pire* (Fig. 5c). Collectively, these data indicate that the primary effect on RBOHD expression is at the protein level in *pbl13* and *pire* genotypes.

4. Regarding the initial comment: “It would be very informative to determine which of the RBOHD phosphosites is the predominant target of PBL13. This could easily be done by comparing trans-phosphorylation of S/T Δ A variants in vitro.” The authors replied: “We agree this could be done with more experimental replicates, but the most important data is the biological phenotype in planta, which clearly exists for S862 and T912 residues.”

While this is true regarding p-site mutational analyses, the authors still do not show any evidence that PBL13 actually phosphorylates these sites in vivo, though they now do provide antibody data suggesting S862 may be phosphorylated in vivo. However, the issue still remains that there are no data regarding site-specificity of RBOHD-C transphosphorylation by PBL13 in vitro or in vivo.

We have attempted to address site specificity *in vivo*, however we were unable to detect the C-terminal residues of RBOHD by mass spectrometry due to short peptide fragments and poor coverage (even when alternative enzymes were used). The N-terminus of RBOHD is easily detected and our lab can reproducibly detect N-terminal phosphorylation in vitro and in vivo. We have demonstrated that RBOHD S862 phosphorylation is enhanced upon co-expression with PBL13 *in planta* (Supplemental Figure 2c-e).

In order to address site specificity of PBL13 *in vitro*, we quantified phosphorylation of RBOHD residues phosphorylated by PBL13. These data demonstrate that S862 had the highest level of phosphorylation and the data are now included in Table S1. We also mutated all six phosphorylated residues on the C-terminus of RBOHD to alanine. We have now included data demonstrating that the phosphorylation of RBOHD is blocked in the 6A mutant in Figure S2B (see page 8 lines 170-176).

5. The authors suggest the C-terminal repeats of PBL13 may be required for transphosphorylation and/or substrate recognition (lines 162-163). The former could be tested easily with use of a generic substrate (i.e myelin basic protein) to test whether the truncated PBL13 is actually transphosphorylation-competent.

The reviewer pointed out this in the last revision and suggested us to address this by textual edits. We did this in the previous revision.

6. In the abstract: “the PBL13 receptor-like cytoplasmic kinase phosphorylates RBOHD’s C-terminus and two phosphorylated residues (S862 and T912) are required for RBOHD activity and stability, respectively”. Please reformulate, as it gives the impression that T912 phosphorylation is required for RBOHD stability, while it actually promotes its degradation.

We have revised this sentence, see page 2, line 30.

7. In Fig. 4c not only the blot with a-MBP should be shown but as well a blot with a-FLAG(Ub) antibodies to demonstrate that the laddering is indeed ubiquitination. The same way as it is done in Fig. 4a,b.

Because PIRE is also present in the reactions, the anti-UB blot will not be able to differentiate between PIRE auto-ubiquitination as shown in Fig 4a, b or RBOHD trans-ubiquitination. For this reason we have referred to the laddering as opposed to an anti-UB blot.

8. Why is the western blot in Fig. 4d cut in 2 parts when it is the same western blot developed with a-HA antibodies? Please show the whole blot so that it is clear that the smear is indeed coming up from RBOHD.

These are two different blots. The lower blot is the input while the upper is the IP. We have now clarified this in the figure and figure legend for Fig 4d. Because we performed IP first to enrich for ubiquitinated proteins, the entire IP gel is a “smear” as would be expected for polyubiquitination and we do not see a clear band for unmodified RBOHD (this is the expected result).

9. It is not clear what signal the authors quantify in Fig. 4e as one can hardly see any signal in either WT or T912A lines. Quantification does not make much sense in this case, and the difference between the lines is clear in Fig. 4d. Similarly, is the quantification shown in Fig. 2e really required/appropriate?

We took the signal from the entire western blot lane and quantified ubiquitinated RBOHD. The advantage of quantification is that this takes into account multiple independent experiments and blots. We have now clarified the approach in the figure legends.

Figure 2e: “After incubation with agarose or TUBE, lanes from the HA immunoblot were quantified in Image Lab, and samples were normalized to the intensity of the agarose control. Statistical differences were determined by student’s *t-test*, $p = 0.05$, $n = 4$ blots \pm SD”

Figure 4e: “Quantification of ubiquitination of RBOHD phosphomutants. After incubation with TUBE, lanes from the HA immunoblot were quantified in Image lab and samples were normalized to the intensity of WT RBOHD ubiquitination. Statistical differences were detected by student t test, $p = 0.05$, $n = 4$ blots \pm SD”

10. Please provide the curves for ROS measurements, so that one can compare the dynamic of this response between the lines tested.

We have included representative curves for ROS measurements in the new Supplemental Figure S5, which revealed that RBOHD^{T912A} lines exhibited a slightly shifted curve, with more robust ROS production at earlier time points compared with wild-type RBOHD complementation lines (Supplemental Fig 5a). The *pbl13*, *pire* and *pbl13 pire* knockouts also exhibited a shifted ROS curve, with more ROS production at earlier time points (Supplemental Fig 5c). Collectively, these data indicate that phosphorylation and ubiquitination of RBOHD can regulate both the amplitude and dynamics of ROS production upon flg22 perception. See page 11, lines 227-234 and page 19, lines 429-434.

11. Fig. S8: This is quite an important result and could be moved to main figures. Same comment for Fig. S1.

We have now included Figure S8 as a main figure (now Figure 5). We considered the request to move Figure S1 as a main figure, but decided to keep it as a supplemental figure. We have data in the main text demonstrating PBL13 and RBOHD co-IP (Fig. 1a) as well as direct interaction between PBL13 and RBOHD's C-terminus Figure 1b. Thus, we felt that the PBL13-RBOHD co-IP data as well as the yeast-two hybrid data in Figure S1 were best left as a supplemental figure.

12. There is higher RBOHD levels compared to Col-0 at 5 and 15 min after flg22 treatment in *pire* but not in *pbl13*. Would it suggest that RBOHD regulation by PIRE after flg22 treatment is PBL13-independent? In agreement with your answer to #4-18.

We thank the reviewer for their detailed observation. We have analyzed the repetitions of this experiment and there is variation in RBOHD accumulation after flg22 treatment when comparing the levels between *pire* and *pbl13*. Although both knockout lines have higher RBOHD accumulation compared to Col-0 before and after flg22 treatment, *pire* does not always have higher accumulation than *pbl13*. We have now included a different repeat of the experiment with more representative accumulation of RBOHD (these data are now shown in Figure 6b).

13. Lines 101-102: it is not correct to state generally that BIK1 is a “positive regulators of defense signaling” as it has recently been shown to act as a negative regulator of defense signaling mediated by LRR-RLPs (Wan et al., New Phytol. 2019). Similar comment for line 124, where it is wrongly indicated that PBL3 is the only RLCK acting a negative regulator of plant innate immune responses.

We have corrected these statements

14. The manuscript still requires significant editing for style and clarity, e.g.:

- a. All protein names should be consistently capitalized.
 - b. Line 79, nicotinamide should not be capitalized.
 - c. Lines 128-130 should be re-worded to make clear Lti6b was the control bait not prey.
 - d. Line 158 “The BIK1 kinase” should just be “BIK1”.
 - e. Line 202: “mimicking dephosphorylation” should rather be a “non-phosphorylatable mutant”, as it otherwise gives a false impression of dynamic regulation.
 - f. Line 220: “Due to the established cross-talk between phosphorylation and ubiquitination”; add reference(s).
- etc...

We have corrected these and carefully edited for style and clarity.

15. Fig 4c. please specific species names.

We have included the specific species names in the figure legend.

16. Mutant nomenclature is incorrect throughout the manuscript, e.g. “pbl13-2::pire-1” should be “pbl13 pire-1”

We have corrected this

17. Given the scale of Fig. 4e, it seems inappropriate to include a break in the y-axis to increase the apparent differences.

We have now adjusted the scale of Fig 4e.

Reviewers' comments:

Reviewer #1 (Remarks to the Author):

I am satisfied by the authors revisions and rebuttal. I have no further comments.

Reviewer #4 (Remarks to the Author):

We thank the authors for their further efforts to improve the manuscript. Here are a few final comments.

Related to our initial comment: "#4-21. Accumulation levels of RBOHD WT, T912D, and T912A should be compared in Col-0 and pire to determine whether accelerated degradation of RBOHD T912D is indeed PIRE-dependent."

We are thankful for the authors for performing additional experiments to address our comment. However, protoplast experiments failed to demonstrate that accelerated degradation of RBOHD T912D is indeed PIRE-dependent. While the protoplast experiments are interesting, without controls (co-transcribed marker gene, ConA treatment, etc), it is not possible to determine from transfection whether T912D is actually degraded or simply not expressing as well as the WT or T912A versions in protoplasts.

In response to our previous comments ("In Fig. 4c not only the blot with a-MBP should be shown but as well a blot with a-FLAG(Ub) antibodies to demonstrate that the laddering is indeed ubiquitination. The same way as it is done in Fig. 4a,b.", the authors replied: "Because PIRE is also present in the reactions, the anti-UB blot will not be able to differentiate between PIRE auto-ubiquitination as shown in Fig 4a, b or RBOHD trans-ubiquitination. For this reason we have referred to the laddering as opposed to an anti-UB blot."

We agree with the limitation of the assay. However, in this case, the authors can perform a-MBP IP, strong washes to release PIRE, and then demonstrate laddering by both a-MBP and a-FLAG IB. The type of laddering they show could be corresponding or not to ubiquitination. Only if one sees laddering by both a-tag and a-Ub antibodies, one can be sure it is ubiquitination. This is a small modification of the assay, and should be straight forward to do.

As the authors now clarified that Figure 4d corresponds to two different blots, they should include the a-TUBE IP blot developed with a-Ub antibodies to demonstrate that the enrichment for ubiquitinated proteins is comparable between the different samples they compare.

Minor comments:

- all protein names should be capitalized.
- line 351: should be made clear this is in protoplasts, not plants.

We thank reviewers for their helpful comments on the revised manuscript. All reviewers appreciated the extensive new experiments that we performed to address the issues raised during the last review. The three previous reviewers are satisfied with the manuscript with the exception of a few experiments for reviewer 4. We have now addressed all of reviewer 4's comments either textually or with new experimentation in Supplemental Figure 10c and d, Figure 4d, and Supplemental Figure 8.

Reviewer #1:

I am satisfied by the authors revisions and rebuttal. I have no further comments.

Reviewer #4 (Remarks to the Author):

We thank the authors for their further efforts to improve the manuscript. Here are a few final comments.

Related to our initial comment: “#4-21. Accumulation levels of RBOHD WT, T912D, and T912A should be compared in Col-0 and *pire* to determine whether accelerated degradation of RBOHD T912D is indeed PIRE-dependent.”

We are thankful for the authors for performing additional experiments to address our comment. However, protoplast experiments failed to demonstrate that accelerated degradation of RBOHD T912D is indeed PIRE-dependent. While the protoplast experiments are interesting, without controls (co-transcribed marker gene, ConA treatment, etc), it is not possible to determine from transfection whether T912D is actually degraded or simply not expressing as well as the WT or T912A versions in protoplasts.

Reviewer #4 requests additional experiments to distinguish if abundance levels of RBOHD and corresponding T912 phosphorylation mutants are different due to variations in RNA expression. Therefore, we performed additional assays to test the transcript accumulation of RBOHD in both Col-0 and *pire* protoplasts by qPCR. These new data are shown in Supplemental Figure 10 c and d. We were unable to detect a gross difference in transcript expression between *RBOHD WT*, *T912D* or *T912A* in *pire*. In Col-0, *RBOHD T912D* accumulated to a slightly higher level than WT or *T912A* (1.2 fold higher), but this small of a difference should not be biologically relevant (Nat Protoc 3, 1101–1108). Also, RBOHD T912D protein accumulation is lower than the other samples, so a slightly higher transcript expression is unlikely to account for the decrease detected at the protein level. Taken together, these data now demonstrate that the differences in accumulation are regulated at the protein level.

We also attempted to perform ConA treatments on transfected protoplasts many times. However, we found that ConA treatment (which contains DMSO), led to lysis of many protoplasts and thus western blotting was not possible. This is likely due to the increased cell permeability induced by DMSO treatment (Free Radical Research 42:5 435-441; Molecular Membrane Biology 29:3 107-113).

In response to our previous comments (“In Fig. 4c not only the blot with α -MBP should be shown but as well a blot with α -FLAG(Ub) antibodies to demonstrate that the laddering is indeed ubiquitination. The

same way as it is done in Fig. 4a,b.”, the authors replied: “Because PIRE is also present in the reactions, the anti-UB blot will not be able to differentiate between PIRE auto-ubiquitination as shown in Fig 4a, b or RBOHD trans-ubiquitination. For this reason we have referred to the laddering as opposed to an anti-UB blot.”

We agree with the limitation of the assay. However, in this case, the authors can perform a-MBP IP, strong washes to release PIRE, and then demonstrate laddering by both a-MBP and a-FLAG IB. The type of laddering they show could be corresponding or not to ubiquitination. Only if one sees laddering by both a-tag and a-Ub antibodies, one can be sure it is ubiquitination. This is a small modification of the assay, and should be straight forward to do.

Reviewer #4 requests additional experiments on figure 4c to distinguish between transubiquitination of RBOHD and autoubiquitination of PIRE. To confirm the laddering observed in Fig 4c is due to ubiquitination, we reconstituted the Ubiquitination reaction. After the reaction was complete, RBOHD proteins were purified with Amylose beads and the resulting protein subjected to anti-MBP and anti-FLAG immunoblot. The resulting data demonstrates that the difference in laddering between RBOHD WT and TTT-DDD is due to enhanced ubiquitination of the TTT-DDD variant. These data are now shown in Fig 4d.

As the authors now clarified that Figure 4d corresponds to two different blots, they should include the a-TUBE IP blot developed with a-Ub antibodies to demonstrate that the enrichment for ubiquitinated proteins is comparable between the different samples they compare.

Unfortunately, we no longer have these samples and both first authors have left the lab for postdoctoral positions during the long time that this paper has been under review. Previously, we demonstrated in Figure 2d that the anti-TUBE IP specifically enriches for ubiquitinated proteins. It is highly unlikely that there would be a gross difference in total ubiquitinated plant proteins between different TUBE IPs as we used the same amount of total protein sample for IPs (shown by equal loading in our coomassie gel at the bottom and also quantified by Bradford assay). We have now added all three replicates of the of the anti-TUBE IPs performed earlier as a new Supplemental Figure 8 and have also added a table with the data used in quantification after normalizing to total RBOHD accumulation.

Collectively, multiple lines of evidence in this manuscript support the findings that PIRE can ubiquitinate RBOHD and mimicking phosphorylation on T912 results in enhanced ubiquitination and decreased protein accumulation of RBOHD both *in vitro* and *in planta* (Fig 4c-g; Fig 5; Fig 7a-b; Supplementary Fig 7; Supplementary Fig 8).

Minor comments:

- all protein names should be capitalized.

We have corrected any errors in protein capitalization.

- line 351: should be made clear this is in protoplasts, not plants.

We have noted this experiment is performed in protoplasts.

REVIEWERS' COMMENTS:

Reviewer #4 (Remarks to the Author):

I am satisfied with the revisions performed by the authors.